# On Transferring Transferability: Towards a Theory for Size Generalization

**Eitan Levin**$^{\diamond}$
Caltech

**Yuxin Ma**$^{\diamond}$
Johns Hopkins University

**Mateo Díaz**
Johns Hopkins University

**Soledad Villar**
Johns Hopkins University

## Abstract

Many modern learning tasks require models that can take inputs of varying sizes. Consequently, dimension-independent architectures have been proposed for domains where the inputs are graphs, sets, and point clouds. Recent work on graph neural networks has explored whether a model trained on low-dimensional data can transfer its performance to higher-dimensional inputs. We extend this body of work by introducing a general framework for transferability across dimensions. We show that transferability corresponds precisely to continuity in a limit space formed by identifying small problem instances with equivalent large ones. This identification is driven by the data and the learning task. We instantiate our framework on existing architectures, and implement the necessary changes to ensure their transferability. Finally, we provide design principles for designing new transferable models. Numerical experiments support our findings.

## 1 Introduction

Modern learning problems often involve inputs of arbitrary size: language has text sequences of arbitrary length, graphs encode networks with an arbitrary number of nodes, and particle systems might have an arbitrary number of points. Although traditional models are typically limited to the specific input dimensions they were trained on, dimension-independent architectures have been proposed for several domains. For instance, graph neural networks (GNN) for graphs [70], DeepSet for sets [91], and PointNet for point clouds [66]. These models have a desirable feature: they can be trained with low-dimensional samples and, then, deployed on higher-dimensional data.

A key question in this context is whether the performance of a model trained with low-dimensional samples transfers across dimensions. This has been thoroughly studied in the context of GNNs [70, 58, 59], where the term *transferability* was first coined.[1] In turn, the tools used to establish transferability of GNNs are, at first sight, problem-specific, including the concept of a graphon, and certain convergence of homomorphism densities. Thus, the analysis does not readily extend beyond GNNs, and so we ask

> *What properties of the model, data, and learning task ensure that the learning performance transfers well across dimensions?*

---

$^{\diamond}$These authors contributed equally to this work.

[1] The term "transferability" is not related to transfer learning or meta-learning. It specifically refers to the phenomenon that consistency of outputs is preserved under size changes for structurally similar inputs, permitting direct performance carryover from small to large problems.

39th Conference on Neural Information Processing Systems (NeurIPS 2025).

This question is relevant for the current pursuit of graph foundation models [53], GNN for combinatorial optimization [12], neural PDE solvers [67, 51], and so on, where pre-trained models aim to be adapted to a wide range of tasks involving different-sized objects. In this paper, we answer this question by identifying assumptions on the learning problem that make a model generalize across dimensions, and give a theoretical framework to understand models with such properties. Our framework requires technical notions that we develop throughout the paper, so before diving into the details, we proceed to give the high-level intuition.

Our framework considers a machine learning model parametrized by $\theta \in \mathbb{R}^k$, where the number of parameters $k$ is fixed. For any fixed $\theta$, the model can be evaluated on inputs of any size $n$. That is, the learned $\theta$ defines a sequence of functions, for example, $(f_n : V_n \to \mathbb{R})_n$ with increasingly larger input sizes.[2] We think of $V_n \subseteq V_\infty$ as a subspace of an infinite-dimensional limit space. Transferability then amounts to asking whether for large enough $n$ and $m$ we have $f_n(x_n) \approx f_m(x_m)$ when the inputs $x_n$ and $x_m$ are close to each other in the limiting space $V_\infty$. For example, we consider graphs of different sizes to be close if their associated step graphons are close in the cut metric, following standard practice in the GNN literature. Similarly, we view point clouds of varying sizes as close if their corresponding empirical measures are close in the Wasserstein metric.

In a nutshell, we identify two properties that guarantee transferability: $(i)$ the functions in the sequence are *compatible* with one another in a precise sense, and $(ii)$ all of them are continuous with respect to a prescribed norm on $V_\infty$. Compatibility ensures that the sequence extends to a function taking infinite-dimensional inputs $f_\infty : V_\infty \to \mathbb{R}$, while continuity ensures that this extension is continuous. This allows us to relate model outputs on two differently-sized inputs via their extension $f_n(x_n) = f_\infty(x_n) \approx f_\infty(x_m) = f_m(x_m)$. Moreover, if we evaluate our models on a sequence of inputs $x_n$ converging in a suitable sense to an infinite-dimensional input $x$, then the evaluations $f_n(x_n)$ converge to the value of the limiting model $f_\infty(x)$. Our framework parallels the ideas from the GNN literature, where graph convergence to a limiting graphon is measured in cut distance—which is equivalent to homomorphism density convergence. However, this framework provides a transparent extension that allows us to analyze other models, e.g., DeepSets and PointNet.

**Our contributions.** Next, we summarize our three core contributions.

(*A framework for transferability*) We introduce a class of any-dimensional models and a formal definition of transferability. We show that transferability holds whenever certain notions of compatibility and continuity hold. We further leverage transferability to establish a generalization bound across dimensions. In our theory, the notion of transferability relies on identifying low-dimensional objects with higher-dimensional ones, and measuring similarity of such objects in an appropriate metric. We illustrate how these choices have to align with the data and learning tasks.

(*Transferability of existing models*) We instantiate our framework on several models, including DeepSet [91], PointNet [66], standard GNNs [70], Invariant Graph Networks (IGN) [56], and DS-CI [5]. Not all of these models satisfy our assumptions. For those that do, our framework yields new and existing transferability results in a unified fashion. For those that do not, we either modify them to ensure transferability or demonstrate numerically that transferability fails and impedes the performance of these models.

(*New transferable models*) We provide design principles for constructing transferable models. Leveraging these ideas, we develop a variant of IGN that is provably transferable, addressing challenges highlighted in previous work [56, 35]. We also design a new transferable model for point cloud tasks that is more computationally efficient than existing methods [5].

**Notation.** The Kronecker product on matrices is denoted by $\otimes$. We write $\mathbb{1}_n$ and $I_n$ for the all-ones vector and the identity matrix of size $n$ respectively. Given two sequences $(a_n), (b_n) \subseteq \mathbb{R}$, we say that $a_n \lesssim b_n$ if there is a constant $C > 0$ so that $a_n \leq Cb_n$ for all $n$. See also Appendix A.

**Outline.** Section 2 introduces the theoretical framework. Section 3 defines transferability, which is used in Section 4 to derive a generalization guarantee. Section 5 instantiates our framework on concrete neural network examples, and Section 6 provides supporting experiments.[3] Related work and limitations are discussed in the Appendix.

---

[2]The output size can also depend on $n$, which is often the case in representation learning.
[3]The code is available at https://github.com/yuxinma98/transferring_transferability.

## 2 How to consistently grow: equivalence of differently sized objects

In this section, we formally introduce the setting for studying transferability. The definitions here build upon the framework introduced in [49, 50] for defining any-dimensional neural networks and optimization problems. Central to our developments is the notion of compatibility between a sequence of functions. Intuitively, a sequence of functions is compatible when they respect the equivalence between low-dimensional objects and high-dimensional version of them. For example, sample statistics like the mean and standard deviation remain invariant when each element is repeated $m$ times. Similarly, various graph parameters such as triangle densities and normalized max-cut values are preserved when each vertex is replaced by $m$ identical copies of itself.

To formalize these notions of equivalence of objects and compatibility between functions, we will leverage the concept of a *consistent sequence*. Intuitively, they are nested sequences of vector spaces with growing groups acting on them, representing problem instances of different sizes. They are interconnected by embeddings that establish equivalence between smaller and larger instances.

**Definition 2.1.** *A **consistent sequence** $\mathbb{V} = \{(V_n)_{n \in \mathbb{N}}, (\varphi_{N,n})_{n \preceq N}, (\mathsf{G}_n)_{n \in \mathbb{N}}\}$ is a sequence of finite-dimensional vectors spaces $V_n$, maps $\varphi_{N,n}$ and groups $\mathsf{G}_n$ acting linearly on $V_n$, indexed by a directed poset[4] $(\mathbb{N}, \preceq)$, such that for all $n \preceq N$, the group $\mathsf{G}_n$ is embedded into $\mathsf{G}_N$, and $\varphi_{N,n}\colon V_n \hookrightarrow V_N$ is a linear, $\mathsf{G}_n$-equivariant embedding.*

Although the definition includes a sequence of groups, it applies to situations without symmetry by setting $\mathsf{G}_n = \{\mathrm{id}\}$ for all $n$. While all the examples in this work involve symmetries, we believe our framework might be relevant in symmetry-free settings.[5]

**Example 2.2.** *Here are two prototypical examples of consistent sequences. In both cases, $V_n = \mathbb{R}^n$, with the symmetric group $\mathsf{S}_n$ acting via coordinate permutation. $(i)$ the sequence indexed by $\mathbb{N}$ with the standard ordering $\leq$, paired with the zero-padding embedding $\mathbb{R}^n \hookrightarrow \mathbb{R}^N$, which maps $(x_1, \dots, x_n)$ to $(x_1, \dots, x_n, 0, \dots, 0)$; and $(ii)$ the sequence indexed by $\mathbb{N}$ with the divisibility ordering ($n \preceq N$ whenever $N$ is divisible by $n$) paired with the duplication embedding $\mathbb{R}^n \hookrightarrow \mathbb{R}^N$, sending a vector $x = (x_1, \dots, x_n)$ to $x \otimes \mathbb{1}_{N/n} = (\underbrace{x_1, \dots, x_1}_{N/n \text{ copies}}, \dots, \underbrace{x_n, \dots, x_n}_{N/n \text{ copies}})$.*

Next, we define a common space for objects of all sizes in a consistent sequence, in which we can compare objects of different sizes and take their limits. For example, if we identify vectors $x$ with their duplication embeddings $x \otimes \mathbb{1}_{N/n}$, we can view the resulting equivalence classes as step functions on $[0, 1]$ taking value $x_i$ on consecutive intervals of length $1/n$.

**Definition 2.3.** *Let $V_\infty = \bigcup V_n$ where we identify $v \in V_n$ with its image $\varphi_{N,n}(v)$ for all $n \preceq N$ to be equivalent. Analogously, we let $\mathsf{G}_\infty$ be the union of groups with equivalent identifications.*

For simplicity, we will often write $v \in V_\infty$ for any finite-dimensional object $v \in V_n$ to denote the equivalence class $[v]$. It is straightforward to check that $G_\infty$ acts on $V_\infty$ via $g \cdot [v] = [g \cdot v]$. Having established how consistent sequences define a desirable equivalence relation in the space of problem instances of varying sizes, we now introduce compatible functions that respect such equivalence.

**Definition 2.4.** *Let $\mathbb{V} = \{(V_n), (\varphi_{N,n}), (\mathsf{G}_n)\}$ and $\mathbb{U} = \{(U_n), (\psi_{N,n}), (\mathsf{G}_n)\}$ be two consistent sequences indexed by $(\mathbb{N}, \preceq)$. A sequence of maps $(f_n\colon V_n \to U_n)$ is **compatible** with respect to $\mathbb{V}, \mathbb{U}$ if $f_N \circ \varphi_{N,n} = \psi_{N,n} \circ f_n$ for all $n \preceq N$, and each $f_n$ is $\mathsf{G}_n$-equivariant.*

It is instructive to visualize compatible maps using the following commutative diagram, which represents a mapping between sequences while ensuring that the diagram commutes:

$$
\begin{array}{ccccccc}
\dots \xhookrightarrow{\cdots} (V_{n_1}, \mathsf{G}_{n_1}) & \xrightarrow{\varphi_{n_2,n_1}} & (V_{n_2}, \mathsf{G}_{n_2}) & \xrightarrow{\varphi_{n_3,n_2}} & (V_{n_3}, \mathsf{G}_{n_3}) & \xhookrightarrow{\cdots} \dots & \\
\Big\downarrow{\scriptstyle f_{n_1}} & & \Big\downarrow{\scriptstyle f_{n_2}} & & \Big\downarrow{\scriptstyle f_{n_3}} & & \text{for } n_1 \preceq n_2 \preceq \dots \\
\dots \xhookrightarrow{\cdots} (U_{n_1}, \mathsf{G}_{n_1}) & \xrightarrow{\psi_{n_2,n_1}} & (U_{n_2}, \mathsf{G}_{n_2}) & \xrightarrow{\psi_{n_3,n_2}} & (U_{n_3}, \mathsf{G}_{n_3}) & \xhookrightarrow{\cdots} \dots &
\end{array}
$$

Equivalently, compatible maps are precisely those sequences $(f_n)$ that extend to equivalence classes:

---

[4]This is a partial order on $\mathbb{N}$ where every two elements have an upper bound, see Appendix C.1.

[5]The symmetric case is relevant because the theoretical setting relates to representation stability [50, 16].

> *A sequence of maps $(f_n)$ is compatible if, and only if, there exists a $\mathsf{G}_\infty$-equivariant map $f_\infty \colon V_\infty \to U_\infty$ such that $f_n = f_\infty \mid_{V_n}$ for all $n$.*

See Appendix C.2 for a proof of this equivalence.

## 2.1 How to define distances across dimensions

Compatible maps are functions that respect the equivalence defined by consistent sequences. We now consider whether these functions further preserve proximity, mapping "nearby" objects to "nearby" outputs. This requires a well-defined notion of distance between objects of different sizes that is consistent with the earlier equivalence.

**Definition 2.5.** *For a consistent sequence $\mathbb{V}$, a sequence of norms $(\| \cdot \|_{V_n})$ on $V_n$ are **compatible** if all the embeddings $\varphi_{N,n}$ and the $\mathsf{G}_n$-actions are isometries. i.e., for all $n \preceq N, x \in V_n, g \in \mathsf{G}_n$, $\|\varphi_{N,n} x\|_{V_N} = \|x\|_{V_n}$ and $\|g \cdot x\|_{V_n} = \|x\|_{V_n}$.*

Equivalently, compatible norms are those that extend to the limit; that is, there exists a norm $\| \cdot \|_{V_\infty}$ on $V_\infty$ such that for any $n$ and $x \in V_n$, $\|x\|_{V_n} = \|x\|_{V_\infty}$, and the $\mathsf{G}_\infty$-action on $V_\infty$ is an isometry with respect to $\| \cdot \|_{V_\infty}$; see Appendix C.3 for a proof of this statement.

**Example 2.6** (Example 2.2 continued)**.** *For our prototypical consistent sequences, the $\ell_p$ norms $\|x\|_p := \left(\sum_i |x_i|^p\right)^{1/p}$ are compatible with the zero-padding embeddings, while the normalized $\ell_p$ norms $\|x\|_{\overline{p}} := \left(\frac{1}{n} \sum_i |x_i|^p\right)^{1/p}$ are compatible with the duplication embeddings.*

With norms in place, we can define a limit space that includes not only equivalence classes of finite-dimensional objects, but also their limits. As we will see, this recovers meaningful spaces such as $L^p([0,1])$ and the space of graphons with the cut norm. Moreover, the orbit spaces of these limits recover probability measures (with the Wasserstein distance) and equivalence classes of graphons (with the cut distance), respectively.

**Definition 2.7.** *The **limit space** is the pair $(\overline{V_\infty}, \mathsf{G}_\infty)$ where $\overline{V_\infty}$ denotes the completion of $V_\infty$ with respect to $\| \cdot \|_{V_\infty}$, endowed with the **symmetrized metric**[6]*

$$\overline{\mathrm{d}}(x,y) := \inf_{g \in \mathsf{G}_\infty} \|g \cdot x - y\|_{V_\infty} \quad \text{for } x, y \in \overline{V_\infty}.$$

## 3 Transferability is *just* continuity

The notion of GNN transferability was established in [70, 48]. It states that the output discrepancy $\|f_n(A_n, X_n) - f_m(A_m, X_m)\|$ between graph signals of sizes $n, m$ sampled from the same graphon signal $(W, f)$ vanishes as $n, m$ grow. Prior studies explore this property under various sampling schemes, norms, and GNN architectures [41, 70, 47, 11, 35].

We formalize the idea of functions "mapping close objects to close outputs" as (Lipschitz) continuity of compatible maps.

**Definition 3.1.** *Let $\mathbb{V}, \mathbb{U}$ be consistent sequences endowed with norms. A sequence of equivariant maps $(f_n \colon V_n \to U_n)$ is **continuously (respectively, $L$-Lipschitz, $L(r)$-locally Lipschitz) transferable** if there exists $f_\infty \colon \overline{V_\infty} \to \overline{U_\infty}$ that is continuous (respectively, $L$-Lipschitz, $L(r)$-Lipschitz on the ball $B(0,r) = \{v \in \overline{V_\infty} : \|v\|_{V_\infty} < r\}$ for all $r > 0$) with respect to $\| \cdot \|_{V_\infty}, \| \cdot \|_{U_\infty}$, such that $f_n = f_\infty|_{V_n}$ for all $n$. Notice that if $(f_n)$ is transferable, then it must be compatible.*

In Appendix D.1 we show that Lipschitz transferability is equivalent to having a compatible sequence $(f_n)$, each Lipschitz with the same constant. For linear maps, it suffices to verify that the operator norms are uniformly bounded. This simplifies the verification of transferability. Importantly, we also show that transferability with respect to the norm $\| \cdot \|_{V_\infty}$ implies transferability with respect to the symmetrized metric $\overline{\mathrm{d}}$.

---

[6]It is a pseudometric on $\overline{V_\infty}$, and a metric on the space whose points are closures of orbits under the action of $\mathsf{G}_\infty$ in $\overline{V_\infty}$ (see Section 3 of [10]).

The following proposition shows that the output discrepancy is controlled whenever a dimension-independent model is continuously or Lipschitz transferable, justifying our terminology. Thus, our framework extends transferability beyond GNNs, highlighting that transferability is equivalent to continuity in the limit space. For a given function $R \colon \mathbb{N} \to \mathbb{R}_+$, we say that $(x_n)$ converges to $x$ in $V_\infty$ at a rate $R(n)$ in $\overline{\mathrm{d}}$ if $\overline{\mathrm{d}}(x_n, x) \lesssim R(n)$ and $R(n) \to 0$.

**Proposition 3.2.** *Let $\mathbb{V}, \mathbb{U}$ be consistent sequences and let $(f_n \colon V_n \to U_n)$ be maps between them.*

*(Transferability) For any sequence $(x_n \in V_n)$ converging to a limiting object $x \in \overline{V_\infty}$ in $\overline{\mathrm{d}}$, if $(f_n)$ is continuously transferable, then $\overline{\mathrm{d}}(f_n(x_n), f_m(x_m)) \to 0$ as $n, m \to \infty$. Furthermore, if $(x_n)$ converges to $x$ at rate $R(n)$ and $(f_n)$ is locally Lipschitz-transferable, then*

$$\overline{\mathrm{d}}(f_n(x_n), f_m(x_m)) \lesssim R(n) + R(m).$$

*(Stability) If $(f_n)$ is $L(r)$-locally Lipschitz transferable, then for any two inputs $x_n \in V_n$ and $x_m \in V_m$ of any two sizes $n, m$ with $\|x_n\|_{V_n}, \|x_m\|_{V_m} \le r$, we have $\overline{\mathrm{d}}(f_n(x_n), f_m(x_m)) \le L(r) \overline{\mathrm{d}}(x_n, x_m)$.*

Several remarks are in order. First, the same results trivially hold when the symmetrized metric is replaced by the norms on $V_\infty, U_\infty$. However, the finite inputs $(x_n)$ are often obtained from the limiting object via random sampling, and converge only in the symmetrized metric. This is the case for all the common sampling strategies used for sets, graphs, and point clouds, which we review in Appendix D.3. In the case of graphs, our framework recovers results from prior work, achieving the same or stronger rate of convergence. Second, some models are only locally Lipschitz outside of a measure-zero set. Transferability still holds for such models for almost all limit objects $x \in \overline{V_\infty}$. This and other extensions are deferred to Appendix D.2. Third, Lipschitz continuity of neural networks has been extensively studied in the context of model *stability* and *robustness* to small input perturbations [25, 85, 43, 27]. Our results generalize the connection between stability and transferability, first established in [72], by showing that Lipschitz continuity of an any-dimensional model implies not only stability, but also *transferability* across input sizes.

## 4   Transferability implies size generalization

We use transferability to derive a generalization bound for models trained on inputs of fixed-size $n$, and evaluate their performance as $n \to \infty$. It can therefore be interpreted as a *size generalization bound*, accounting for distributional shifts induced by size variation. We study a supervised learning task with input and output spaces modeled by consistent sequences $\mathbb{V}$ and $\mathbb{U}$ satisfying the following.

**Assumption 4.1.** *Let $\mu$ be a probability distribution supported on a product of subsets $X \times Y \subseteq \overline{V_\infty} \times \overline{U_\infty}$ whose orbit closures are compact in the symmetrized metrics. Let $\mathcal{S}_n \colon \overline{V_\infty} \times \overline{U_\infty} \to V_n \times U_n$ be a random sampling procedure such that $\mathcal{S}_n(x, y) \in X \times Y$ almost surely for $(x, y) \in \mathrm{supp}(\mu)$. This sampling induces a distribution $\mu_n$ on $V_n \times U_n$ by drawing $(x, y) \sim \mu$ and then sampling $(x_n, y_n) \sim \mathcal{S}_n(x, y)$. A dataset $s = \{(x_i, y_i)\}_{i=1}^N$ is drawn i.i.d. from $\mu_n$. Suppose the loss function $\ell \colon \overline{U_\infty} \times \overline{U_\infty} \to \mathbb{R}$ is bounded by $M$ and $c_\ell$-Lipschitz. Assume training is performed using a locally Lipschitz transferable neural network model, and $\mathcal{A}_s \colon \overline{V_\infty} \to \overline{U_\infty}$ is the hypothesis learned (given dataset $s$), which is $c_s$-Lipschitz on $X \times Y$ in $\overline{\mathrm{d}}$.*

In Appendix E, we further break down these technical assumptions and provide detailed motivation to make them more accessible. We also discuss two concrete examples of sets and graphs to justify the practicality of the key assumptions. The following generalization bound follows by applying the results from [88] to our setting.

**Proposition 4.2.** *Consider a learning task with input and output spaces modeled by consistent sequences $\mathbb{V}$ and $\mathbb{U}$ satisfying Assumption 4.1. For any $\delta > 0$, with probability at least $1 - \delta$,*

$$\left| \frac{1}{N} \sum_{i=1}^N \ell(\mathcal{A}_s(x_i), y_i) - \mathbb{E}_{(x,y) \sim \mu} \ell(\mathcal{A}_s(x), y) \right|$$
$$\le c_l(c_s \vee 1) \left( \xi^{-1}(N) + W_1(\mu, \mu_n) \right) + M \sqrt{(2 \log 2)\xi^{-1}(N)^2 + \frac{2 \log(1/\delta)}{N}}$$

(1)

where $\xi(r) := \frac{C_X(r/4)C_Y(r/4)}{r^2}$, $C_X(\varepsilon)$ and $C_Y(\varepsilon)$ are the $\varepsilon$-covering numbers in $\overline{\mathrm{d}}$ of $X$ and $Y$, respectively, and $W_1$ denotes the Wasserstein-1 distance. Moreover, if the sampling $\mathcal{S}_n$ has convergence rate $R(n)$ in expectation, then $W_1(\mu, \mu_n) \lesssim R(n)$ as $n \to \infty$ and the bound (1) converges to 0 as $n, N \to \infty$.

We defer the proof to Appendix E. This bound shows that generalization improves with greater model transferability (i.e., smaller Lipschitz constants) and higher-dimensional training data (i.e., large $n$), while it deteriorates with increasing complexity of the data space (i.e., larger covering numbers).

## 5   Transferable neural networks

We now apply our general framework to the settings of sets, graphs, and point clouds. In each case, we identify suitable consistent sequences, analyze the compatibility and transferability of existing neural network models with respect to these sequences, and propose a principled recipe for designing new transferable models. This analysis provides new insights into the tasks for which each model is best suited, while offering provable size-generalization guarantees.

> *A transferable neural network learns a fixed set of parameters $\theta$ that define transferable functions $(f_n)$ for any $n$-dimensional normed space $V_n$. The compatibility conditions on $(f_n)$ and the sequence of norms $\|\cdot\|_{V_n}$ describes the implicit inductive bias of the model.*

Our analysis hinges on the following proposition, which observes that compatibility and transferability are preserved under composition and reduces the verification of Lipschitz continuity from the limit space to each finite-dimensional space. This significantly simplifies the task of establishing transferability in complex neural networks. The following result is proved in Appendix E.1.

**Proposition 5.1.** *Let $(V_n^{(i)})_n, (U_n^{(i)})_n$ be consistent sequences for $i = 1, \ldots, D$. For each $i$, let $(W_n^{(i)}\colon V_n^{(i)} \to U_n^{(i)})$ be linear maps and $(\rho_n^{(i)}\colon U_n^{(i)} \to V_n^{(i+1)})$ be nonlinearities, and assume that $(W_n^{(i)}), (\rho_n^{(i)})$ are compatible, that $\sup_{n,i} \|W_n^{(i)}\|_{\mathrm{op}} < \infty$, and that $\rho_n^{(i)}$ is $L(r)$-Lipschitz on balls $B(0, r)$ in $U_n^{(i)}$ for all $r > 0$ and all $n$. Then the sequence of neural networks $(\widehat{f}_n = W_n^{(D)} \circ \rho_n^{(D-1)} \circ \ldots \circ \rho_n^{(1)} \circ W_n^{(1)})$ is $\widehat{L}(r)$-locally Lipschitz transferable for explicit $\widehat{L}(r)$.*

### 5.1   Sets

**Consistent sequence of sets.** We have described in Examples 2.2 and 2.6 two consistent sequences that formalize equivalence across sets of varying sizes: the *zero-padding consistent sequence* $\mathbb{V}_{\mathrm{zero}}$ and the *duplication consistent sequence* $\mathbb{V}_{\mathrm{dup}}$, along with norms on $\mathbb{R}^n$ that are compatible with each sequence. The resulting limit spaces and symmetrized metrics recover interesting mathematical structures, as summarized in Table 1. Further details are provided in Appendix F.1. See Figure 1 for an illustration.

| Consistent Sequence | Limit space | Orbit closures of $\overline{V_\infty}$ | Orbits of $V_\infty$ |
|---|---|---|---|
| $\mathbb{V}_{\mathrm{zero}}$ | $\ell_p$ | Ordered sequences in $\ell_p$ | Ordered sequences with finitely-many nonzeros |
| $\mathbb{V}_{\mathrm{dup}}$ | $L^p([0,1])$ | Probability measures on $\mathbb{R}$ with Wasserstein $p$-metric | Empirical measures |

Table 1: Limit spaces and their orbit spaces for consistent sequences of sets.

**Permutation-invariant neural networks on sets.** Prominent invariant neural networks on multi-sets including *DeepSet* [91], *normalized DeepSet* [9], and *PointNet* [66] take the form $f_n(X) = \sigma\left(\mathrm{Agg}_{i=1}^n \rho(X_{i:})\right)$, where $\rho\colon \mathbb{R}^d \to \mathbb{R}^h$, $\sigma\colon \mathbb{R}^h \to \mathbb{R}$ are fully-connected neural networks, both independent of $n$, and $\mathrm{Agg}$ denotes a permutation-invariant aggregation. Sum ($\mathrm{Agg}_{i=1}^n := \sum_{i=1}^n$), mean ($\mathrm{Agg}_{i=1}^n := \frac{1}{n} \sum_{i=1}^n$) and max ($\mathrm{Agg}_{i=1}^n := \max_{i=1}^n$) aggregations yield DeepSet, normalized DeepSet and PointNet, respectively. Using Proposition 5.1, we examine the transferability of these models, demonstrating that they parameterize functions on different limit spaces, and hence are suitable for different tasks, see Corollary 5.2, Table 2, and Figure 2. This is proved in Appendix F.2.

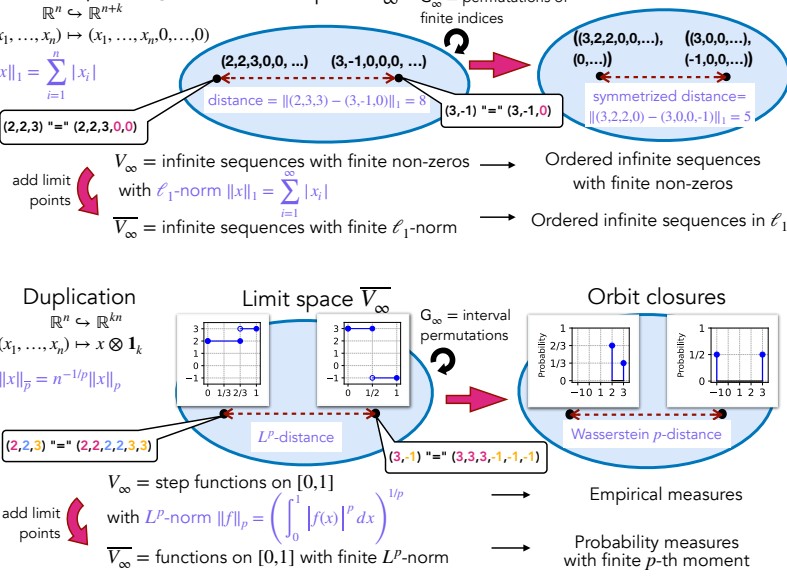

Figure 1: Two examples of consistent sequences on sets. *(top)* Zero-padding consistent sequence for sets. *(bottom)* Duplication consistent sequence for sets.

**Corollary 5.2.** *The transferability of the three models are summarized in Table 2. Particularly, they extend to Lipschitz functions* $\text{DeepSet}_\infty : l_1(\mathbb{R}^d) \to \mathbb{R}$, $\overline{\text{DeepSet}}_\infty : \mathcal{P}_1(\mathbb{R}^d) \to \mathbb{R}$, $\text{PointNet}_\infty : \mathcal{P}_\infty(\mathbb{R}^d) \to \mathbb{R}$ *given by*

$$\text{DeepSet}_\infty((x_i)) = \sigma\left(\sum_{i=1}^\infty \rho(x_i)\right), \overline{\text{DeepSet}}_\infty(\mu) = \sigma\left(\int \rho d\mu\right), \text{PointNet}_\infty(\mu) = \sigma\left(\sup_{x \in \text{supp}(\mu)} \rho(x)\right).$$

| Model | $\mathbb{V}_{\text{zero}}, \|\cdot\|_1$ | $\mathbb{V}_{\text{dup}}, \|\cdot\|_{\overline{p}}$ |
|---|---|---|
| DeepSet | Lipschitz transferable if $\rho(0) = 0$ | Incompatible |
| $\overline{\text{DeepSet}}$ | Incompatible | Lipschitz transferable |
| PointNet | Incompatible | Compatible, Lipschitz transferable if $p = \infty$ |

Table 2: Transferability of invariant neural networks on sets, assuming the nonlinearities $\sigma, \rho$ are Lipschitz.

## 5.2 Graphs

**Consistent sequences for graph signals.** To model graph signals, we consider the sequence of vector spaces $V_n = \mathbb{R}^{n \times n}_{\text{sym}} \times \mathbb{R}^{n \times d}$, representing the adjacency matrix of a weighted graph with $n$ nodes and node features of dimension $d$. The symmetric group $\mathsf{S}_n$ acts on $(A, X) \in V_n$ by $g \cdot (A, X) = (gAg^\top, gX)$, capturing the invariance to node ordering.

While other embeddings are possible, in this work we focus on the *duplication-consistent sequence* $\mathbb{V}^G_{\text{dup}}$ for graphs, illustrated in Figure 3. Specifically, for any $n|N$, the embedding is defined by $\varphi_{N,n}(A, X) = \left(A \otimes \mathbb{1}_{N/n}\mathbb{1}^\top_{N/n}, X \otimes \mathbb{1}_{N/n}\right)$, which corresponds to replacing each node with $N/n$ duplicated copies. These embeddings precisely identify graph signals that induce the same step graphon signal. We consider three compatible norms: the $p$-norm $\|(A, X)\|_{\overline{p}} = \left(\frac{1}{n^2}\sum_{i,j}|A_{ij}|^p\right)^{1/p} + \left(\frac{1}{n}\sum_i \|X_{i:}\|^p_{\mathbb{R}^d}\right)^{1/p}$, the operator $p$-norm $\|(A, X)\|_{\text{op},p} := \frac{1}{n}\|A\|_{\text{op},p} + \left(\frac{1}{n}\sum_i \|X_{i:}\|^p_{\mathbb{R}^d}\right)^{1/p}$, and the cut norm $\|(A, X)\|_\square := \|A\|_\square + \|X\|_\square$. In all cases, the limit space is the space of graphon signals and the symmetrized metrics recover extensively studied graphon distances in [54, 47].

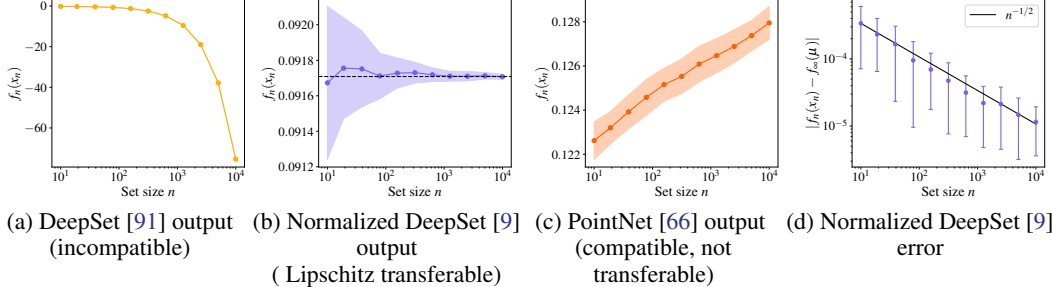

(a) DeepSet [91] output (incompatible)

(b) Normalized DeepSet [9] output ( Lipschitz transferable)

(c) PointNet [66] output (compatible, not transferable)

(d) Normalized DeepSet [9] error

Figure 2: Transferability of invariant networks on sets under $(\mathbb{V}_{\mathrm{dup}}, \|\cdot\|_{\overline{1}})$. The plots show outputs of untrained, randomly initialized models on input sets of increasing size $n$. Each set consists of $n$ i.i.d. samples from $\mathcal{N}(0,1)$, a distribution with non-compact support. Error bars indicate one standard deviation above and below the mean over 100 random samples. *(a)(b)(c)*: Model output $f_n(X_n)$ vs. set size $n$. For normalized DeepSet, the dashed line represents the limiting value $f_\infty(\mu) = \sigma\left(\int \rho(x)\, d\mu(x)\right)$ for $\mu = \mathcal{N}(0,1)$, computed via numerical integration. While the outputs of DeepSet and PointNet diverge as $n$ increases, the transferable model, normalized DeepSet, converges to the theoretical limit, i.e., $f_n(X_n) \to f_\infty(\mu)$. *(d)*: Convergence error $|f_n(X_n) - f_\infty(\mu)|$ vs. set size $n$ for normalized DeepSet (both axes in log scale), demonstrating the expected $O(n^{-1/2})$ convergence rate as predicted by Proposition 3.2. See Appendix F.2 for further discussion.

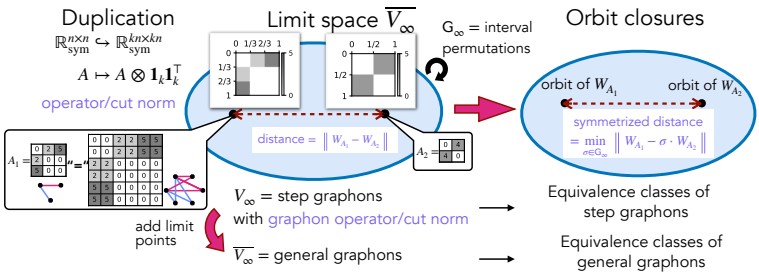

Figure 3: Duplication consistent sequence for graphs

***Message Passing Neural Networks (MPNNs)*** Message-passing-based GNNs form the most widely used and general paradigm, encompassing many existing models [31]. Instantiating Proposition 5.1, we analyze in Appendix G.2 the transferability of MPNNs under constraints on the message function, update function, and local aggregation, and compare our results with [70, 47].

**Making *Invariant Graph Networks* (IGN) Transferable** IGNs [56] are an alternative approach to designing GNNs by alternating linear $\mathsf{S}_n$-equivariant layers with pointwise nonlinearities. IGN is incompatible with respect to $\mathbb{V}_{\mathrm{dup}}^G$ because the linear layers used in [56] are incompatible. Nevertheless, the hypotheses in Proposition 5.1 provide a systematic approach for modifying IGN to achieve transferability, leading to two newly proposed models. We highlight this recipe for constructing transferable neural networks is general.

*Generalizable Graph Neural Network (GGNN):* We use compatible linear layers and message-passing-like nonlinearities of the form $(A, X) \mapsto \left(A, \sigma\left(\sum_{s=0}^{S} n^{-s} A^s X_s\right)\right)$, where $\sigma$ acts entrywise. The GGNNs are compatible and at least as expressive as the GNNs in [70].

*Continuous GGNN:* We restrict the linear layers in GGNN to the subspace that has bounded operator norm on the limit space. This model is transferable in both operator 2-norm and the cut norm.

The complete description of these models and proof of their transferability are in Appendix G.3. The transferability of various GNN models with respect to $(\mathbb{V}_{\mathrm{dup}}^G, \|\cdot\|_{\mathrm{op},p}), p = 2$ is illustrated in Figure 4 through a numerical experiment.

### 5.3 Point clouds

**Consistent sequences for point clouds.** For $k$-dimensional point clouds, we consider a sequence of vector spaces $V_n = \mathbb{R}^{n \times k}$, representing sets of $n$ points in $\mathbb{R}^k$, where $k$ is fixed (typically $k = 2$

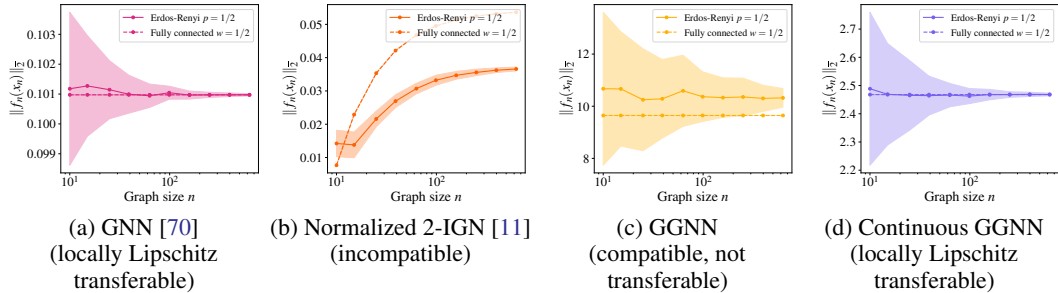

(a) GNN [70]
(locally Lipschitz
transferable)

(b) Normalized 2-IGN [11]
(incompatible)

(c) GGNN
(compatible, not
transferable)

(d) Continuous GGNN
(locally Lipschitz
transferable)

Figure 4: Transferability of equivariant GNNs with respect to $(\mathbb{V}_{\mathrm{dup}}^G, \|\cdot\|_{\mathrm{op},2})$. The plots show outputs of untrained, randomly initialized models for two sequences of input graph signals $(A_n, X_n)$: (dashed lines) Fully-connected weighted graphs $A_n = \frac{\mathbb{1}_n \mathbb{1}_n^\top}{2}$, $X_n = \mathbb{1}_n$. (solid lines) $A_n$ is drawn i.i.d. from the Erdős–Rényi model $G(n, 1/2)$, with $X_n = \mathbb{1}_n$. These two sequences represent different samplings of the same underlying constant graphon signal, where $W \equiv 1/2$ and $f \equiv 1$. Error bars indicate one standard deviation above and below the mean over 100 random samples. *(dashed lines)*: For the fully connected model each finite graph signal exactly induces the underlying graphon signal. The outputs of all compatible models ((a), (c), (d)) remain constant over $n$, whereas the incompatible model (b) does not. *(solid lines)*: The outputs of all transferable models ((a), (d)) converge to the same limit as Sequence (1), while the discontinuous model (c) does not.

or 3). Unlike the sets models in Section 5.1, here we consider not only the permutation symmetry $\mathsf{S}_n$, which acts on the rows and ensures invariance to the ordering of points, but also the additional symmetry given by the orthogonal group $\mathsf{O}(k)$, which acts on the columns to capture rotational and reflectional symmetries in the Euclidean space $\mathbb{R}^k$. i.e. $(g, h) \cdot X = gXh^\top$.

We consider the duplication consistent sequences of point clouds $\mathbb{V}_{\mathrm{dup}}^P$. Specifically, for any $n \mid N$, the embedding is defined as $\varphi_{N,n}(X) = X \otimes \mathbb{1}_{N/n}$. The group embedding $\mathsf{S}_n \times \mathsf{O}(k) \hookrightarrow \mathsf{S}_N \times \mathsf{O}(k)$ is given by $(g, h) \mapsto (g \otimes I_{N/n}, h)$. We further endow each $V_n$ with the normalized $\ell_p$ norm $\|X\|_{\overline{p}} = \left( \frac{1}{n} \sum_{i=1}^n \|X_{i:}\|_2^p \right)^{1/p}$. The orbit closures of $\overline{V_\infty}$ can be identified with the orbit space under $\mathsf{O}(k)$ of probability measures on $\mathbb{R}^k$ with finite $p$th moment.

**Invariant neural networks on point clouds.** First, we analyze the *DeepSet for Conjugation Invariance* (DS-CI) proposed in [5]. In Appendix H.2, we show that the normalized variants of DS-CI is "approximately" locally Lipschitz transferable with respect to $(\mathbb{V}_{\mathrm{dup}}^P, \|\cdot\|_{\overline{p}})$.

We also introduce a more time and space-efficient model, *normalized SVD-DeepSet*, defined as: $\overline{\mathrm{SVD\text{-}DS}}_n(X) = \overline{\mathrm{DeepSet}}_n(XV)$, where for an input point cloud $X \in \mathbb{R}^{n \times k}$, we compute its singular value decomposition $X = U\Sigma V^\top$ with ordered singular values. This model effectively applies a canonicalization step with respect to the $\mathsf{O}(k)$ action. We prove that it is locally Lipschitz-transferable outside of a zero-measure set (which exists for any canonicalization [23]) in Appendix H.3. In Appendix H.3, we illustrate the transferability of these models when input point clouds are i.i.d. samples from an $\mathsf{O}(k)$-invariant measure on $\mathbb{R}^k$ through a numerical experiment.

## 6 Size generalization experiments

Our theoretical framework emphasizes understanding how solutions to small problem instances inform solutions to larger ones, i.e., whether the target function is compatible with respect to a given consistent sequence, and continuous with respect to an associated norm. Under this view, size generalization can be provably achieved using a neural network model that is *aligned* with the target function: *Compatibility alignment* ensures that the model is compatible with respect to the same consistent sequence as the target function, and yields a function on the correct limit space. *Continuity alignment* further requires the model to be continuously transferable with respect to the same norm as the target function, providing stronger inductive bias and asymptotic guarantees.

We evaluate the effect of these alignments through experiments where models are trained on small inputs of fixed size $n_{\mathrm{train}}$ and tested on larger inputs $n_{\mathrm{test}} \geq n_{\mathrm{train}}$. In each experiment, all models have approximately the same number of parameters to ensure fair comparisons. In this Section we show a selected set of results. See Appendix I for a full discussion.

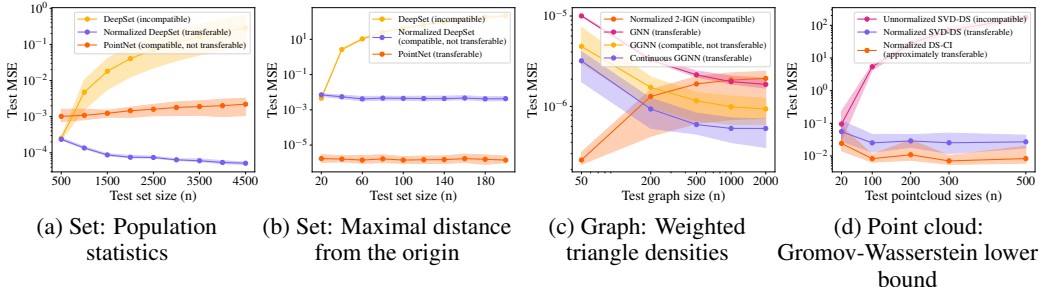

| (a) Set: Population statistics | (b) Set: Maximal distance from the origin | (c) Graph: Weighted triangle densities | (d) Point cloud: Gromov-Wasserstein lower bound |

Figure 5: Size generalization experiments: Mean test MSE (over 10 random runs) against test input dimensionality $n$. Error bars indicate the min/max range in (a)(b)(d), and $20^{th}/80^{th}$ percentiles in (c) for legibility.

**Size generalization on sets.** We consider two learning tasks on sets of arbitrary size, where the target functions exhibit distinct properties, leading to different models being more suitable.

*Experiment 1: Population Statistics.* We tackle Task 3 from Section 4.1.1 of [91] (other tasks yield similar results). The dataset consists of $N$ sample sets. We first randomly choose a unit vector $v \in \mathbb{R}^{32}$. For each set, we sample a parameter $\lambda \in [0, 1]$ and then generate a set of $n$ points from $\mu(\lambda) = \mathcal{N}(0, I + \lambda vv^\top)$. The target function involves learning the mutual information, which depends on the underlying probability measure $\mu(\lambda)$ and is continuous with respect to the Wasserstein $p$-distance. Based on the transferability analysis in Table 2, normalized DeepSet is well-suited for this task, as it aligns with the target function at the continuity level, whereas PointNet aligns only at the compatibility level, and DeepSet lacks alignment at either level. Indeed, normalized DeepSet has the best in-distribution test performance and size generalization behavior as shown in Figure 5a.

*Experiment 2: Maximal Distance from the Origin.* In this task, each dataset consists of $N$ sets where each set contains $n$ two-dimensional points sampled as follows. First, a center is sampled from $\mathcal{N}(0, I_2)$ and a radius is sampled from $\mathrm{Unif}([0, 1])$, which together define a circle. The set then consists of $n$ points sampled uniformly along the circumference. The goal is to learn the maximum Euclidean norm among the points in each set. The target function in this case only depends on a point cloud via its support, and is continuous with respect to the Hausdorff distance. Consequently, for this task, PointNet is well-suited, as it aligns with the task at the level of continuity. (It was proved in [9] that PointNet extends to a function that is continuous with respect to the Hausdorff distance. We discuss the relationship between this result and ours in the PointNet part of Appendix F.2.) In contrast, normalized DeepSet aligns only at the compatibility level, while DeepSet remains unaligned. The comparison of various model's performance, shown in Figure 5b is as expected.

**Size Generalization on Graphs.** The dataset consists of $N$ attributed graphs $(A, x)$, where $A_{ij} = A_{ji} \overset{i.i.d}{\sim} \mathrm{Unif}([0, 1])$ for $i \leq j$, and $x_i \overset{i.i.d}{\sim} \mathrm{Unif}([0, 1])$. (We also experimented with simple graphs; see Appendix I.2 for details.) The task is to learn the signal-weighted triangle density $y_i = \frac{1}{n^2} \sum_{j,k \in [n]} A_{ij} A_{jk} A_{ki} x_i x_j x_k$, which depends on the underlying graphon and is continuous with respect to the cut norm. Figure 5c shows that our proposed continuous GGNN, which aligns with the target function at the continuity level, achieves the best test performance on large graphs. Although the message-passing GNN is also provably transferable, it lacks sufficient expressiveness for this task [14], leading to poor performance.

**Size Generalization on Point Clouds.** We follow the setup in Section 7.2 of [5], using the Model-Net10 dataset [87]. From Class 2 (chair) and Class 7 (sofa), we randomly selected 80 point clouds each, splitting them into 40 for training and 40 for testing. This results in $40 \times 40$ cross-class pairs. The objective is to learn the third lower bound of the Gromov-Wasserstein distance [60], which is invariant and continuous with respect to the Wasserstein $p$-metric (proven in Appendix I.4). Figure 5d shows that normalized DS-CI and normalized SVD-DS, both aligned with the target at the continuity level, achieve good test performance and size generalization. While normalized SVD-DS underperforms compared to normalized DS-CI, it offers superior time and memory efficiency.

## Acknowledgments and Disclosure of Funding

We thank Ben Blum-Smith, Ningyuan (Teresa) Huang, and Derek Lim for helpful discussions. EL is partially supported by AFOSR FA9550-23-1-0070 and FA9550-23-1-0204. YM is funded by NSF BSF 2430292. MD is partially supported by NSF CCF 2442615 and DMS 2502377. SV is partially supported by NSF CCF 2212457, the NSF–Simons Research Collaboration on the Mathematical and Scientific Foundations of Deep Learning (MoDL) (NSF DMS 2031985), NSF CAREER 2339682, NSF BSF 2430292, and ONR N00014-22-1-2126.

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

## A  Notation

We use $\mathbb{R}$ and $\mathbb{N}$ to denote the sets of real numbers and natural numbers. We use the pair $(\mathbb{N}, \preceq)$ to denote a directed poset on natural numbers, used for indexing. We write $[n]$ to denote the set $\{1, 2, \ldots, n\}$. The symbols $\mathbb{V}$ and $\mathbb{U}$ are used for consistent sequences, and $V_n$, $U_n$ for the corresponding vector spaces at finite levels. General groups are denoted by $\mathsf{G}_n$; we use $\mathsf{S}_n$ for the symmetric group and $\mathsf{O}(k)$ for the orthogonal group. We denote embeddings of vector spaces from dimension $n$ to $N$ by $\varphi_{N,n}$ and $\psi_{N,n}$. Embeddings of groups are denoted by $\theta_{N,n}$. We use "$\circ$" to denote the composition of functions, and $\mathrm{id}$ for the identity element in a group or the identity map, depending on the context. Binary operations in groups are denoted by "$*$", with subscripts such as "$*_n$" used when clarity is needed, e.g., to indicate the operation in the group $\mathsf{G}_n$. Group actions are denoted by "$\cdot$", with subscripts such as "$\cdot_n$" when needed.

We write $\sim$ to denote equivalence relations, and use $[x]$ to denote the equivalence class of $x$, i.e., $[x] = \{y : y \sim x\}$. We write $\vee$ and $\wedge$ for maximum and minimum respectively. In a metric space, we use $B(x_0, r)$ to denote the ball centered at $x_0$ with radius $r$, i.e. $B(x_0, r) := \{x : d(x_0, x) < r\}$. Given a set $S$, the symbol $\overline{S}$ denotes either completion or closure, depending on the context. Given a function $f$, the symbol $\overline{f}$ denotes a normalized variant of that function. Given two sequences $(a_n), (b_n) \subseteq \mathbb{R}$, we write $a_n \lesssim b_n$ if there exists a constant $C > 0$ such that $a_n \leq C b_n$ for all $n \in \mathbb{N}$. For a given function $R : \mathbb{N} \to \mathbb{R}_+$, we say a sequence $(x_n)$ converges to $x$ at rate $R(n)$ with respect to distance metric $d$ if $d(x_n, x) \lesssim R(n)$ and $R(n) \to 0$. For a matrix $X$, we use $X_{i:}$ to denote the vector formed by its $i$-th row. This work employs several norms: $\|\cdot\|_p$ refers to the standard $\ell_p$-norm (for vectors and infinite sequences) or $L^p$-norm (for functions), while $\|\cdot\|_{\overline{p}}$ denotes their normalized counterparts which will be defined later. The operator norm of a linear map $T : V \to W$ between two normed spaces is denoted $\|T\|_{\mathrm{op}} := \sup_{\|v\|=1} \|Tv\|$; its definition inherently depends on the specific norms chosen for $V$ and $W$. Particularly, $\|\cdot\|_{\mathrm{op},p,q}$ denotes the operator norm when the domain is equipped with the $\ell_p$-norm (or $L^p$-norm) and the codomain with the $\ell_q$-norm (or $L^q$-norm), we use $\|\cdot\|_{\mathrm{op},p}$ as an abbreviation for $\|\cdot\|_{\mathrm{op},p,p}$.

## B  Related work

**GNN transferability.** The work on GNN transferability under the graphon framework was pioneered by [70], focusing on a variant of graph convolutional network (GCN) for deterministic graphs obtained from the same graphon. In parallel, [48] explores transferability with respect to an alternative limit space to graphon in the form of a topological space, and [41] examines an arguably equivalent notion—convergence and stability—by analyzing random graphs sampled from a graphon. This line of research has since been further developed [71, 58], extending to more general message-passing networks (MPNNs) [18], more general notion of graph limits [46], and other models [73, 81, 11, 35]. Our framework unifies and recovers several of the above-mentioned results, which we briefly discuss in Appendix G.2. Furthermore, we develop new insights into the transferability of Invariant Graph Networks (IGNs) [56], a topic previously examined in [11, 35]. We leverage our framework to advance this line of inquiry and elaborate on the connections between our approach and prior work in Appendix G.3.

**GNN generalization.** A related line of research concerns the generalization theory of GNNs. Generalization bounds have been derived based on various frameworks, including Rademacher complexity [28, 55], VC dimension [75, 62, 24], the PAC-Bayesian approach [52, 39], the neural tangent kernel [22], and covering numbers [59, 57, 47, 79, 68]. Notions of size generalization without an explicit notion of convergence have been studied in [89, 4]. For a comprehensive overview, we refer the reader to the recent survey [80]. While most research on GNN generalization bounds consider

graphs of fixed or bounded size, [47, 68] explore the "uniform regime," addressing unbounded graph sizes. This perspective is closely aligned with the transferability theory, as they both consider continuous extensions over suitable limit spaces. Our work formalizes this connection, showing that transferability implies generalization (Section 4, Appendix E). We derive a generalization bound via covering numbers using the same proof strategy as prior works, but it applies well beyond GNNs and offers a setting that directly connects to the notion of size generalization.

**Equivariant machine learning.** Our theory naturally applies to equivariant machine learning models, i.e., neural networks with symmetries imposed. We particularly focus on models from [91, 56, 66, 72, 5]. There are many other equivariant machine learning models we haven't discussed here, such as the ones expressed in terms of group convolutions [17, 45, 33], representation theory [44, 78, 29], canonicalization [40], and invariant theory [83, 6, 84], among others.

**Representation stability and any-dimensional learning.** As noted by [49, 50], the presence of symmetry allows functions operating on inputs of arbitrary dimension to be parameterized with a finite number of parameters, which may be partially explained by the theory of representation stability [15]. Leveraging techniques from this theory, [50] provides the first general theoretical framework for any-dimensional equivariant models. Recent work applies similar techniques to study the generalization properties across dimensions of any-dimensional regression models [20]. Our work builds on the theoretical foundation established by this line of research.

**Any-dimensional expressivity and universality.** Our framework naturally prompts the question of expressivity for any-dimensional neural networks on their limit spaces. While we do not pursue this direction here, related questions have been independently studied. [9] shows that normalized DeepSets and PointNet are universal for uniformly continuous functions on suitable limit spaces, closely tied to our results (see Appendix F.2.1). In GNNs, the notion of "uniform expressivity"—expressing logical queries without parameter dependence on input size—has been explored in [34, 2, 42, 69]. Our framework offers complementary insights despite differing foundations.

## C  Consistent sequences and compatible, transferable maps: details and missing proofs from Section 2

### C.1  Consistent sequences and limit space

The concept of consistent sequences originated in the theory of representation stability [16]. We generalize this notion, originally considering a sequence of vector spaces, by allowing indexing over any directed poset on the natural numbers. Although we focus on natural numbers in this work, our theory generalizes to any directed poset.

**Directed poset indexing.** We require the indexing set $(\mathbb{N}, \preceq)$ to be a *directed poset* on natural numbers, meaning that $\mathbb{N}$ is the set of natural numbers equipped with a binary operation $\preceq$ satisfying:

*(Partial order)* The binary operation $\preceq$ is a partial order; that is, it satisfies: reflexivity ($a \preceq a$ for all $a \in \mathbb{N}$), transitivity (if $a \preceq b$ and $b \preceq c$, then $a \preceq c$), and antisymmetry (if $a \preceq b$ and $b \preceq a$, then $a = b$).

*(Upper bound condition)* For every pair $a, b \in \mathbb{N}$, there exists $c \in \mathbb{N}$ such that $a \preceq c$ and $b \preceq c$.

The directed poset indexing generalizes the notion of sequences to allow a more complex and flexible way of defining how smaller problem instances are embedded into larger ones, permitting "branching" directions of growth. We will see that the upper bound condition is crucial, as it ensures that any two problem instances are comparable—meaning they can both be embedded into a third, larger problem dimension and compared there. In this work, we only consider two cases: the natural numbers with the standard ordering $\leq$, and with the divisibility ordering, where $a \preceq b$ if and only if $a \mid b$.

**Definition C.1** (Consistent sequence: detailed version of Definition 2.1)**.** *A **consistent sequence** of group representations over directed poset $(\mathbb{N}, \preceq)$ is $\mathbb{V} = \{(V_n)_{n \in \mathbb{N}}, (\varphi_{N,n})_{n \preceq N}, (\mathsf{G}_n)_{n \in \mathbb{N}}\}$, where*

1. *(**Groups**) $(\mathsf{G}_n)$ is a sequence of groups indexed by $\mathbb{N}$ such that whenever $n \preceq N$, $\mathsf{G}_n$ is embedded into $\mathsf{G}_N$ via an injective group homomorphism $\theta_{N,n} \colon \mathsf{G}_n \to \mathsf{G}_N$, where*

$$\theta_{i,i} = \mathrm{id}_{\mathsf{G}_i} \quad \text{for all } i \in \mathbb{N},$$
$$\theta_{k,j} \circ \theta_{j,i} = \theta_{k,i} \quad \text{whenever } i \preceq j \preceq k \text{ in } \mathbb{N}.$$

2. (**Vector spaces**) $(V_n)$ is a sequence of (finite-dimensional, real) vector spaces indexed by $\mathbb{N}$, such that each $V_n$ is a $\mathsf{G}_n$-representation, and whenever $n \preceq N$, $V_n$ is embedded into $V_N$ through a linear embedding $\varphi_{N,n} \colon V_n \hookrightarrow V_N$, where

$$\varphi_{i,i} = \mathrm{id}_{V_i} \quad \text{for all } i \in \mathbb{N},$$

$$\varphi_{k,j} \circ \varphi_{j,i} = \varphi_{k,i} \quad \text{whenever } i \preceq j \preceq k \text{ in } \mathbb{N}.$$

3. (**Equivariance**) Every $\varphi_{N,n}$ is $\mathsf{G}_n$-equivariant, i.e.,

$$\varphi_{N,n}(g \cdot v) = \theta_{N,n}(g) \cdot \varphi_{N,n}(v) \text{ for all } g \in \mathsf{G}_n, v \in V_n.$$

Given a consistent sequence, we can first "summarize" the sequence of groups $(\mathsf{G}_n)$ into a single limit group $\mathsf{G}_\infty$, and likewise "summarize" the sequence of vector spaces $(V_n)$ into a single limit vector space $V_\infty$. We can then consider the action of $\mathsf{G}_\infty$ on $V_\infty$.

**Definition C.2** (Limit group: detailed version of Definition 2.3). *The **limit group** $\mathsf{G}_\infty$ is defined as the disjoint union $\bigsqcup_n \mathsf{G}_n$ modulo the equivalence relation that identifies each element $g \in \mathsf{G}_n$ with its images under the transition maps $\theta_{N,n}(g)$ for all $N \geq n$, i.e.,*

$$\mathsf{G}_\infty := \bigsqcup_n \mathsf{G}_n / \sim,$$

*where $g \sim \theta_{N,n}(g)$ whenever $n \preceq N$ and $g \in \mathsf{G}_n$. This construction is also known as the **direct limit** of groups, and is denoted by $\mathsf{G}_\infty = \varinjlim \mathsf{G}_n$. For each $g \in \mathsf{G}_n$, its equivalence class in $\mathsf{G}_\infty$ is denoted by $[g]$, representing the corresponding limiting object.*

The group structure on $\mathsf{G}_\infty$ is inherited from the groups $(\mathsf{G}_n)$ as follows. For $g_n \in \mathsf{G}_n$ and $g_m \in \mathsf{G}_m$, define the binary operation on equivalence classes by

$$[g_n] * [g_m] := [\theta_{N,n}(g_n) *_N \theta_{N,m}(g_m)],$$

where $N \in \mathbb{N}$ is a common upper bound of $n$ and $m$ in $(\mathbb{N}, \preceq)$, and $*_N$ denotes the group operation in $\mathsf{G}_N$. It is straightforward to check that this operation is well-defined.

**Definition C.3** (Limit space of consistent sequence: detailed version of Definition 2.3). *Define $V_\infty$ as the disjoint union $\bigsqcup V_n$ modulo an equivalence relation identifying each element $v \in V_n$ with its images under the transition map $\varphi_{N,n}(v)$ for all $n \preceq N$, i.e.,*

$$V_\infty := \bigsqcup_n V_n / \sim,$$

*where $v \sim \varphi_{N,n}(v)$ whenever $n \preceq N$. This construction is also known as the **direct limit** of vector spaces, and is denoted as $V_\infty = \varinjlim V_n$.*

The vector space structure on $V_\infty$ is inherited from the vector spaces $(V_n)$ as follows. For $v_n \in V_n, v_m \in V_m$, the addition and scalar multiplication on the equivalent classes are defined by

$$[v_n] + [v_m] := [\varphi_{N,n}(v_n) + \varphi_{N,m}(v_m)] \quad \text{where } N \text{ is an upper bound for } n, m \text{ in } (\mathbb{N}, \preceq),$$
$$\lambda[v_n] := [\lambda v_n].$$

It is straightforward to check that these operations are well-defined. The limit group $\mathsf{G}_\infty$ acts on $V_\infty$ by

$$[g] \cdot [v] := [\theta_{N,n}(g) \cdot_N \varphi_{N,m}(v)] \text{ for } g \in \mathsf{G}_n, v \in V_m,$$

where $N$ is an upper bound of $n, m$ in $(\mathbb{N}, \preceq)$, and $\cdot_N$ is the group action of $\mathsf{G}_N$ on $V_N$. It is also easy to check that this group action is well-defined, and $V_\infty$ is a $\mathsf{G}_\infty$-representation. The orbit space of $V_\infty$ under the action of $\mathsf{G}_\infty$ is

$$\{\mathsf{G}_\infty \cdot x : x \in V_\infty\},$$

where $\mathsf{G}_\infty \cdot x := \{g \cdot x : g \in \mathsf{G}_\infty\}$ is the orbit of point $x \in V_\infty$ under the $\mathsf{G}_\infty$-action. The orbits form a partition of $V_\infty$ into disjoint subsets.

**Consistent sequences without symmetries.** A special case of consistent sequences arises when $\mathsf{G}_n = \{\mathrm{id}\}$, the trivial group, for all $n \in \mathbb{N}$. In this case, the structure reduces to a directed system of vector spaces

$$\mathbb{V} = \{(V_n)_{n \in \mathbb{N}}, (\varphi_{N,n})_{n \preceq N}\}.$$

Hence, our theory of size generalization applies in scenarios where no intrinsic symmetries are present.

**Trivial consistent sequence.** Another special case arises when $V_n = V$, a fixed vector space, for all $n \in \mathbb{N}$, with embeddings given by $\varphi_{N,n} = \mathrm{id}_V$ for all $n \preceq N$. This yields the *trivial consistent sequence* associated with $V$, which we denote by $\mathbb{V}_V$. This construction is useful for modelling non-size-dependent spaces, such as the output space in graph-level classification or regression tasks.

**Direct sum and tensor product.** Given a consistent sequence $\mathbb{V} = \{(V_n), (\varphi_{N,n}), (\mathsf{G}_n)\}$, we can define its *direct sum* and *tensor product*. Both of them are also consistent sequences.

**Definition C.4.** *The $d$-th **direct sum** of $\mathbb{V}$ is defined as*

$$\mathbb{V}^{\oplus d} := \{(V_n^{\oplus d}), (\varphi_{N,n}^{\oplus d}), (\mathsf{G}_n)\},$$

*where $V_n^{\oplus d}$ denotes the direct sum of $d$ copies of $V_n$ and $\varphi_{N,n}^{\oplus d} \colon V_n^{\oplus d} \to V_N^{\oplus d}$ is defined by applying $\varphi_{N,n}$ to each component. The group $\mathsf{G}_n$ acts on $V_n^{\oplus d}$ by simultaneously acting on every copy of $V_n$, i.e. $g \cdot (v_1, \dots, v_d) := (g \cdot v_1, \dots, g \cdot v_d)$.*

**Definition C.5.** *The $d$-th **tensor product** of $\mathbb{V}$ is defined as*

$$\mathbb{V}^{\otimes d} := \{(V_n^{\otimes d}), (\varphi_{N,n}^{\otimes d}), (\mathsf{G}_n)\},$$

*where $V_n^{\otimes d}$ denotes the $d$-fold tensor product of $V_n$. $\varphi_{N,n}^{\otimes d} \colon V_n^{\otimes d} \to V_N^{\otimes d}$ is uniquely defined by $\varphi_{N,n}^{\otimes d}(v_1 \otimes \cdots \otimes v_d) := \varphi_{N,n}(v_1) \otimes \dots \varphi_{N,n}(v_d)$, and the group action of $\mathsf{G}_n$ on $V_n^{\otimes d}$ is uniquely defined by $g \cdot (v_1 \otimes \cdots \otimes v_d) := (g \cdot v_1) \otimes \cdots \otimes (g \cdot v_d)$. The universal property of tensor product guarantees that $\varphi_{N,n}^{\otimes d}$ and the group action mentioned are well-defined. Similarly, we also consider the $d$-**th symmetric tensors** of $\mathbb{V}$ as*

$$\mathrm{Sym}^d(\mathbb{V}) := \{(\mathrm{Sym}^d(V_n)), (\varphi_{N,n}^{\otimes d}), (\mathsf{G}_n)\},$$

*where $\mathrm{Sym}^d(V_n)$ denotes the space of symmetric tensors of order $d$ defined on $V_n$, i.e. the subspace of $V_n^{\otimes d}$ invariant under the action of the symmetric group $\mathsf{S}_d$.*

The direct sum, $\mathbb{V}^{\oplus d}$, is particularly useful for incorporating hidden channels into our analysis, as it effectively adds the extra channel dimensions to our data. In contrast, the tensor product, $\mathbb{V}^{\otimes d}$, is helpful to extend a consistent sequence on vectors or sets to higher-order objects such as matrices or graphs. For example, the duplication consistent sequences for graphs exactly arise as the 2nd symmetric tensors of the duplication consistent sequences for sets.

## C.2 Compatible maps

Recall from Definition 2.4 that a sequence of maps $(f_n \colon V_n \to U_n)$ is *compatible* with respect to the consistent sequences $\mathbb{V}, \mathbb{U}$ if, for all $n \preceq N$,

$$f_N \circ \varphi_{N,n} = \psi_{N,n} \circ f_n,$$

and each $f_n$ is $\mathsf{G}_n$-equivariant. This condition is equivalent to the existence of an extension to the limit map $f_\infty$.

**Proposition C.6** (Compatible maps and extension to limit). *Let $\mathbb{V} = \{(V_n), (\varphi_{N,n}), (\mathsf{G}_n)\}$ and $\mathbb{U} = \{(U_n), (\psi_{N,n}), (\mathsf{G}_n)\}$ be two consistent sequences. A sequence of maps $(f_n \colon V_n \to U_n)$ is compatible if and only if it extends to the limit; that is, there exists a $\mathsf{G}_\infty$-equivariant map*

$$f_\infty \colon V_\infty \to U_\infty$$

*such that $f_n = f_\infty \mid_{V_n}$ for all $n$.*

*Proof.* ($\Leftarrow$) Suppose there exists a $\mathsf{G}_\infty$-equivariant map $f_\infty$ such that $f_n = f_\infty|_{V_n}$ for all $n$. Then for all $n \preceq N$ and $x \in V_n$,

$$[f_n(x)] = f_\infty([x]) = f_\infty([\varphi_{N,n}(x)]) = [f_N(\varphi_{N,n}(x))],$$

which implies $f_N \circ \varphi_{N,n} = \psi_{N,n} \circ f_n$. Moreover, for all $n \in \mathbb{N}$, $x \in V_n$, and $g \in \mathsf{G}_n$,

$$[f_n(g \cdot x)] = f_\infty([g \cdot x]) = f_\infty([g] \cdot [x]) = [g] \cdot f_\infty([x]) = [g] \cdot [f_n(x)] = [g \cdot f_n(x)],$$

so each $f_n$ is $\mathsf{G}_n$-equivariant.

($\Rightarrow$) Conversely, suppose $(f_n)$ are compatible. Define

$$f_\infty \colon V_\infty \to W_\infty, \quad f_\infty([x]) := [f_n(x)] \quad \text{if } x \in V_n.$$

Compatibility ensures this is well-defined. To verify equivariance, let $g \in \mathsf{G}_m$ and $x \in V_n$, and let $N$ be a common upper bound of $n$ and $m$ in $(\mathbb{N}, \preceq)$. Then,

$$
\begin{aligned}
f_\infty([g] \cdot [x]) &= f_\infty([\theta_{N,m}(g) \cdot \varphi_{N,n}(x)]) \\
&= [f_N(\theta_{N,m}(g) \cdot \varphi_{N,n}(x))] \\
&= [\theta_{N,m}(g) \cdot f_N(\varphi_{N,n}(x))] \\
&= [\theta_{N,m}(g) \cdot \psi_{N,n}(f_n(x))] \\
&= [g] \cdot f_\infty([x]).
\end{aligned}
$$

Therefore, $f_\infty$ is $\mathsf{G}_\infty$-equivariant. $\qquad\square$

This proposition implies that learning a function on the infinite-dimensional space $V_\infty$, a task that may appear difficult, reduces to learning a compatible sequence of functions on the finite-dimensional vector spaces along the sequence, which is a more tractable problem.

## C.3 Metrics on consistent sequences

In Section 2.1, we introduced a norm on $V_\infty$ so as to define distance between objects of different dimensions. In this appendix, we take a more general perspective and examine metric structures on consistent sequences, and present detailed proofs. The same proofs carry over to the norm setting with minimal modification.

**Definition C.7** (Compatible metrics: generalized version of Definition 2.5). *Let* $\mathbb{V} = \{(V_n), (\varphi_{N,n}), (\mathsf{G}_n)\}$ *be a consistent sequence. A sequence of metrics* $(d_n)$ *on the vector spaces* $V_n$ *is said to be **compatible** if all the embeddings* $\varphi_{N,n}$ *and the* $\mathsf{G}_n$-*actions are isometries. That is, for all* $n \preceq N$, $x, y \in V_n$, *and* $g \in \mathsf{G}_n$, *we have:*

$$d_N(\varphi_{N,n}(x), \varphi_{N,n}(y)) = d_n(x, y), \quad \text{and} \quad d_n(g \cdot x, g \cdot y) = d_n(x, y).$$

Similar to compatible maps, this is equivalent to the existence of an extension to a metric $d_\infty$ on $V_\infty$.

**Proposition C.8** (Compatible metrics and extension to the limit). *A sequence of metrics* $(d_n)$ *on the spaces* $V_n$ *is compatible if and only if it extends to a metric on the limit space. That is, there exists a metric* $d_\infty$ *on* $V_\infty$ *such that*

$$d_n(x, y) = d_\infty([x], [y]) \quad \text{for all } n \in \mathbb{N}, \ x, y \in V_n,$$

*and the* $\mathsf{G}_\infty$-*action on* $V_\infty$ *is an isometry with respect to* $d_\infty$, *i.e.,*

$$d_\infty(g \cdot x, g \cdot y) = d_\infty(x, y) \quad \text{for all } x, y \in V_\infty, \ g \in \mathsf{G}_\infty.$$

*Proof.* The proof is primarily a matter of bookkeeping, similar in spirit to Proposition C.6. For completeness, we present the full argument below.

($\Leftarrow$) Suppose $(d_n)$ extends to a metric $d_\infty$ on $V_\infty$. Then for all $n \preceq N$ and $x, y \in V_n$, we have

$$d_n(x, y) = d_\infty([x], [y]) = d_\infty([\varphi_{N,n}(x)], [\varphi_{N,n}(y)]) = d_N(\varphi_{N,n}(x), \varphi_{N,n}(y)).$$

Thus, the embeddings $\varphi_{N,n}$ are isometries. Moreover, for any $g \in \mathsf{G}_n$ and $x, y \in V_n$,

$$d_n(g \cdot x, g \cdot y) = d_\infty([g \cdot x], [g \cdot y]) = d_\infty([g] \cdot [x], [g] \cdot [y]) = d_\infty([x], [y]) = d_n(x, y),$$

so the group actions are isometries as well.

($\Rightarrow$) Conversely, suppose the collection $(d_n)$ is compatible. Define a metric $d_\infty \colon V_\infty \times V_\infty \to \mathbb{R}$ as follows: for $x \in V_n$ and $y \in V_m$, let $N$ be any common upper bound of $n$ and $m$ in $(\mathbb{N}, \preceq)$, and set

$$d_\infty([x], [y]) := d_N(\varphi_{N,n}(x), \varphi_{N,m}(y)).$$

Compatibility of the metrics ensures that $d_\infty$ is well-defined. It is also easy to check that $d_\infty$ is a metric. Moreover, by construction, $d_n = d_\infty|_{V_n}$ for all $n$.

To verify that the $\mathsf{G}_\infty$-action is isometric, take $x \in V_{n_1}$, $y \in V_{n_2}$, and $g \in \mathsf{G}_n$ for some $n \in \mathbb{N}$. Let $N$ be a common upper bound of $n, n_1, n_2$ in $(\mathbb{N}, \preceq)$. Then,

$$
\begin{aligned}
d_\infty([g] \cdot [x], [g] \cdot [y]) &= d_N(\theta_{N,n}(g) \cdot \varphi_{N,n_1}(x), \theta_{N,n}(g) \cdot \varphi_{N,n_2}(y)) \\
&= d_N(\varphi_{N,n_1}(x), \varphi_{N,n_2}(y)) \\
&= d_\infty([x], [y]).
\end{aligned}
$$

This completes the proof. $\qquad\square$

With the metric structure in place, we define the limit space via the completion of the metric space. The *completion* of a metric space $M$ is a complete metric space $\overline{M}$—that is, a space in which every Cauchy sequence converges—that contains $M$ as a dense subset (i.e., the smallest closed subset of $\overline{M}$ containing $M$ is $\overline{M}$ itself).

**Definition C.9** (Limit space: detailed version of Definition 2.7)**.** *Let $\mathbb{V}$ be a consistent sequence, and let $V_\infty$ be equipped with the metric $d_\infty$. Denote by $\overline{V_\infty}$ the completion of $V_\infty$ with respect to $d_\infty$. The $\mathsf{G}_\infty$-action on $V_\infty$ extends to a well-defined action on $\overline{V_\infty}$ as follows: for any $x \in \overline{V_\infty}$ and $g \in \mathsf{G}_\infty$, choose a sequence $(x_n)$ in $V_\infty$ such that $x_n \to x$ in $\overline{V_\infty}$, and define*

$$
g \cdot x := \lim_{n \to \infty} g \cdot x_n.
$$

*This limit exists because $(g \cdot x_n)$ is a Cauchy sequence, as the $\mathsf{G}_\infty$-action on $V_\infty$ is isometric. The resulting action on $\overline{V_\infty}$ is linear and isometric. We define the **limit space** of the consistent sequence $\mathbb{V}$ to be the $\mathsf{G}_\infty$-representation $\overline{V_\infty}$.*

The set of orbit closures in $\overline{V_\infty}$ under the action of $\mathsf{G}_\infty$ is

$$
\left\{ \overline{\mathsf{G}_\infty \cdot x} : x \in \overline{V_\infty} \right\}.
$$

where $\overline{\mathsf{G}_\infty \cdot x}$ is the closure of the orbit $\mathsf{G}_\infty \cdot x$. Intuitively, $\overline{V_\infty}$ includes not only elements from finite-dimensional objects (elements in $V_\infty$), but also additional points that are "reachable" as limits of finite-dimensional objects. We can further define a symmetrized metric on the limit space.

**Proposition C.10** (Symmetrized metric)**.** *Let $x, y \in \overline{V_\infty}$. Define*

$$
\overline{\mathrm{d}}(x, y) := \inf_{g \in \mathsf{G}_\infty} d_\infty(g \cdot x, y).
$$

*Then $\overline{\mathrm{d}}$ is a pseudometric on $\overline{V_\infty}$ and induces a metric on the space of orbit closures in $\overline{V_\infty}$ under the $\mathsf{G}_\infty$-action. We refer to $\overline{\mathrm{d}}$ as the* symmetrized metric.

*Proof.* The non-negativity of $\overline{\mathrm{d}}$ follows directly from the non-negativity of $d_\infty$.

(*Symmetry*) Since $d_\infty(g \cdot x, y) = d_\infty(x, g^{-1} \cdot y)$ for any $g \in \mathsf{G}_\infty$ by isometry of the $\mathsf{G}_\infty$-action, taking the infimum over all $g \in \mathsf{G}_\infty$ (equivalently over $g^{-1}$) yields

$$
\overline{\mathrm{d}}(x, y) = \overline{\mathrm{d}}(y, x).
$$

(*Triangle inequality*) Let $\varepsilon > 0$. Then there exist $g, h \in \mathsf{G}_\infty$ such that

$$
\overline{\mathrm{d}}(x, y) > d_\infty(g \cdot x, y) - \varepsilon, \quad \overline{\mathrm{d}}(y, z) > d_\infty(h \cdot y, z) - \varepsilon.
$$

Using the isometry of the group action:

$$
\begin{aligned}
\overline{\mathrm{d}}(x, y) + \overline{\mathrm{d}}(y, z) &> d_\infty(g \cdot x, y) + d_\infty(h \cdot y, z) - 2\varepsilon \\
&= d_\infty(hg \cdot x, h \cdot y) + d_\infty(h \cdot y, z) - 2\varepsilon \\
&\geq d_\infty(hg \cdot x, z) - 2\varepsilon \\
&\geq \overline{\mathrm{d}}(x, z) - 2\varepsilon.
\end{aligned}
$$

Since this holds for arbitrary $\varepsilon > 0$, the triangle inequality follows.

(*Definiteness*) We have $\overline{\mathrm{d}}(x, y) = 0$ if and only if there exists a sequence $(g_n) \subseteq \mathsf{G}_\infty$ such that $d_\infty(g_n \cdot x, y) \to 0$. This is precisely the condition that $y \in \overline{\mathsf{G}_\infty \cdot x}$, i.e., $x$ and $y$ lie in the same orbit closure.

Therefore, $\overline{\mathrm{d}}$ is a pseudometric on $\overline{V_\infty}$, and descends to a true metric on the space of orbit closures which completes the proof. $\qquad\square$

# D    Transferability: details and missing proofs from Section 3

## D.1    Transferable maps

Following Definition 3.1 of continuously, $L$-Lipschitz, and $L(r)$-locally Lipschitz transferable, we further define the following notion: If $(f_n)$ is a sequence of equivariant maps extending to $f_\infty$ which is locally Lipschitz at $x_0$[7], we say $(f_n)$ is *locally Lipschitz transferable at $x_0$*. Being locally Lipschitz transferable at $x_0$ is a weaker condition than being $L(r)$-locally Lipschitz transferable, which is itself weaker than $L$-Lipschitz transferable. This definition is useful when studying models which are discontinuous on negligible sets of inputs, and which are therefore not $L(r)$-locally Lipschitz transferable. These often come up when constructing architectures based on canonicalizations [23], as commonly done for point clouds for instance (see Section 5.3).

We first show a useful property that continuity/Lipschitz with respect to $\|\cdot\|_{V_\infty}, \|\cdot\|_{U_\infty}$ implies the same property with respect to the symmetrized metrics, even though the converse does not hold. We will again state and prove the results for metrics instead.

**Proposition D.1** (Continuity in $d_\infty$ implies continuity in $\overline{d}$)**.** *Let $\mathbb{V}, \mathbb{U}$ be consistent sequences. If $f_\infty : \overline{V_\infty} \to \overline{U_\infty}$ is continuous (respectively, $L(r)$-Lipschitz on $B(0,r)$ for all $r > 0$, $L$-Lipschitz, locally Lipschitz at $x_0 \in \overline{V_\infty}$) with respect to $d_\infty^V$ and $d_\infty^U$, then $f_\infty$ satisfies the same property with respect to the symmetrized metrics $\overline{d}_V$ and $\overline{d}_U$.*

*Proof.* (*Continuity*) Let $x_n \to x$ in $\overline{V_\infty}$ with respect to the symmetrized metric $\overline{d}_V$, and $\varepsilon > 0$. By continuity of $f_\infty$ with respect to $d_\infty$, there exists $\delta > 0$ such that whenever $d_\infty(x,y) < \delta$, $d_\infty(f_\infty(x), f_\infty(y)) < \varepsilon$. Take $N$ such that for all $n \geq N$, $\overline{d}_V(x_n, x) < \frac{\delta}{2}$. Moreover, for each $n$, choose $g_n \in \mathsf{G}_\infty$ such that $d_\infty(g_n \cdot x_n, x) \leq \overline{d}_V(x_n, x) + \frac{\delta}{2}$. Then for all $n \geq N$, $d_\infty(g_n \cdot x_n, x) < \delta$ and hence

$$\overline{d}_U(f_\infty(x_n), f_\infty(x)) \leq d_\infty(g_n \cdot f_\infty(x_n), f_\infty(x)) = d_\infty(f_\infty(g_n \cdot x_n), f_\infty(x)) < \varepsilon.$$

Therefore $f_\infty(x_n) \to f_\infty(x)$ with respect to $\overline{d}_U$.

(*$L(r)$-Lipschitz on $B(0,r)$*) Suppose $f_\infty$ is $L(r)$-Lipschitz on $B(0,r)$ for all $r > 0$ with respect to $d_\infty$. Consider $x, y$ such that $\overline{d}_V(0, x) = d_\infty^V(0, x) < r$, and $\overline{d}_V(0, y) = d_\infty^V(0, y) < r$. Then for any $g \in \mathsf{G}_\infty$, we have $d_\infty^V(0, g \cdot x) = d_\infty^V(0, x) < r$. Hence,

$$\overline{d}_U(f_\infty(x), f_\infty(y)) \leq d_\infty(g \cdot f_\infty(x), f_\infty(y)) = d_\infty(f_\infty(g \cdot x), f_\infty(y)) \leq L(r) d_\infty(g \cdot x, y).$$

Take infimum over $g \in \mathsf{G}_\infty$, get $\overline{d}_U(f_\infty(x), f_\infty(y)) \leq L(r) \overline{d}_V(x, y)$.

(*Lipschitz*) For any $x, y \in V_\infty, g \in \mathsf{G}_\infty$,

$$\overline{d}_U(f_\infty(x), f_\infty(y)) \leq d_\infty(g \cdot f_\infty(x), f_\infty(y)) = d_\infty(f_\infty(g \cdot x), f_\infty(y)) \leq L d_\infty(g \cdot x, y).$$

Take infimum over $g \in \mathsf{G}_\infty$, get $\overline{d}_U(f_\infty(x), f_\infty(y)) \leq L \overline{d}_V(x, y)$.

(*Locally Lipschitz at $x_0$*) Suppose $d_\infty(f(x), f(x_0)) \leq L d_\infty(x, x_0)$ whenever $d_\infty(x, x_0) < r$. Then for any such $x$ we have $\overline{d}_V(x, x_0) = \inf_{g \in \mathsf{G}_\infty^{(r)}} d_\infty(x, x_0)$ where $\mathsf{G}_\infty^{(r)} = \{g \in \mathsf{G}_\infty : d_\infty(x, x_0) \leq r\}$. Moreover, for any $g \in \mathsf{G}_\infty^{(r)}$ we have $\overline{d}_U(f(x), f(x_0)) \leq d_\infty(f(g \cdot x), f(x_0)) \leq L d_\infty(g \cdot x, x)$, so after taking infimum over such $g$ we conclude that $\overline{d}_U(f(x), f(x_0)) \leq L \overline{d}_V(x, x_0)$ whenever $\overline{d}_V(x, x_0) < r$, as desired. This completes the proof of the proposition. $\square$

Finally, we state and prove a set of more concrete characterizations of Lipschitz transferable sequence of functions defined in Definition 3.1, which are straightforward to check.

**Proposition D.2.** *Let $\mathbb{V}, \mathbb{U}$ be consistent sequences endowed with metrics.*

---

[7]We say $f_\infty$ is locally Lipschitz *at $x_0$* if there exists $r > 0$ and $L > 0$ such that for all $x \in B(x_0, r)$, $d_\infty(f_\infty(x), f_\infty(x_0)) \leq L d_\infty(x, x_0)$. Notice that this is slightly different from saying $f_\infty$ is locally Lipschitz *around $x_0$*, which means that there exists $r > 0$ and $L > 0$ such that $f_\infty$ is $L$-Lipschitz on $B(x_0, r)$.

1. **(General case)** *A compatible sequence of functions $(f_n : V_n \to U_n)$ is L-Lipschitz (respectively, $L(r)$-locally Lipschitz) transferable if and only if for all $n$, $f_n$ is L-Lipschitz (respectively, $L(r)$-Lipschitz on $B_n(0, r) := \{v \in V_n : d_n^V(0, v) < r\}$).*

2. **(Linear maps)** *When the metrics are induced by norms, a compatible sequence of linear maps $(W_n : V_n \to U_n)$ is continuously (respectively, L-Lipschitz) transferable if and only if $\sup_n \|W_n\|_{\mathrm{op}} < \infty$ (respectively, $\sup_n \|W_n\|_{\mathrm{op}} \le L$).*

*Proof.* We begin by deriving the result for the general case.

**General case.** The "$\Rightarrow$" direction again follows immediately from $d_n = d_\infty|_{V_n}$, $f_n = f_\infty|_{V_n}$ for all $n$. We focus on proving "$\Leftarrow$". First, by Proposition C.6, compatibility implies that the sequence $(f_n)$ extends to a function $f_\infty : V_\infty \to U_\infty$.

(*Lipschitz*) Suppose $f_n$ is L-Lipschitz for all $n$. For any $x \in V_n$ and $y \in V_m$, let $N$ be a common upper bound of $n$ and $m$ in $(\mathbb{N}, \preceq)$. Then:

$$
\begin{aligned}
d_\infty^U(f_\infty([x]), f_\infty([y])) &= d_N^U\left(f_N(\varphi_{N,n}(x)), f_N(\varphi_{N,m}(y))\right) \\
&\le L\, d_N^V\left(\varphi_{N,n}(x), \varphi_{N,m}(y)\right) \\
&= L\, d_\infty^V([x], [y]).
\end{aligned}
$$

Hence, $f_\infty$ is L-Lipschitz on $V_\infty$.

Since Lipschitz continuity implies Cauchy continuity, $f_\infty$ extends uniquely to $f_\infty : \overline{V_\infty} \to \overline{U_\infty}$.

For any Cauchy sequences $(x_n)$ and $(y_n)$ in $V_\infty$ with limits $x, y \in \overline{V_\infty}$, we have:

$$
\begin{aligned}
d_\infty^U(f_\infty(x), f_\infty(y)) &= \lim_{n\to\infty} d_\infty^U(f_\infty(x_n), f_\infty(y_n)) \\
&\le \lim_{n\to\infty} L\, d_\infty^V(x_n, y_n) = L\, d_\infty^V(x, y),
\end{aligned}
$$

which shows that $f_\infty$ remains L-Lipschitz after extending to $\overline{V_\infty}$.

(*$L(r)$-locally Lipschitz*) Suppose each $f_n$ is $L(r)$-Lipschitz on $B_n(0, r)$ for all $r > 0$. As above, for any $x \in V_n$ and $y \in V_m$ with $d_n(0, x), d_m(0, y) < r$, let $N$ be a common upper bound of $n, m$ in $(\mathbb{N}, \preceq)$. Then $\varphi_{N,n}(x), \varphi_{N,m}(y) \in B_N(0, r)$, and by the $L(r)$-Lipschitz property of $f_N$ on $B_N(0, r)$, we get

$$
d_\infty^U(f_\infty([x]), f_\infty([y])) \le L(r) \cdot d_\infty^V([x], [y]).
$$

Thus, $f_\infty : V_\infty \to U_\infty$ is $L(r)$-Lipschitz on $\{v \in V_\infty : d_\infty^V(0, v) < r\}$.

This implies $f_\infty$ is Cauchy continuous: any Cauchy sequence $(x_n)$ lies within some ball of radius $R$, and since $f_\infty$ is Lipschitz continuous there, $(f_\infty(x_n))$ is also Cauchy. Hence, $f_\infty$ extends uniquely to $\overline{V_\infty}$, and it is easy to check that it is $L(r)$-Lipschitz on $B(0, r)$ for all $r$.

**Linear maps.** We leverage the argument for the general case to derive the two stated claims.

(*Lipschitz*) By our argument for normed spaces, $(W_n)$ is L-Lipschitz transferable if and only if for all $n$, $W_n$ is L-Lipschitz. By linearity of each $W_n$, this is equivalent to $\|W_n\|_{\mathrm{op}} \le L$ for all $n$.

(*Continuity*) It is sufficient to prove that $(W_n)$ is continuously transferable if and only if it is L-Lipschitz transferable for some $L > 0$. The "$\Leftarrow$" direction is immediate. To prove "$\Rightarrow$", suppose $(W_n)$ extends to a continuous function $W_\infty : \overline{V_\infty} \to \overline{U_\infty}$. Then $W_\infty$ is linear on $V_\infty$ because for any $x \in V_n, y \in V_m$, and any common upper bound $N$ of $n, m$ in $(\mathbb{N}, \preceq)$, we have

$$
W_\infty(a[x] + b[y]) = [W_N(a\varphi_{N,n}(x) + b\varphi_{N,m}(y))] = aW_\infty([x]) + bW_\infty([y]).
$$

By continuity of $W_\infty$, it remains linear on $\overline{V_\infty}$. The result follows since for linear operators, Lipschitz continuity and continuity are equivalent.

Thus, the proof is finished.

$\square$

## D.2 Convergence, transferability and stability

**Stability.** The following stability result states that small perturbations of the input (e.g., adding a small number of nodes to a graph) lead to small changes in the output. It resembles the stability considered in [70].

**Proposition D.3** (Stability: detailed version of Proposition 3.2)**.** *If the sequence of maps* $(f_n \colon V_n \to U_n)$ *is* $L(r)$*-locally Lipschitz transferable, then for any two inputs* $x_n \in V_n$ *and* $x_m \in V_m$ *of any two sizes* $n, m$ *with* $d_n^V(0, x_n), d_m^V(0, x_m) \leq r$, *we have*

$$d_\infty^U([f_n(x_n)], [f_m(x_m)]) \leq L d_\infty^V([x_n], [x_m]).$$

*Moreover, the same holds when replacing every* $d_\infty$ *with the symmetrized metric* $\overline{\mathrm{d}}$.

**Convergence and transferability: deterministic sampling.** For a sequence of inputs $(x_n)$ sampled (deterministically) from the same underlying limiting object $x$, the outputs of a transferable function satisfy $f_n(x_n) \approx f_m(x_m)$ for big $n, m$, and converge as $f_n(x_n) \to f_\infty(x)$. We provide examples of sampling procedures later in Appendix D.3.

**Proposition D.4** (Convergence and transferability: detailed version of Proposition 3.2)**.** *Let* $(x_n \in V_n)_{n \in \mathbb{N}}$ *be a sequence of inputs sampled from a limiting object* $x \in \overline{V_\infty}$, *such that* $[x_n] \to x$ *at a rate* $R(n)$ *with respect to* $d_\infty^V$.

1. **(Asymptotic)** *If* $(f_n \colon V_n \to U_n)_{n \in \mathbb{N}}$ *is continuously transferable, then the following holds.*

   *(Convergence) The sequence* $[f_n(x_n)] \to f_\infty(x)$ *with respect to* $d_\infty^U$.
   *(Transferability) The distance* $d_\infty^U([f_n(x_n)], [f_m(x_m)]) \to 0$ *as* $n, m \to \infty$.

2. **(Nonasymptotic)** *If* $(f_n \colon V_n \to U_n)_{n \in \mathbb{N}}$ *is locally Lipschitz transferable at* $x$, *then the following holds.*

   *(Convergence) The sequence* $[f_n(x_n)] \to f_\infty(x)$ *at a rate* $R(n)$ *with respect to* $d_\infty^U$.
   *(Transferability) The distance is bounded by* $d_\infty^U([f_n(x_n)], [f_m(x_m)]) \lesssim R(n) + R(m)$.

That is, Lipschitzness provides quantitative guarantees for the convergence rate. We remark that by By Proposition D.1, the same holds when replacing every $d_\infty$ with the symmetrized metric $\overline{\mathrm{d}}$.

*Proof.* We start by noticing that both transferability results directly follows from convergence, thanks to the triangle inequality

$$d_\infty([f_n(x_n)], [f_m(x_m)]) \leq d_\infty([f_n(x_n)], f_\infty(x)) + d_\infty([f_m(x_m)], f_\infty(x)).$$

To show convergence in under continuous transferability, observe that if $f_\infty$ is continuous, then $[f_n(x_n)] = f_\infty([x_n]) \to f_\infty(x)$ immediately follows.

Next, we establish the guarantee under local Lipschitz transferability at $x$. Suppose $f_\infty$ is locally Lipschitz at $x$. Then there exists $r > 0$ such that for all $y \in B(x, r)$, we have $d_\infty(f_\infty(x), f_\infty(y)) \leq L d_\infty(x, y)$. Let $N$ be large enough so that $x_n \in B(x, r)$ for all $n \geq N$. Then for all $n \geq N$, we have

$$d_\infty(f_\infty(x), [f_n(x_n)]) = d_\infty(f_\infty(x), f_\infty([x_n])) \leq L \cdot d_\infty(x, [x_n]) \lesssim R(n),$$

as claimed; finishing the proof of the proposition. $\square$

**Convergence and transferability: random sampling.** Under random sampling of inputs, we need to specify the mode of convergence. In the case where $x_n \to x$ almost surely at rate $R(n)$, the results are identical to the deterministic case: both convergence and transferability hold almost surely. We now consider a different mode of convergence—convergence in expectation. As we will see in Appendix D.3, many common sampling procedures satisfy this condition.

**Proposition D.5** (Convergence and transferability: Random sampling)**.** *Let* $(x_n \in V_n)$ *be a sequence of inputs randomly sampled from a limiting object* $x \in \overline{V_\infty}$, *such that* $[x_n] \to x$ *in expectation at rate* $R(n)$ *with respect to* $d_\infty^V$, *i.e.* $\mathbb{E}[d_\infty^V([x_n], x)] \lesssim R(n)$ *and* $R(n) \to 0$. *Suppose* $(f_n \colon V_n \to U_n)$ *is locally Lipschitz transferable at* $x$, *i.e.,* $f_\infty$ *is* $L$*-Lipschitz on* $B(x, r)$. *Further, assume that there exists* $M > 0$ *such that*

$$\mathbb{E}\left[d_\infty(f_\infty([x_n]), f_\infty(x)) \mathbb{1}([x_n] \notin B(x, r))\right] \leq M \mathbb{E}\left[d_\infty([x_n], x)\right]. \tag{2}$$

*(Convergence)* The function values converge in expectation
$$\mathbb{E}[d_\infty([f_n(x_n)], f_\infty(x))] \lesssim R(n).$$

*(Transferability)* The distance converges in expectation
$$\mathbb{E}[d_\infty([f_n(x_n)], [f_m(x_m)])] \lesssim R(n) + R(m).$$

The assumptions in this proposition are rather mild. Indeed, (2) amounts to a localized version of uniform integrability. Simple arguments show that these assumptions are satisfied under any of the following scenarios: $(i)$ the sequence $(f_n)$ is globally Lipschitz transferable, $(ii)$ the map $f_\infty$ is bounded or $(iii)$ the sequence $(x_n)$ is supported on $B(x, r)$. Furthermore, the same conclusion remains valid when replacing $d_\infty$ with the symmetrized metric $\overline{\mathrm{d}}$.

*Proof.* Suppose for all $y \in B(x, r)$, we have $d_\infty(f_\infty(x), f_\infty(y)) \leq L d_\infty(x, y)$. Then,

$$
\begin{aligned}
\mathbb{E}[d_\infty(f_\infty(x), [f_n(x_n)])] &\leq \mathbb{E}\big[d_\infty(f_\infty(x), f_\infty([x_n])) \cdot \mathbb{1}\{[x_n] \in B(x, r)\}\big] + M\mathbb{E}\,[d_\infty(x, [x_n])] \\
&\leq L \cdot \mathbb{E}\big[d_\infty(x, [x_n]) \cdot \mathbb{1}\{[x_n] \in B(x, r)\}\big] + M\mathbb{E}\,[d_\infty(x, [x_n])] \\
&\lesssim R(n).
\end{aligned}
$$

Transferability then follows by the triangle inequality. $\square$

### D.3 Convergence rates under sampling

Propositions D.4 and D.5 show that for Lipschitz transferable models $(f_n)$, the convergence of $f_n(x_n)$ to $f_\infty(x)$ is at least as fast as the convergence of $x_n$ to $x$, characterized by the rate $R(n)$. This rate depends on the specific application and sampling scheme. Below, we review common sampling schemes and their associated convergence rates from the literature.

#### D.3.1 Random sampling

**Empirical distributions and signals.** Suppose $p \in [1, \infty]$ and $\mu \in \mathcal{P}_q(\mathbb{R}^d)$ for some $q > 2p$. Suppose $X \in \mathbb{R}^{n \times d}$ has rows sampled i.i.d. from $\mu$, and let $\mu_X = \frac{1}{n} \sum_{i=1}^n \delta_{X_{i:}}$ be the corresponding (empirical) distribution on $\mathbb{R}^d$. Then, we have [26]

$$
\mathbb{E}[W_p(\mu, \mu_X)] \lesssim \begin{cases} n^{-1/2p} & \text{if } p > d/2, \\ n^{-1/2p} \log^{1/p}(1+n) & \text{if } p = d/2, \\ n^{-1/d} & \text{if } p < d/2. \end{cases} \tag{3}
$$

Similarly, suppose $f \in L^\infty([0, 1])$ is a bounded signal sampled by $x_i = f(t_i)$ where $t_1, \dots, t_n$ are i.i.d. uniform $[0, 1]$, and if $f_n$ be the step function corresponding to $x$. Noting that $\mu_f$ has moments of all orders and that $\mu_{f_n}$ is the empirical measure obtained by sampling $n$ iid points from $\mu_f$, we conclude that $\mathbb{E}\,\overline{\mathrm{d}}_p(f, f_n) = \mathbb{E}W_p(\mu_f, \mu_{f_n})$ converges at the rates (3) with $d = 1$, where $\overline{\mathrm{d}}_p$ is the symmetrized metric with respect to the $L^p$ norm on functions.

**Point clouds.** Again let $p \in [1, \infty]$ and $\mu \in \mathcal{P}_q(\mathbb{R}^k)$ with $q > 2p$, and suppose $X \in \mathbb{R}^{n \times k}$ has rows sampled i.i.d. from $\mu$. Let $G \in \mathsf{O}(k)$ be a random (or deterministic) rotation, sampled independently of $X$, and consider the rotated point cloud $XG$. Then the expected symmetrized metric between $\mu_{XG}$ and $\mu$ can be bounded by (3) since

$$
\begin{aligned}
\mathbb{E}\left[\inf_{g \in \mathsf{O}(k)} W_p(\mu, \mu_{XGg^{-1}})\right] &= \mathbb{E}\left\{\mathbb{E}\left[\inf_{g \in \mathsf{O}(k)} W_p(\mu, \mu_{XGg^{-1}})\,\Big|\, G\right]\right\} \\
&\leq \mathbb{E}[W_p(\mu, \mu_X)].
\end{aligned} \tag{4}
$$

**Graphons.** Let $W: [0, 1]^2 \to [0, 1]$ and $A_n \in \mathbb{R}^{n \times n}_{\mathrm{sym}}$ be sampled as $(A_n)_{i,j} \sim \mathrm{Ber}(W(x_i, x_j))$ where $x_1, \dots, x_n$ are i.i.d. $\mathrm{Unif}([0, 1])$. Let $W_{A_n}$ be the step graphon associated to $A_n$. Then, [54, §10.4] implies

$$
\mathbb{E}[\delta_\square(W, W_{A_n})] \lesssim \frac{1}{\sqrt{\log(n)}}.^8
$$

---

[8] In fact, this bound holds with probability at least $1 - \exp(-\frac{n}{2\log n})$.

where $\delta_\square$ is the cut distance of graphons; see [54, §8.2.2] for a formal definition. Moreover, we have $W_{A_n} \to W$ in cut metric almost surely [54, Cor. 11.15]. Similarly, if $T_W : L^2([0,1]) \to L^2([0,1])$ is the integral operator associated with $W$, then by [37, Equation 4.4 and Lemma E.6],

$$\mathbb{E}[\overline{d}_2(T_W, T_{W_n})] = \mathbb{E}\left[\inf_{\sigma \in \mathsf{G}_\infty} \|T_W - T_{\sigma \cdot W_n}\|_{\mathrm{op},2}\right] \leq 2^{3/2}\mathbb{E}[\delta_\square(W, W_{X_n})]^{1/2} \lesssim (\log n)^{-1/4}.$$

See Appendix G.1 for the definitions of the symmetrized metric and norm on graphons as integral operators.

### D.3.2 Deterministic sampling

**Uniform grid.** Suppose $f : [0,1]^k \to \mathbb{R}$ is $L$-Lipschitz with respect to $\|\cdot\|_p$, and consider its values on a uniform grid $X_{i_1,\ldots,i_k} = f((i_1-1)/n, \ldots, (i_k-1)/n) \in (\mathbb{R}^n)^{\otimes k}$. If we extend $X$ to a step function as usual by $f_X(x_1, \ldots, x_k) = X_{\lceil x_1 n \rceil, \ldots, \lceil x_k n \rceil}$, then

$$\begin{aligned}
\|f - f_X\|_q &\leq \|f - f_X\|_\infty \\
&\leq L \sup_{x_1,\ldots,x_k \in [0,1]} \|(x_1, \ldots, x_k) - ((\lceil x_1 n \rceil - 1)/n, \ldots (\lceil x_k n \rceil - 1)/n)\|_p \\
&= L\|(1/n, \ldots, 1/n)\|_p = \frac{Lk^{1/p}}{n},
\end{aligned}$$

for all $q \in [1, \infty]$. Also note that if we evaluate an $L$-Lipschitz graphon $W : [0,1]^2 \to [0,1]$ on such a uniform grid, we have

$$\|T_W - T_{W_n}\|_{\mathrm{op}} \leq \|W - W_n\|_2 \leq \frac{L\sqrt{2}}{n}.$$

**Local averaging.** For any $f \in L^p([0,1]^k)$, we can locally average it over hypercubes of side length $1/n$ to produce values

$$X_{i_1,\ldots,i_k} = n^k \int_{(i_1-1)/n}^{i_1/n} \cdots \int_{(i_k-1)/n}^{i_k/n} f(x_1, \ldots, x_k) \, dx_1 \cdots dx_k,$$

and again extend these values to a step function $f_X$. In this case,

$$\|f - f_X\|_p \leq 2\mathrm{dist}(f, V_n),$$

so, we get the optimal rate of convergence.

## E   Generalization bounds: details and missing proofs from Section 4

We apply the framework connecting robustness and generalization established by [88], which is built on the idea that algorithmic robustness—that is, a model's stability to input perturbations—is fundamentally linked to its ability to generalize. We refer readers to [88] for the necessary background. This framework has also been recently employed to derive generalization bounds for GNNs in [79], though their analysis is restricted to graphs with bounded size. Similar techniques are used in [47, 68].

**Any-dimensional generalization bound from algorithmic robustness.** We consider an any-dimensional supervised learning task where consistent sequences model the input and output space

$$\mathbb{V} = \{(V_n), (\varphi_{N,n}), (\mathsf{G}_n)\} \quad \text{and} \quad \mathbb{U} = \{(U_n), (\psi_{N,n}), (\mathsf{G}_n)\},$$

with associated symmetrized metrics $\overline{d}_V$ and $\overline{d}_U$. The dataset $s$ consists of $N$ input-output pairs $(x_i, y_i) \in X \times Y \subseteq V_\infty \times U_\infty$, where $X \times Y$ are subsets whose sequence of orbit closures are compact in the symmetrized metrics. More precisely, $(x_i, y_i)$ are finite-dimensional representatives of equivalence classes in $V_\infty \times U_\infty$. The hypothesis class $\mathcal{H}$ consists of functions $\overline{V_\infty} \to \overline{U_\infty}$ parametrized by neural networks. A learning algorithm $\mathcal{A}$ is a mapping

$$\mathcal{A} : (V_\infty \times U_\infty)^N \to \mathcal{H}.$$

We write $\mathcal{A}_s$ for the hypothesis learned from the dataset $s$.

Assume training is performed using a neural network model that is $L(r)$-*locally Lipschitz transferable*. (Recall from Proposition D.1 that this implies $\mathcal{A}_s$ is $L(r)$-locally Lipschitz on $B(0,r)$ for all $r > 0$ with respect to the symmetrized metrics $\overline{d}_V$ and $\overline{d}_U$.) Since $X \times Y$ is compact, and hence bounded, there exists a constant $c_s > 0$ such that $\mathcal{A}_s \colon \overline{V_\infty} \to \overline{U_\infty}$ is $c_s$-Lipschitz on $X \times Y$ with respect to the symmetrized metrics. Further, let the loss function $\ell \colon \overline{U_\infty} \times \overline{U_\infty} \to \mathbb{R}$ be bounded by $M$, and $c_\ell$-Lipschitz with respect to the product metric $d\big((x,y),(x',y')\big) := \overline{d}_U(x,x') + \overline{d}_U(y,y')$. By applying the framework of [88] to the limit space, we immediately obtain a generalization bound for learning tasks where the data consists of inputs of varying dimensions. We note that this is not the result stated in Proposition 4.2 of the main paper; the version claimed there will be established later in Proposition E.3.

**Proposition E.1** (Any-dimensional generalization bound). *Assume that the training data consists of $N$ i.i.d. samples $s = (x_i, y_i) \sim \hat{\mu}$ from a measure $\hat{\mu}$ supported on $X \times Y \subseteq V_\infty \times U_\infty$, where $X$ and $Y$ have finite $\varepsilon$-covering numbers $C_X(\varepsilon), C_Y(\varepsilon)$ with respect to the symmetrized metrics for all $\varepsilon > 0$. Then, for any $\delta > 0$, with probability at least $1 - \delta$, the generalization error satisfies*

$$\left| \frac{1}{N} \sum_{i=1}^N \ell(\mathcal{A}_s(x_i), y_i) - \mathbb{E}_{(x,y) \sim \hat{\mu}} \ell(\mathcal{A}_s(x), y) \right|$$

$$\leq \inf_{\gamma > 0} \left( c_\ell(c_s \vee 1)\gamma + M\sqrt{\frac{2 C_X(\gamma/4) C_Y(\gamma/4) \log 2 + 2\log(1/\delta)}{N}} \right) \tag{5}$$

$$\leq c_\ell(c_s \vee 1)\xi^{-1}(N) + M\sqrt{(2\log 2)\xi^{-1}(N)^2 + \frac{2\log(1/\delta)}{N}}, \tag{6}$$

*where $\xi(r) := \frac{C_X(r/4) C_Y(r/4)}{r^2}$ and we set $\gamma = \xi^{-1}(N)$ in the second line to obtain the third.*

**Remark E.2.** *We make the following observations.*

1. *The bound (6) converges to 0 as $N \to \infty$. Indeed, $\xi$ is strictly decreasing, and hence its inverse is well-defined and also strictly decreasing. Since $\xi(x) \to \infty$ as $x \to 0^+$, we get $\xi^{-1}(x) \to 0$ as $x \to \infty$.*

2. *The generalization bound reveals that the ability to generalize improves with greater model transferability/stability (i.e., smaller Lipschitz constants), and deteriorates with increasing geometric complexity of the data space (i.e., larger covering numbers).*

3. *We emphasize that $\hat{\mu}$ is a distribution on $V_\infty \times U_\infty$, ensuring that every sample drawn from $\hat{\mu}$ admits a finite-dimensional representative, i.e., $(x_i, y_i) \in V_n \times U_n$ for some $n$. This reflects the realistic setting in which data consists of finite-dimensional inputs. This stands in contrast to prior work on GNN generalization bounds [47, 68], which considers data distributions on $\overline{V_\infty}$—the space of graphon signals in [48], and the space of iterated degree measures in [68]. Such an assumption is somewhat unrealistic, as many elements in these spaces cannot be realized as finite-dimensional data.*

4. *Note that $\hat{\mu}$ induces a distribution $(\hat{\mu}(V_n \times U_n))_{n \in \mathbb{N}}$ over sample dimensions in $\mathbb{N}$, which inherently places less weight on larger sizes. Consequently, the generalization bound does not offer guarantees on the* asymptotic performance *of the model as the input dimension $n \to \infty$. Next, we will derive the second generalization bound that addresses this problem.*

*Proof.* For all $(x_1, y_1), (x_2, y_2) \in X \times Y$,

$$|\ell(\mathcal{A}_s(x_1), y_1) - \ell(\mathcal{A}_s(x_2), y_2)| \leq c_\ell(c_s \overline{d}_V(x_1, x_2) + \overline{d}_U(y_1, y_2))$$

$$\leq c_\ell(c_s \vee 1)(\overline{d}_V(x_1, x_2) + \overline{d}_U(y_1, y_2)).$$

Applying [88, Theorem 14] yields that the algorithm $\mathcal{A}$ is $(C_X(\gamma/4) C_Y(\gamma/4), \gamma c_\ell(c_s \vee 1))$-robust [88, Definition 2] for all $\gamma > 0$. Further, applying [88, Theorem 3] gives the generalization bound (5). Finally, (6) follows by taking the $\gamma$ as defined. $\square$

**Size-generalization bound: train on finite sizes and test on the limit space.** The previous generalization bound follows the classical statistical learning setup, where both training and test data are

assumed to be sampled i.i.d. from the same distribution. However, in any-dimensional learning, we are typically concerned with a different scenario: training on data of smaller sizes and testing on data of larger sizes. This motivates the need for a new form of generalization bound that accounts for such settings.

We propose the following set-up (described in the main paper). Let $\mu$ be a probability distribution supported on $X \times Y \subseteq \overline{V_\infty} \times \overline{U_\infty}$, which are subsets whose sequence of orbit closures is compact in the symmetrized metrics. Consider a random sampling procedure

$$\mathcal{S}_n : \overline{V_\infty} \times \overline{U_\infty} \to V_n \times U_n,$$

such that for all $n$ and for all $(x, y) \in \mathrm{supp}(\mu)$, we have $\mathrm{supp}(\mathcal{S}_n(x, y)) \subseteq X \times Y$. This sampling induces a distribution $\mu_n$ on $V_n \times U_n$ via the sampling procedure; that is,

$$\mu_n := \mathrm{Law}(x_n, y_n), \quad \text{where } (x_n, y_n) \sim \mathcal{S}_n(x, y) \text{ and } (x, y) \sim \mu.$$

**Proposition E.3** (Size-generalization bound). *Suppose the training data consists of $N$ i.i.d. samples $s = (x_i, y_i) \sim \mu_n$. Then, for any $\delta > 0$, with probability at least $1 - \delta$, the generalization error satisfies*

$$\left| \frac{1}{N} \sum_{i=1}^N \ell(\mathcal{A}_s(x_i), y_i) - \mathbb{E}_{(x,y)\sim\mu} \ell(\mathcal{A}_s(x), y) \right|$$

$$\leq c_\ell (c_s \vee 1) \left( \xi^{-1}(N) + W_1(\mu, \mu_n) \right) + M\sqrt{(2\log 2)\,\xi^{-1}(N)^2 + \frac{2\log(1/\delta)}{N}}, \quad (7)$$

*where $W_1$ denotes the Wasserstein-1 distance, and $\xi(r) := \frac{C_X(r/4)\,C_Y(r/4)}{r^2}$, with $C_X(\varepsilon)$ and $C_Y(\varepsilon)$ denoting the $\varepsilon$-covering numbers of $X$ and $Y$, respectively, with respect to the symmetrized metrics. Moreover, assuming that the sampling procedure converges in expectation at a rate $R(n)$, i.e.,*

$$\mathbb{E}_{(x_n, y_n)\sim\mathcal{S}_n(x,y)} \left[ \overline{\mathrm{d}}_V(x, [x_n]) + \overline{\mathrm{d}}_U(y, [y_n]) \right] \lesssim R(n),$$

*we have that*

$$\left| \frac{1}{N} \sum_{i=1}^N \ell(\mathcal{A}_s(x_i), y_i) - \mathbb{E}_{(x,y)\sim\mu} \ell(\mathcal{A}_s(x), y) \right|$$

$$\lesssim c_\ell (c_s \vee 1) \left( \xi^{-1}(N) + R(n) \right) + M\sqrt{(2\log 2)\,\xi^{-1}(N)^2 + \frac{2\log(1/\delta)}{N}}. \quad (8)$$

**Remark E.4.** *We make the following observations.*

1. *The bound (8) converges to $0$ if both the training input dimension $n$ and the amount of data $N$ goes to $\infty$. Indeed, we have justified in Remark E.2 that (6) converges to $0$ as $N \to \infty$. The only additional term in (8) is $R(n)$ which converges to $0$ as $n \to \infty$.*

2. *This new generalization bound aligns with the setup where training is performed on inputs of fixed size $n$ (also naturally extends to inputs of varying finite sizes), and testing evaluates the asymptotic performance as $n \to \infty$. It can therefore be interpreted as a* size generalization bound*, accounts for distributional shifts induced by size variation. As a consequence, an additional term appears in the bound, reflecting the convergence rate of the sampling procedure.*

*Proof.* By triangle inequality,

$$\left| \frac{1}{N} \sum_{i=1}^N \ell(\mathcal{A}_s(x_i), y_i) - \mathbb{E}_{(x,y)\sim\mu} \ell(\mathcal{A}_s(x), y) \right| \leq \underbrace{\left| \frac{1}{N} \sum_{i=1}^N \ell(\mathcal{A}_s(x_i), y_i) - \mathbb{E}_{(x,y)\sim\mu_n} \ell(\mathcal{A}_s(x), y) \right|}_{=:T_1}$$

$$+ \underbrace{\left| \mathbb{E}_{(x,y)\sim\mu} \ell(\mathcal{A}_s(x), y) - \mathbb{E}_{(x,y)\sim\mu_n} \ell(\mathcal{A}_s(x), y) \right|}_{=:T_2}.$$

We bound the two terms separately. By the Kantorovich-Rubinstein duality, we have almost surely that

$$T_2 \leq c_\ell (c_s \vee 1) \cdot W_1(\mu, \mu_n).$$

To bound $T_1$, recall that $\text{supp}(\mathcal{S}_n(x,y)) \subseteq X \times Y$ for all $(x,y) \in \text{supp}(\mu)$. It follows that $\text{supp}(\mu_n) \subseteq X \times Y$, which is therefore totally bounded. Its covering number is upper bounded by that of $X \times Y$. Applying Proposition E.1 with $\hat{\mu} = \mu_n$ yields (7).

Finally, to bound $W_1(\mu, \mu_n)$, we note that the sampling procedure induces a natural coupling between $\mu$ and $\mu_n$. Using this coupling,

$$W_1(\mu, \mu_n) \le \mathbb{E}_{(x,y)\sim\mu,\ (x_n,y_n)\sim\mathcal{S}_n(x,y)} \left[ \overline{\mathrm{d}}_V(x, [x_n]) + \overline{\mathrm{d}}_U(y, [y_n]) \right] \lesssim R(n),$$

which yields (8); completing the proof. $\qquad\square$

**Practicality of the assumptions.** Finally, we reflect on the key assumptions required for the bound (8) to hold and assess their practicality in more specific settings. First, we assumed that the loss function is bounded (and, consequently, Lipschitz continuous). We note that these assumptions are relatively standard [1, 76, 3]. When the predictions and target outputs are bounded, several widely-used loss functions, such as cross-entropy, L1-loss, L2-loss, and Huber loss, are all bounded and Lipschitz continuous. Moreover, many clipped loss functions also satisfy this assumption. Second, regarding the assumption of a sampling procedure that converges in expectation at a rate $R(n)$, we refer readers to Appendix D.3 for examples involving sets, graphs, and point clouds. Most importantly, the bound critically depends on the compactness of $X$ and $Y$ in the symmetrized metrics, and on the sampling procedure generating finite-size samples that remain within this compact space, i.e., $\text{supp}(\mathcal{S}_n(x,y)) \subset X \times Y$ for all $(x,y) \in \text{supp}(\mu)$. Below, we provide two concrete examples involving sets and graphs where these assumptions hold, and where bounds for the covering numbers are explicitly known. In these specific settings, our generalization bound applies directly.

**Example E.5** (Probability measures supported on compact set). *Consider $\mathbb{V}_{\text{dup}}^{\oplus d}$, the duplication consistent sequence for sets endowed with the normalized $\ell_p$ metric. See Appendix F.1 for the precise definitions. The limit space is $\overline{V_\infty} = L^p([0,1], \mathbb{R}^d)$, and the space of orbit closures of $\mathbb{V}$ is $\mathcal{P}_p(\mathbb{R}^d)$, the space of probability measures on $\mathbb{R}^d$ with finite $p$-th moment, endowed with the Wasserstein-$p$ distance. Fix a compact set $\Omega \subseteq \mathbb{R}^d$, and let*

$$X := \left\{ f \in L^p([0,1], \mathbb{R}^d) : \mu_f \text{ is supported on } \Omega \right\}.$$

*Note that the sequence of orbit closures in $X$, namely $\{\mu_f : f \in X\}$, is compact with respect to $W_p$. A bound on the covering number $C_X$ is given by [63, Theorem 2.2.11]. Consider the sampling procedure $\mathcal{S}_n : L^p([0,1], \mathbb{R}^d) \to \mathbb{R}^{n \times d}$ defined by drawing $z_i \overset{i.i.d.}{\sim} \text{Unif}([0,1])$, and setting $\mathcal{S}_n(f)_{i:} = f(z_i)$ for $i = 1, \dots, n$. Then, for all $f \in X$, each entry of $\mathcal{S}_n(f)$ lie in $\Omega$. Hence, we have $\mathcal{S}_n(f) \in X$. Our generalization bound therefore applies to this setting.*

**Example E.6** (Graphon signals with cut distance). *Consider $\mathbb{V}_{\text{dup}}^G$, the duplication-consistent sequence for graph signals endowed with the cut metrics. See Appendix G.1 for the precise definitions. Define the space*

$$X = \left\{ W \colon [0,1]^2 \to [0,1] \text{ measurable} : W(x,y) = W(y,x) \right\} \times \{ f \in L^\infty([0,1], \mathbb{R}) : \|f\|_\infty \le r \}.$$

*By [47, Theorem 3], the sequence of orbit closures in $X$ is compact with respect to the cut metric on graphon signals. Moreover, a bound on the covering number is also provided in the same result. Consider the sampling procedure $\mathcal{S}_n \colon X \to \mathbb{R}_{\text{sym}}^{n \times n} \times \mathbb{R}^n$ defined as follows: draw $z_i \overset{i.i.d.}{\sim} \text{Unif}([0,1])$, and set $\mathcal{S}_n(W, f) = (A, X)$, where $A_{ij} \sim \text{Ber}(W(z_i, z_j))$ and $X_i = f(z_i)$ for $i = 1, \dots, n$. Then, for all $(W, f) \in X$, the sampled pair $\mathcal{S}_n(W, f)$ belongs to $X$. This is the standard sampling procedure for graphon signals, and once again our generalization bound applies.*

### E.1 Transferable neural networks

To prove the transferability of a neural network, we first observe that compatibility and transferability are preserved under composition. Therefore, it suffices to verify these properties for each individual layer. Moreover, by Propositions C.6 and D.2, it is enough to prove the compatibility and Lipschitz continuity of each $f_n$ on the finite space, rather than analyzing the limiting function $f_\infty$ directly. This idea is formalized in the following proposition, which serves as a key tool in our transferability analysis of neural networks in the later sections. Importantly, this provides a general and easy-to-apply proof strategy. In contrast, previous works often begin by characterizing a natural limiting function $f_\infty$ (e.g., a graphon neural network), and then directly prove its Lipschitz continuity in the limit space—a process that typically requires case-specific proof techniques.

**Proposition E.7** (Transferable networks: detailed version of Proposition 5.1). *Let $(V_n^{(i)})_n, (U_n^{(i)})_n$ be consistent sequences for $i = 1, \ldots, D$. For each $i$, let $(W_n^{(i)}: V_n^{(i)} \to U_n^{(i)})$ be linear maps and $(\rho_n^{(i)}: U_n^{(i)} \to V_n^{(i+1)})$ be nonlinearities. Assume the following three properties hold.*

1. *The maps $\left(W_n^{(i)}\right), \left(\rho_n^{(i)}\right)$ are compatible.*

2. *The linear maps are uniformly bounded $\sup_{n,i} \|W_n^{(i)}\|_{\mathrm{op}} = L_W < \infty$.*

3. *The map $\rho_n^{(i)}$ is $L_i(r)$-Lipschitz on $\left\{u \in U_n^{(i)} : \|u\| < r\right\}$ for all $n$.*

*Then the composition $\left(W_n^{(D)} \circ \rho_n^{(D-1)} \circ \ldots \circ \rho_n^{(1)} \circ W_n^{(1)}\right)$ is locally Lipschitz transferable, extending to a function on $\overline{V_\infty} \to \overline{U_\infty}$ that is $L_{\mathrm{NN}}(r)$-Lipschitz on $\left\{v \in \overline{V_\infty^{(1)}} : \|v\| < r\right\}$, where we inductively define*

$$\ell_1 = L_1(L_W r), \quad \ell_{i+1} = L_{i+1}(L_W^{i+1} \ell_i), \qquad L_{\mathrm{NN}}(r) = L_W^D \prod_{i=1}^{D-1} \ell_i. \tag{9}$$

*In particular, if $\rho_n^{(i)}$ is $L_\rho$-Lipschitz for all $i, n$ then the composition is Lipschitz transferable, extending to a function on $\overline{V_\infty} \to \overline{U_\infty}$ that is $L_{\mathrm{NN}}$-Lipschitz where $L_{\mathrm{NN}} = L_W^D L_\rho^{D-1}$.*

**Remark E.8.** *By Proposition D.1, the composition is also $L_{\mathrm{NN}}(r)$-Lipschitz with respect to the symmetrized metrics on the same $r$-ball.*

*Proof.* Note that if $f_1: \overline{V_\infty^{(1)}} \to \overline{V_\infty^{(2)}}$ and $f_2: \overline{V_\infty^{(2)}} \to \overline{V_\infty^{(3)}}$ are $L_1(r)$- and $L_2(r)$-locally Lipschitz, respectively, then $f_2 \circ f_2$ is $L_2(L_1(r))L_1(r)$-locally Lipschitz. Our claim follows by an inductive application of this fact and Proposition D.2. □

# F  Example 1 (sets): details and missing proofs from Section 5.1

In this section, we study the transferability of architectures taking sets as inputs. Section F.1 introduces the consistent sequences we consider and Section F.2 presents results for three architectures: DeepSets[91], normalized DeepSets [9], and PointNet [66].

## F.1  Consistent sequences on sets

We present two examples of consistent sequences on sets. See Figure 1 in the main text for a graphical illustration.

**Zero-padding consistent sequence $\mathbb{V}_{\mathrm{zero}}$ with $\ell_p$ norm**

The zero-padding consistent sequence $\mathbb{V}_{\mathrm{zero}} = \{(V_n), (\varphi_{N,n}), (\mathsf{G}_n)\}$ is defined as follows: The index set $\mathbb{N} = (\mathbb{N}, \leq)$ is the poset of natural numbers with the standard ordering. Let $V_n = \mathbb{R}^n$ for every $n \in \mathbb{N}$, and the zero-padding embedding is given by, for $n \leq N$,

$$\begin{aligned} \varphi_{N,n}: \quad & \mathbb{R}^n \hookrightarrow \mathbb{R}^N \\ & (x_1, \ldots, x_n) \mapsto (x_1, \ldots, x_n, \underbrace{0, \ldots, 0}_{(N-n) \; 0\text{'s}}). \end{aligned}$$

The group of permutations $\mathsf{S}_n$ on $n$ letters acts on $\mathbb{R}^n$ by permuting coordinates: $(g \cdot x)_i := x_{g^{-1}(i)}$ for $g \in \mathsf{S}_n$. The embedding of groups is given by, for $n \leq N$,

$$\begin{aligned} \theta_{N,n}: \quad & \mathsf{S}_n \hookrightarrow \mathsf{S}_N \\ & g \mapsto \begin{bmatrix} g & 0 \\ 0 & I_{N-n} \end{bmatrix}. \end{aligned}$$

That is, view $\mathsf{S}_n$ as the subgroup of $\mathsf{S}_N$ which acts trivially on $n+1, \ldots, N$. In this case, $V_\infty$ can be identified with $\ell_0$, i.e., the space of infinite scalar sequences with finitely many nonzero entries. The limit group $\mathsf{G}_\infty$ is the group of permutations of $\mathbb{N}$ fixing all but finitely many indices.

We associate every infinite sequence $(x_i)_{i=1}^\infty \in V_\infty$ with the tuple of sequences $\big((x_i^+)_{i=1}^\infty, (x_i^-)_{i=1}^\infty\big)$. The sequence $(x_i^+)_{i=1}^\infty$ comprises the positive entries of $(x_i)$, ordered in descending order and extended with trailing zeros. The sequence $(x_i^-)_{i=1}^\infty$ comprises the negative entries of $(x_i)$, ordered in ascending order and similarly extended with trailing zeros. Notice that different sequences in $V_\infty$ are associated with the same tuple of sequences if and only if they belong to the same orbit under the action of $\mathsf{G}_\infty$. Hence, the orbit space of $V_\infty$ under the action of $\mathsf{G}_\infty$ can be identified with tuples of *ordered* infinite sequences. Specifically, one sequence consists of non-negative entries arranged in descending order, and the other sequence consists of non-positive entries arranged in ascending order, with both sequences having finitely many non-zero entries.

**The $\ell_p$ norm on $\mathbb{V}_{\mathrm{zero}}$.** We can endow each $V_n$ with the $\ell_p$-norms

$$\|x\|_p = \begin{cases} \left(\sum_{i=1}^n |x_i|^p\right)^{1/p} & \text{if } p \in [1, \infty), \\ \max_{i=1}^n |x_i| & \text{if } p = \infty. \end{cases}$$

It is easy to check by Proposition C.8 that this induces a norm on $V_\infty = \mathbb{R}^\infty$, which coincides with the $\ell_p$-norms on infinite sequences, which is also denoted as $\|\cdot\|_p$. The limit space is then

$$\overline{V_\infty} = \begin{cases} \ell_p = \{(x_i)_{i=1}^\infty : \sum_{i=1}^\infty |x_i|^p < \infty\} & \text{if } p \in [1, \infty), \\ c_0 = \{(x_i)_{i=1}^\infty : \lim_{i \to \infty} |x_i| = 0\} & \text{if } p = \infty. \end{cases}$$

Similarly, the space of orbit closures of $\overline{V_\infty}$ under the action of $\mathsf{G}_\infty$ can be identified with tuples of *ordered* infinite sequences. Specifically, one sequence consists of non-negative entries arranged in descending order, and the other sequence consists of non-positive entries arranged in ascending order, with both sequences in $\ell_p$ (for $p \in [1, \infty)$) or $c_0$ (for $p = \infty$). This space is endowed with the symmetrized metric $\overline{d}_p(x, y) = \min_{\sigma \in \mathsf{G}_\infty} \|x - \sigma \cdot y\|_p$.

**The $\ell_p$ norm on the direct sum $\mathbb{V}_{\mathrm{zero}}^{\oplus d}$.** The direct sum $\mathbb{V}_{\mathrm{zero}}^{\oplus d} = \{(\mathbb{R}^{n \times d}), (\varphi_{N,n}^{\oplus d}), (\mathsf{S}_n)\}$ defined in Definition C.4 extends the above to the case of a set of vectors in $\mathbb{R}^d$. To endow it with a norm, we first fix an arbitrary norm $\|\cdot\|_{\mathbb{R}^d}$ on $\mathbb{R}^d$. Then the $\ell_p$-norm on $\mathbb{R}^{n \times d}$ is defined analogously with respect to $\|\cdot\|_{\mathbb{R}^d}$, i.e., for $X \in \mathbb{R}^{n \times d}$,

$$\|X\|_p = \begin{cases} \left(\sum_{i=1}^n \|X_{i:}\|_{\mathbb{R}^d}^p\right)^{1/p} & \text{if } p \in [1, \infty), \\ \max_{i=1}^n \|X_{i:}\|_{\mathbb{R}^d} & \text{if } p = \infty. \end{cases}$$

Analogously, in this case, $V_\infty$ can be seen as the space of infinite sequences in $\mathbb{R}^d$ with finitely many nonzero entries, and the limit space $\overline{V_\infty}$ is the corresponding $\ell_p$ space (if $p \in [1, \infty)$) or $c_0$ space (if $p = \infty$). The space of orbit closures can be seen as tuples of infinite sequences in $\mathbb{R}^d$, ordered in lexicographic order.

**Duplication consistent sequence $\mathbb{V}_{\mathrm{dup}}$ with normalized $\ell_p$-norms**

The duplication consistent sequence $\mathbb{V}_{\mathrm{dup}} = \{(V_n), (\varphi_{N,n}), (\mathsf{G}_n)\}$ is defined as follows. The index set $(\mathbb{N}, \cdot \mid \cdot)$ is the set of natural numbers with divisibility partial order, where $n \preceq N$ if and only if $n \mid N$. Let $V_n = \mathbb{R}^n$ for all $n \in \mathbb{N}$, and the duplication embeddings is given by

$$\varphi_{N,n} : \quad \mathbb{R}^n \hookrightarrow \mathbb{R}^N$$
$$(x_1, \ldots, x_n) \mapsto x \otimes \mathbb{1}_{N/n} = (\underbrace{x_1, \ldots, x_1}_{N/n \text{ times}}, \ldots, x_n, \ldots, x_n),$$

for $n \preceq N$. The group embeddings are given by

$$\theta_{N,n} : \quad \mathsf{S}_n \hookrightarrow \mathsf{S}_N$$
$$g \mapsto g \otimes I_{N/n},$$

for $n \preceq N$. That is, $g \in \mathsf{S}_n$ acts on $[N]$ by sending $(i-1)N/n + j$ to $(g(i) - 1)N/n + j$ for $i = 1, \ldots, n$ and $j = 1, \ldots, N/n$.

In this case, $V_\infty$ can be identified with step functions on $[0,1]$ whose discontinuity points are in $\mathbb{Q}$: each $x \in \mathbb{R}^n$ corresponds to $f_x \colon [0,1] \to \mathbb{R}$ where $f_x(t) = x_{\lceil tn \rceil}$ for $t > 0$ and $f(0) = x_1$. In other words, $f_x$ is the step function which takes value $x_i$ on $I_{i,n} = \left(\frac{i-1}{n}, \frac{i}{n}\right]$ for $i = 1, \dots, n$. Indeed, all equivalent objects $x$ and $\varphi_{N,n} x$ correspond to the same function in this way. Therefore, $V_\infty$ can be seen as the union of step functions of this form for $n \in \mathbb{N}$. Under this identification, permutations $\mathsf{S}_n$ permute the $n$ intervals $I_{i,n}$ and act on functions by $g \cdot f = f \circ g^{-1}$. The limit group $\mathsf{G}_\infty$ is the union of such interval permutations. The orbit space of the $\mathsf{G}_\infty$-action on $V_\infty$ can be identified with monotonically increasing step functions on $[0,1]$ whose discontinuity points are in $\mathbb{Q}$. Alternatively, the orbit space can be identified with the space of empirical measures on $\mathbb{R}$: each $x \in \mathbb{R}^n$ corresponds to the empirical measure $\mu_x = \frac{1}{n} \sum_{i=1}^n \delta_{x_i}$. Indeed, any equivalent objects $x$ and $\varphi_{N,n} x$ are identified with the same measure; furthermore, this resulting measure is constant on orbits under the $\mathsf{G}_\infty$-action.

Under the identification of $V_\infty$ with step functions, a step function $f \in V_\infty$ is identified with the probability measure $\mu_f$, defined as the distribution of $f(T)$ for $T$ uniformly sampled from $[0,1]$. Indeed, all elements in the orbit of $f$ under the $\mathsf{G}_\infty$-action correspond to this same measure $\mu_f$. Note that the generalized inverse CDF of $f(T)$ is precisely the 'sorted' version of $f$ (called its increasing rearrangement), relating the above two perspectives on the orbit space. The latter view of the orbit space as a sequence of measures generalizes readily to other consistent sequences obtained from $\mathbb{V}_{\mathrm{dup}}$, such as $\mathbb{V}_{\mathrm{dup}}^{\oplus d}$ which we consider below, so we shall take this view from now.

**Normalized $\ell_p$ norm on $\mathbb{V}_{\mathrm{dup}}$.** We can endow each space $V_n$ with the normalized $\ell_p$-norms

$$\|x\|_{\overline{p}} = \begin{cases} \left(\frac{1}{n} \sum_{i=1}^n |x_i|^p\right)^{1/p} & \text{if } p \in [1, \infty), \\ \max_{i=1}^n |x_i| & \text{if } p = \infty. \end{cases}$$

Using Proposition C.8, it is straightforward to verify that this defines a norm on $V_\infty$. Under the identification of $V_\infty$ with step functions, the induced norm on $V_\infty$ coincides with the conventional $L^p$ norm on measurable functions, given by

$$\|f\|_p = \begin{cases} \left(\int_0^1 |f(t)|^p \, dt\right)^{1/p} & \text{if } p \in [1, \infty), \\ \sup_{t \in [0,1]} |f(t)| & \text{if } p = \infty. \end{cases}$$

That is, for any $x \in \mathbb{R}^n$, we have $\|x\|_{\overline{p}} = \|f_x\|_p$, where $f_x$ is the corresponding step function. The limit space is then

$$\overline{V_\infty} = \begin{cases} L^p([0,1]) = \left\{ f \colon [0,1] \to \mathbb{R} \text{ measurable} : \int_0^1 |f(t)|^p \, dt < \infty \right\} & \text{if } p \in [1, \infty), \\ \left\{ f \colon [0,1] \to \mathbb{R} \; : \; \begin{array}{l} f \text{ is bounded and continuous on } [0,1] \setminus \mathbb{Q}, \\ \text{with left and right limits at every } x \in [0,1] \cap \mathbb{Q} \end{array} \right\} & \text{if } p = \infty, \end{cases}$$

where the result for $p = \infty$ follows from [21, Chap. VII.6]. These are a subspace of so-called *regulated functions*, which have left and right limits at each $x \in [0,1]$.

When $p \in [1, \infty)$, the space of orbit closures, equipped with the symmetric metric, can be identified with $\mathcal{P}_p(\mathbb{R})$, the space of probability measures on $\mathbb{R}$ with finite $p$-th moment, endowed with the Wasserstein $p$-distance. In the case $p = \infty$, the space of orbit closures corresponds to a subset of $\mathcal{P}_\infty(\mathbb{R})$, the space of probability measures on $\mathbb{R}$ with bounded support, equipped with the Wasserstein $\infty$-distance. This is formalized and proved by the following propositions:

**Proposition F.1.** *For any $p \in [1, \infty]$ and all $f, g \in \overline{V_\infty}$, the symmetrized metric $\overline{\mathrm{d}}_p(f,g) := \inf_{\sigma \in \mathsf{G}_\infty} \|\sigma \cdot f - g\|_p$ equals the Wasserstein $p$-distance between the associated measures:*

$$\overline{\mathrm{d}}_p(f,g) = W_p(\mu_f, \mu_g).$$

*Proof.* We first prove they match on $V_\infty$. Consider vectors $x \in \mathbb{R}^n$, $y \in \mathbb{R}^m$ under the action of $\mathsf{S}_n$, $\mathsf{S}_m$ respectively, and let $N = \mathrm{lcm}(n, m)$. Then, by standard results on the Wasserstein distance of empirical measures [65, §2.2],

$$\overline{\mathrm{d}}_p(x,y) = \begin{cases} \min_{\sigma \in \mathsf{S}_N} \left(\frac{1}{N} \sum_{i=1}^N \left| \varphi_{N,n}(x)_{\sigma(i)} - \varphi_{N,m}(y)_i \right|^p \right)^{1/p} = W_p(\mu_x, \mu_y) & \text{if } p \in [1, \infty), \\ \min_{\sigma \in \mathsf{S}_N} \max_{i=1}^N \left| \varphi_{N,n}(x)_{\sigma(i)} - \varphi_{N,m}(y)_i \right| = W_\infty(\mu_x, \mu_y) & \text{if } p = \infty. \end{cases}$$

where $\mu_x$ was defined above, and $W_p$ is the Wasserstein $p$-distance. Hence under the identification with step functions, for any $f, g \in V_\infty$ we have $\overline{d}_p(f, g) = W_p(\mu_f, \mu_g)$.

Now consider the limit points. Let $(f_n), (g_n)$ be two Cauchy sequences in $V_\infty$ with $f_n \to f, g_n \to g$ in $\overline{V_\infty}$ with respect to the $L^p$ norm. Then

$$\left| \overline{d}_p(f_n, g_n) - \overline{d}_p(f, g) \right| \leq \overline{d}_p(f, f_n) + \overline{d}_p(g_n, g) \leq \|f - f_n\|_p + \|g_n - g\|_p \to 0.$$

Similarly, since for any $\tilde{f}, \tilde{g} \in \overline{V_\infty}$,

$$W_p(\mu_{\tilde{f}}, \mu_{\tilde{g}}) \leq \left( \mathbb{E}_{T \sim \mathrm{Unif}[0,1]} |\tilde{f}(T) - \tilde{g}(T)|^p \right)^{1/p} = \|\tilde{f} - \tilde{g}\|_p,$$

we also get

$$|W_p(\mu_{f_n}, \mu_{g_n}) - W_p(\mu_f, \mu_g)| \to 0.$$

But for all $n$, $\overline{d}_p(f_n, g_n) = W_p(\mu_{f_n}, \mu_{g_n})$, so by the uniqueness of the limit, $\overline{d}_p(f, g) = W_p(\mu_f, \mu_g)$. □

**Proposition F.2.** *For $p \in [1, \infty)$, the space of orbit closures $\{\mu_f : f \in \overline{V_\infty}\}$ coincides with $\mathcal{P}_p(\mathbb{R})$. For $p = \infty$, this set is a subset of $\mathcal{P}_\infty(\mathbb{R})$.*

*Proof.* For $p \in [1, \infty)$, by definition, the space of orbit closures is the set of probability measures $\{\mu_f : f \in L^p([0, 1])\}$. We claim that this set is equal to $\mathcal{P}_p(\mathbb{R})$. Observe that $((\mathbb{E}_{X \sim \mu_f} |X|^p)^{1/p} = \|f\|_p$. On the one hand, this implies that if $f \in L^p([0, 1])$, then $\mu_f \in \mathcal{P}_p(\mathbb{R})$. Conversely, given any $\mu \in \mathcal{P}_p(\mathbb{R})$, let $f$ be the generalized inverse of the CDF of $\mu$, then $\mu_f = \mu$ and $f \in L^p([0, 1])$. Hence $\mu \in \mathcal{P}_p(\mathbb{R})$ implies that $\mu = \mu_f$ for $f \in L^p([0, 1])$.

For $p = \infty$, note that any $f \in \overline{V_\infty}$ is bounded, so the support of $\mu_f$ is compact, implying that $\mu_f \in \mathcal{P}_\infty(\mathbb{R})$. □

**Norms on $V_{\mathrm{dup}}^{\oplus d}$.** Similarly, we fix an arbitrary norm $\| \cdot \|_{\mathbb{R}^d}$ on $\mathbb{R}^d$ and define the norms on $\mathbb{R}^{n \times d}$ with respect to $\| \cdot \|_{\mathbb{R}^d}$:

$$\|X\|_{\overline{p}} := \begin{cases} \left( \frac{1}{n} \sum_{i=1}^n \|X_{i:}\|_{\mathbb{R}^d}^p \right)^{1/p} & \text{if } p \in [1, \infty), \\ \max_{i=1}^n \|X_{i:}\|_{\mathbb{R}^d} & \text{if } p = \infty. \end{cases} \tag{10}$$

For $p \in [1, \infty)$, the space of orbit closures can be identified with $\mathcal{P}_p(\mathbb{R}^d)$ endowed with the Wasserstein $p$-distance with respect to $\| \cdot \|_{\mathbb{R}^d}$. For $p = \infty$, the space of orbit closures can be seen as a subset of $\mathcal{P}_\infty(\mathbb{R}^d)$ with the Wasserstein-$\infty$ distance with respect to $\| \cdot \|_{\mathbb{R}^d}$. This is because $\mathbb{R}^d$ is a standard Borel space [77, Thm. 3.3.13], so the same arguments as in Propositions F.1-F.2 apply.

### F.2 Invariant networks on sets

We consider three prominent permutation-invariant neural network architectures for set-structured data: DeepSets [91], normalized DeepSets [9], and PointNet [66]. These models are defined as follows:

$$\mathrm{DeepSet}_n(X) = \sigma \left( \sum_{i=1}^n \rho(X_{i:}) \right), \quad \overline{\mathrm{DeepSet}}_n(X) = \sigma \left( \frac{1}{n} \sum_{i=1}^n \rho(X_{i:}) \right), \quad \text{and}$$

$$\mathrm{PointNet}_n(X) = \sigma \left( \max_{i=1}^n \rho(X_{i:}) \right),$$

where $\rho \colon \mathbb{R}^d \to \mathbb{R}^h$ and $\sigma \colon \mathbb{R}^h \to \mathbb{R}$ are multilayer perceptrons (MLPs). In the case of PointNet, the maximum is taken entrywise over vectors in $\mathbb{R}^h$.

They follow the same paradigm $f_n(X) = \sigma \left( \mathrm{Agg}_{i=1}^n \rho(X_{i:}) \right)$ where the three models use different permutation-invariant aggregations $\mathrm{Agg}$. We refer the reader to [9] for a comprehensive study of the expressive power of these models in the any-dimensional setting. In particular, they show that normalized DeepSets (respectively, PointNet) can uniformly approximate all set functions that are uniformly continuous with respect to the Wasserstein-1 distance (respectively, the Hausdorff distance). In contrast, our work focuses on transferability and size generalization, rather than expressive power.

### F.2.1 Transferability analysis: proof of Corollary 5.2

We prove the Corollary by instantiating Proposition E.7. The invariant network is given by the following composition

$$\mathbb{R}^{n \times d} \xrightarrow[\text{row-wise}]{\rho^{\oplus n}} \mathbb{R}^{n \times h} \xrightarrow{\text{Agg}_{i=1}^{n}} \mathbb{R}^{h} \xrightarrow{\sigma} \mathbb{R},$$

where we use $\rho^{\oplus n}$ to denote the row-wise application of the same $\rho : \mathbb{R}^d \to \mathbb{R}^h$. Thus, it suffices to analyze each term in this composition individually.

**DeepSet.** Notice that the sum aggregation is not compatible with the duplication embedding. Indeed, for any $x \in \mathbb{R}^n$ such that $\sum_i x_i \neq 0$, and for $n \mid N, n \neq N$,

$$\sum_{i=1}^{N} (x \otimes \mathbb{1}_{N/n})_i = (N/n) \sum_{i=1}^{n} x_i \neq \sum_{i=1}^{n} x_i.$$

Therefore, DeepSet is not compatible with respect to the duplication consistent sequence in general. We now prove its compatibility and transferability with respect to zero-padding.

**Corollary F.3.** *Fix arbitrary norms $\|\cdot\|_{\mathbb{R}^d}$ on $\mathbb{R}^d$ and $\|\cdot\|_{\mathbb{R}^h}$ on $\mathbb{R}^h$. Let $\rho : \mathbb{R}^d \to \mathbb{R}^h$ be $L_\rho$-Lipschitz with $\rho(0) = 0$, and $\sigma : \mathbb{R}^h \to \mathbb{R}$ be $L_\sigma$-Lipschitz, with respect to the norms $\|\cdot\|_{\mathbb{R}^d}, \|\cdot\|_{\mathbb{R}^h}$, and $|\cdot|$. Then, the sequence of maps $(\mathrm{DeepSet}_n)$ is $(L_\rho L_\sigma)$-Lipschitz transferable with respect to the zero-padding consistent sequence $\mathbb{V}_{\mathrm{zero}}^{\oplus d}$ (equipped with the $\ell_1$-norm induced by $\|\cdot\|_{\mathbb{R}^d}$) and the trivial consistent sequence $\mathbb{V}_{\mathbb{R}}$ (with absolute value norm). Therefore, $(\mathrm{DeepSet}_n)$ extends to*

$$\mathrm{DeepSet}_\infty : \ell_1(\mathbb{R}^d) \to \mathbb{R}, \quad \mathrm{DeepSet}_\infty((x_i)_{i=1}^\infty) = \sigma \left( \sum_{i=1}^{\infty} \rho(x_i) \right),$$

*which is $(L_\rho L_\sigma)$-Lipschitz with respect to the $\ell_1$-norm on the infinite sequences.*

*Proof.* We model each intermediate space with consistent sequences:

$$\mathbb{V}_{\mathrm{zero}}^{\oplus d} = (\mathbb{R}^{n \times d}) \xrightarrow[\text{(row-wise)}]{\rho^{\oplus n}} V_{\mathrm{zero}}^{\oplus h} = (\mathbb{R}^{n \times h}) \xrightarrow{\sum_{i=1}^{n}} \mathbb{V}_{\mathbb{R}^h} = (\mathbb{R}^h) \xrightarrow{\sigma} \mathbb{V}_{\mathbb{R}} = (\mathbb{R}).$$

We first check the compatibility of each map.

- As long as $\rho(0) = 0$, the $\rho$-map is compatible because

  $$\rho^{\oplus N} \left( \begin{bmatrix} X \\ 0 \end{bmatrix} \right) = \begin{bmatrix} \rho^{\oplus n}(X) \\ 0 \end{bmatrix} \text{ for all } X \in \mathbb{R}^{n \times d}, 0 \in \mathbb{R}^{(N-n) \times d}, n \leq N,$$

  and the row-wise application makes sure $\rho$ is $\mathsf{S}_n$-equivariant.

- The sum aggregation $\mathrm{Agg}_{i=1}^{n} = \sum_{i=1}^{n}$ is compatible because adding zeros does not change the sum, and the summation operation is $\mathsf{S}_n$-invariant.

- The map $\sigma$ is between two trivial consistent sequences. Hence it is automatically compatible.

Endow $\mathbb{V}_{\mathrm{zero}}^{\oplus d}$ with the $\ell_1$ norm induced by $\|\cdot\|_{\mathbb{R}^d}$, and $\mathbb{V}_{\mathrm{zero}}^{\oplus h}$ with the $\ell_1$ norm with induced by $\|\cdot\|_{\mathbb{R}^h}$. $\mathbb{V}_{\mathbb{R}^h}, \mathbb{V}_{\mathbb{R}}$ are endowed with $\|\cdot\|_{\mathbb{R}^h}$ and $|\cdot|$ respectively. Next, we check the Lipschitz transferability of each map.

- The $\rho$-map is $L_\rho$-Lispchitz transferable map because for all $n$, we can prove $\rho^{\oplus n} : \mathbb{R}^{n \times d} \to \mathbb{R}^{n \times h}$ (applying the same $\rho$ row-wise) is $L_\rho$ Lipschitz with respect to the $\ell_1$ norms:

$$\|\rho^{\oplus n}(X) - \rho^{\oplus n}(Y)\|_1 = \sum_{i=1}^{n} \|\rho(X_{i:}) - \rho(Y_{i:})\|_{\mathbb{R}^h} \leq \sum_{i=1}^{n} L_\rho \|X_{i:} - Y_{i:}\|_{\mathbb{R}^d} = L_\rho \|X - Y\|_1.$$

- The sum aggregation $\mathrm{Agg}_{i=1}^{n} = \sum_{i=1}^{n}$ is 1-Lipschitz transferable because for all $n$,

$$\left\| \sum_{i=1}^{n} X_{i:} - \sum_{i=1}^{n} Y_{i:} \right\|_{\mathbb{R}^h} \leq \sum_{i=1}^{n} \|X_{i:} - Y_{i:}\|_{\mathbb{R}^h} = \|X - Y\|_1.$$

  We highlight that this does not necessarily hold for other $\ell_p$ norms for $p \neq 1$.

- The map $\sigma$ is $L_\sigma$-Lipschitz transferable.

Thus, the result follows from Proposition 5.1; completing the proof. $\square$

**Normalized DeepSet.** The mean aggregation is not compatible with the zero-padding embedding. Consider a vector $x = (x_1, \ldots, x_n) \in \mathbb{R}^n$ such that $\sum_i x_i \neq 0$, and suppose $n < N$. When zero-padded to length $N$, we obtain

$$\tilde{x} = (x_1, \ldots, x_n, 0, \ldots, 0) \in \mathbb{R}^N.$$

Then

$$\frac{1}{N} \sum_{i=1}^N \tilde{x}_i = \frac{1}{N} \sum_{i=1}^n x_i \neq \frac{1}{n} \sum_{i=1}^n x_i.$$

Therefore, normalized DeepSet is not compatible with respect to the zero-padding consistent sequence in general.

We now prove its compatibility and transferability with respect to the duplication consistent sequence with normalized $\ell_p$ norm.

**Corollary F.4.** *Fix arbitrary norms $\|\cdot\|_{\mathbb{R}^d}$ on $\mathbb{R}^d$ and $\|\cdot\|_{\mathbb{R}^h}$ on $\mathbb{R}^h$. Let $\rho : \mathbb{R}^d \to \mathbb{R}^h$ be $L_\rho$-Lipschitz, and $\sigma : \mathbb{R}^h \to \mathbb{R}$ be $L_\sigma$-Lipschitz, with respect to the norms $\|\cdot\|_{\mathbb{R}^d}, \|\cdot\|_{\mathbb{R}^h}$, and $|\cdot|$. Then, for all $p \in [1, \infty]$, the sequence of maps $(\overline{\mathrm{DeepSet}}_n)$ is $(L_\rho L_\sigma)$-Lipschitz transferable with respect to the duplication consistent sequence $\mathbb{V}_{\mathrm{dup}}^{\oplus d}$ (equipped with the normalized $\ell_p$ norm induced by $\|\cdot\|_{\mathbb{R}^d}$) and the trivial consistent sequence $\mathbb{V}_{\mathbb{R}}$ (with absolute value norm). Therefore, $(\overline{\mathrm{DeepSet}}_n)$ extends to*

$$\overline{\mathrm{DeepSet}}_\infty : \mathcal{P}_p(\mathbb{R}^d) \to \mathbb{R}, \quad \overline{\mathrm{DeepSet}}_\infty(\mu) = \sigma\left(\int \rho \, d\mu\right),$$

*which is $(L_\rho L_\sigma)$-Lipschitz with respect to the Wasserstein-$p$ distance on $\|\cdot\|_{\mathbb{R}^d}$.*

*Proof.* We model each intermediate space with consistent sequences:

$$\mathbb{V}_{\mathrm{dup}}^{\oplus d} = (\mathbb{R}^{n \times d}) \xrightarrow[\text{(row-wise)}]{\rho^{\oplus n}} V_{\mathrm{dup}}^{\oplus h} = (\mathbb{R}^{n \times h}) \xrightarrow{\frac{1}{n} \sum_{i=1}^n} \mathbb{V}_{\mathbb{R}^h} = (\mathbb{R}^h) \xrightarrow{\sigma} \mathbb{V}_{\mathbb{R}} = (\mathbb{R}).$$

We first consider the compatibility of each map.

- The $\rho$-map is compatible because $\rho^{\oplus N}(X \otimes \mathbb{1}_{N/n}) = \rho^{\oplus n}(X) \otimes \mathbb{1}_{N/n}$ for all $n \mid N$, and the row-wise application makes sure $\rho$ is $\mathsf{S}_n$-equivariant.

- The mean aggregation $\mathrm{Agg}_{i=1}^n = \frac{1}{n} \sum_{i=1}^n$ is compatible because for all $n \mid N, X \in \mathbb{R}^{n \times d}$,

$$\frac{1}{N} \sum_{i=1}^N (X \otimes \mathbb{1}_{N/n})_{i:} = \frac{1}{N} \sum_{i=1}^n (N/n) X_{i:} = \frac{1}{n} \sum_{i=1}^n X_{i:},$$

  and the mean operation is $\mathsf{S}_n$-invariant.

- The map $\sigma$ is again automatically compatible.

Endow $\mathbb{V}_{\mathrm{dup}}^{\oplus d}$ with the normalized $\ell_p$ norm with respect to $\|\cdot\|_{\mathbb{R}^d}$, and $\mathbb{V}_{\mathrm{dup}}^{\oplus h}$ with the normalized $\ell_p$ norm with respect to $\|\cdot\|_{\mathbb{R}^h}$. The trivial consistent sequences $\mathbb{V}_{\mathbb{R}^h}, \mathbb{V}_{\mathbb{R}}$ are endowed with $\|\cdot\|_{\mathbb{R}^h}$ and $|\cdot|$ respectively. Next, we check the Lipschitz transferability of each map in the composition.

- The $\rho$-map is $L_\rho$-Lipschitz because for all $n$, we can prove $\rho^{\oplus n} : \mathbb{R}^{n \times d} \to \mathbb{R}^{n \times h}$ is $L_\rho$ Lipschitz with respect to the normalized $\ell_p$ norm:

$$\|\rho^{\oplus n}(X) - \rho^{\oplus n}(Y)\|_{\overline{p}} = \left(\frac{1}{n} \sum_{i=1}^n \|\rho(X_{i:}) - \rho(Y_{i:})\|_{\mathbb{R}^h}^p\right)^{1/p}$$

$$\leq \left(\frac{1}{n} \sum_{i=1}^n L_\rho^p \|X_{i:} - Y_{i:}\|_{\mathbb{R}^d}^p\right)^{1/p}$$

$$= L_\rho \|X - Y\|_{\overline{p}}.$$

- The mean aggregation $\text{Agg}_{i=1}^n = \frac{1}{n} \sum_{i=1}^n$ is 1-Lipschitz transferable because

$$\left\| \frac{1}{n} \sum_{i=1}^n X_{i:} - \frac{1}{n} \sum_{i=1}^n Y_{i:} \right\|_{\mathbb{R}^h} \leq \frac{1}{n} \sum_{i=1}^n \|X_{i:} - Y_{i:}\|_{\mathbb{R}^h} = \|X - Y\|_{\bar{1}} \leq \|X - Y\|_{\bar{p}},$$

where the last inequality follows from Hölder's inequality.

- The map $\sigma$ is $L_\sigma$-Lipschitz transferable.

The result follows from an application of Proposition 5.1; completing the proof. $\square$

**Remark F.5.** *The same result was also proved in [9, Theorem 3.7] by directly verifying the Lipschitz property of $\overline{\text{DeepSet}}_\infty$: for all $p \geq 1$,*

$$\left| \overline{\text{DeepSet}}_\infty(\mu_X) - \overline{\text{DeepSet}}_\infty(\mu_Y) \right| \leq L_\sigma \left| \int \rho \, d(\mu_X - \mu_Y) \right|$$

$$\leq L_\sigma L_\rho W_1(\mu_X, \mu_Y) \leq L_\sigma L_\rho W_p(\mu_X, \mu_Y),$$

*where the second inequality follows from the Kantorovich-Rubinstein duality. Our methods provide an alternative proof, using a proof technique that applies more generally (Proposition 5.1).*

Following this result, we can directly apply Propositions D.4 and D.5, along with the convergence rates described in Appendix D.3, which immediately yields the following transferability result.

**Corollary F.6** (Transferability of normalized DeepSet). *We have the following transferability results for normalized DeepSet:*

1. *(**Uniform grid sampling**) Let $(X_n) \in \mathbb{R}^{n \times d}$ be a sequence of matrices sampled from the same signal $f$ via the "uniform grid" sampling scheme, i.e., taking $(X_n)_{i:} = f\left(\frac{i-1}{n}\right)$ for all $i \in [n]$. Suppose $f$ is Lipschitz. Then,*

$$\left| \overline{\text{DeepSet}}_n(X_n) - \overline{\text{DeepSet}}_m(X_m) \right| \lesssim n^{-1} + m^{-1}.$$

2. *(**Random signal sampling**) Let $(X_n) \in \mathbb{R}^{n \times d}$ be a sequence of matrices sampled from the same signal $f$ via the random signal sampling scheme, i.e. taking $(X_n)_{i:} = f(x_i)$ for all $i \in [n]$, where $x_1, \ldots, x_n$ are sampled i.i.d. from $\text{Unif}([0,1])$. Suppose $f \in L^3([0,1], \mathbb{R}^d)$. Then,*

$$\mathbb{E} \left| \overline{\text{DeepSet}}_n(X_n) - \overline{\text{DeepSet}}_m(X_m) \right|$$

$$\lesssim \begin{cases} n^{-1/2} + m^{-1/2} & \text{if } d = 1, \\ n^{-1/2} \log(1+n) + m^{-1/2} \log(1+m) & \text{if } d = 2, \\ n^{-1/d} + m^{-1/d} & \text{if } d > 2. \end{cases}$$

3. *(**Empirical distributions**) Let $(X_n) \in \mathbb{R}^{n \times d}$ be a sequence of matrices sampled from the same underlying distribution $\mu \in \mathcal{P}_3(\mathbb{R}^d)$, i.e., each $X_n$ has rows sampled i.i.d. from $\mu$. Then,*

$$\mathbb{E} \left| \overline{\text{DeepSet}}_n(X_n) - \overline{\text{DeepSet}}_m(X_m) \right|$$

$$\lesssim \begin{cases} n^{-1/2} + m^{-1/2} & \text{if } d = 1, \\ n^{-1/2} \log(1+n) + m^{-1/2} \log(1+m) & \text{if } d = 2, \\ n^{-1/d} + m^{-1/d} & \text{if } d > 2. \end{cases}$$

**PointNet.** The max aggregation is not compatible with zero-padding. Consider a vector $x = (x_1, \ldots, x_n) \in \mathbb{R}^n$ where all entries $x_i < 0$, and suppose $n < N$. When zero-padded to length $N$, we obtain

$$\tilde{x} = (x_1, \ldots, x_n, 0, \ldots, 0) \in \mathbb{R}^N.$$

Then, we have

$$\max_{1 \leq i \leq N} \tilde{x}_i = 0 \neq \max_{1 \leq i \leq n} x_i.$$

Hence, unless we restrict the model to avoid all-negative entries, PointNet is not compatible with the zero-padding consistent sequence. We now prove its compatibility and transferability with respect to the duplication sequence with the $\ell_\infty$ norm.

**Corollary F.7.** *Fix arbitrary norms $\|\cdot\|_{\mathbb{R}^d}$ on $\mathbb{R}^d$ and $\|\cdot\|_\infty$ on $\mathbb{R}^h$. Let $\rho : \mathbb{R}^d \to \mathbb{R}^h$ be $L_\rho$-Lipschitz, and $\sigma : \mathbb{R}^h \to \mathbb{R}$ be $L_\sigma$-Lipschitz, with respect to the norms $\|\cdot\|_{\mathbb{R}^d}$, $\|\cdot\|_\infty$ on $\mathbb{R}^h$, and $|\cdot|$. Then, the sequence of maps $(\mathrm{PointNet}_n)$ is $(L_\rho L_\sigma)$-Lipschitz transferable with respect to the duplication consistent sequence $\mathbb{V}_{\mathrm{dup}}^{\oplus d}$ (equipped with the $\ell_\infty$-norm induced by $\|\cdot\|_{\mathbb{R}^d}$) and the trivial consistent sequence $\mathbb{V}_{\mathbb{R}}$ (with absolute value norm). Therefore, $(\mathrm{PointNet}_n)$ extends to*

$$\mathrm{PointNet}_\infty : \mathcal{P}_\infty(\mathbb{R}^d) \to \mathbb{R}, \quad \mathrm{PointNet}_\infty(\mu) = \sigma\left(\sup_{x\in\mathrm{supp}(\mu)} \rho(x)\right),$$

*which is $(L_\rho L_\sigma)$-Lipschitz with respect to the Wasserstein-$\infty$ distance on $\|\cdot\|_{\mathbb{R}^d}$.*

*Proof.* We again consider consistent sequences

$$\mathbb{V}_{\mathrm{dup}}^{\oplus d} = (\mathbb{R}^{n\times d}) \xrightarrow[\text{(row-wise)}]{\rho^{\oplus n}} V_{\mathrm{dup}}^{\oplus h} = (\mathbb{R}^{n\times h}) \xrightarrow{\max_{i=1}^n} \mathbb{V}_{\mathbb{R}^h} = (\mathbb{R}^h) \xrightarrow{\sigma} \mathbb{V}_{\mathbb{R}} = (\mathbb{R}).$$

For compatibility, it is left to check that $\mathrm{Agg}_{i=1}^n = \max_{i=1}^n$ is compatible. Indeed, for any $X \in \mathbb{R}^{n\times d}$, $n \mid N$, we have $\max_{i=1}^N (X \otimes \mathbb{1}_{N/n})_{i:} = \max_{i=1}^n X_{i:}$, and the $\max$ operation is $\mathsf{S}_n$-invariant. Endow $\mathbb{V}_{\mathrm{dup}}^{\oplus d}$ with the $\ell_\infty$ norm with respect to $\|\cdot\|_{\mathbb{R}^d}$, and $\mathbb{V}_{\mathrm{dup}}^{\oplus h}$ with the $\ell_\infty$ norm with respect to $\|\cdot\|_\infty$ on $\mathbb{R}^h$. The trivial consistent sequences $\mathbb{V}_{\mathbb{R}^h}, \mathbb{V}_{\mathbb{R}}$ are endowed with $\|\cdot\|_\infty$ and $|\cdot|$ respectively. Next, we check Lipschitz transferability of each map.

- We have proved in the proof for normalized DeepSet that $\rho, \sigma$ are $L_\rho, L_\sigma$ Lipschitz transferable respectively.

- For any $j \in [d]$, $|\max_{i=1}^n X_{ij} - \max_{i=1}^n Y_{ij}| \leq \max_{i=1}^n |X_{ij} - Y_{ij}|$. Take max over $j \in [d]$, we conclude

$$\left\|\max_{i=1}^n X_{i:} - \max_{i=1}^n Y_{i:}\right\|_\infty = \max_{j=1}^d \left|\max_{i=1}^n X_{ij} - \max_{i=1}^n Y_{ij}\right| \leq \max_{i=1}^n \|X_{i:} - Y_{i:}\|_\infty.$$

  Hence, $\mathrm{Agg}_{i=1}^n = \max_{i=1}^n$ is 1-Lipschitz transferable.

The result follows from Proposition 5.1; which completes the proof. $\square$

**Remark F.8.** $\mathrm{PointNet}_\infty$ *produces identical outputs for probability measures with the same support. Thus, it can be viewed as a function*

$$\mathrm{PointNet}_\infty : \mathcal{K}(\mathbb{R}^d) \to \mathbb{R}, \quad \mathrm{PointNet}_\infty(X) = \sigma\left(\sup_{x\in X} \rho(x)\right),$$

*where $\mathcal{K}(\mathbb{R}^d)$ denotes the set of non-empty compact subsets of $\mathbb{R}^d$. The $W_\infty$ distance on $\mathcal{P}_\infty(\mathbb{R}^d)$ induces the quotient metric $d_{\mathcal{K}}$ on $\mathcal{K}(\mathbb{R}^d)$ via the equivalence relation $\mu \sim \nu$ if $\mathrm{supp}(\mu) = \mathrm{supp}(\nu)$. Our results imply that $\mathrm{PointNet}_\infty$ is $(L_\rho L_\sigma)$-Lipschitz with respect to $d_{\mathcal{K}}$.*

*A more commonly used metric on $\mathcal{K}(\mathbb{R}^d)$ is the Hausdorff distance, defined by*

$$d_H(X, Y) := \max\left\{\sup_{x\in X}\inf_{y\in Y}\|x-y\|, \sup_{y\in Y}\inf_{x\in X}\|x-y\|\right\}. \tag{11}$$

*[9, Theorem 3.7] shows that $\mathrm{PointNet}_\infty$ is $(2L_\rho L_\sigma)$-Lipschitz with respect to $d_H$. It is easy to see that $d_H \leq d_{\mathcal{K}}$, but we leave exploring further relations between these two metrics to future work.*

Finally, we show that the sequence of maps $(\mathrm{PointNet}_n)$ is, in general, not transferable with respect to the duplication-based consistent sequence $\mathbb{V}_{\mathrm{dup}}^{\oplus d}$ when equipped with the normalized $\ell_p$ norm for any $p \in [1, \infty)$. Consider the sequence of matrices $(X^{(n)} \in \mathbb{R}^{n\times h})_n$ where the first row is the all-one vector $\mathbb{1}_h^\top$, and the remaining $n-1$ rows are zero vectors. Then,

$$\|X^{(n)} - 0\|_{\bar{p}} \to 0 \quad \text{as } n \to \infty,$$

which implies that $[X^{(n)}] \to [0]$ in $V_\infty$. However,

$$\max_{i=1}^n X_{i:}^{(n)} = \mathbb{1}_h \quad \text{for all } n.$$

This demonstrates that the max aggregation $\mathrm{Agg}_{i=1}^n = \max_{i=1}^n$ is not continuously transferable under the normalized $\ell_p$ norm, and so neither is the sequence $(\mathrm{PointNet}_n)$.

**Related work.** Several previous works [92, 64, 9] studied (variants of) normalized DeepSets in the infinite-dimensional limit as operating in the space of probability measures, and investigated their approximation power and generalization behavior.

Moreover, while our analysis primarily focused on *invariant* neural networks on sets, it is also natural to design and analyze *equivariant* neural networks on sets with our theoretical framework. In particular, we may consider such neural networks that are transferable with respect to duplication-consistent sequences, which parametrize measure-to-measure functions. For example, [19] proposed a general framework for designing measure-to-measure neural network architectures. Additionally, [86, 30, 74] analyzed transformers (without causal masking or positional encoding) as measure-to-measure functions in the limit space. We leave the analysis of the transferability of these models within our framework as a future direction.

### F.2.2 Explanation of transferability plots (Figure 2)

Our numerical experiment in Figure 2 illustrates the second column of Table 2 for $p = 1$: the Lipschitz transferability of normalized DeepSet, and non-transferability of DeepSet and PointNet, with respect to $(\mathbb{V}_{\mathrm{dup}}, \|\cdot\|_{\overline{1}})$. First, applying Proposition D.5 to normalized DeepSet, and recalling the convergence rate of empirical distributions given in (3) for $d = 1, p = 1$, we get the following corollary.

**Corollary F.9** (Convergence and transferability of normalized DeepSet). *Let $(x_n \in \mathbb{R}^n)_{n \in \mathbb{N}}$ be a sequence of inputs with entries $(x_n)_i \overset{i.i.d.}{\sim} \mu$ for $i = 1, \ldots, n$, where $\mu$ is a probability measure on $\mathbb{R}$ with finite expectation. Define the empirical measure $\mu_x := \frac{1}{n} \sum_{i=1}^n \delta_{x_i}$ for $x \in \mathbb{R}^n$. Then $\mathbb{E}[W_1(\mu, \mu_{x_n})] \lesssim n^{-1/2}$, and hence*

$$\mathbb{E}\left[\left|\overline{\mathrm{DeepSet}}_n(x_n) - \overline{\mathrm{DeepSet}}_\infty(\mu)\right|\right] \lesssim n^{-1/2},$$
$$\mathbb{E}\left[\left|\overline{\mathrm{DeepSet}}_n(x_n) - \overline{\mathrm{DeepSet}}_m(x_m)\right|\right] \lesssim n^{-1/2} + m^{-1/2}.$$

Indeed, Figure 2(b) shows convergence of the model outputs, and Figure 2(d) confirms that the convergence rate is $O(n^{-1/2})$, as predicted. Figure 2(a) illustrates the divergence of outputs from DeepSet. This occurs because the sum $\sum_{i=1}^n \rho(x_i) = O(n)$. If the function $\sigma$ in DeepSet is unbounded, this leads to unbounded (blow-up) outputs as $n$ increases. Figure 2(c) shows divergent outputs from PointNet. When the input distribution $\mu$ has compact support, the output of PointNet will converge, although without guarantees on the rate. However, in our experiment where $\mu = \mathcal{N}(0, 1)$ has non-compact support. If $\rho$ in the PointNet is unbounded, the maximum value $\max_{i=1}^n \rho(x_i)$ diverges almost surely as $n \to \infty$. This again results in blow-up outputs.

## G Example 2 (graphs): details and missing proofs from Section 5.2

In this section, we study transferability for GNN architectures. Section G.1 introduces the consistent sequences we consider. Section G.2 studies existing architectures and Section G.3 introduces a new class of GNNs with better transferability properties.

### G.1 Duplication consistent sequence for graphs

We present an examples of consistent sequences on graphs, illustrated graphically in Figure 3. Start with the duplication consistent sequence for sets $\mathbb{V}_{\mathrm{dup}}$ defined in F.1, we define

$$\mathbb{V}_{\mathrm{dup}}^G := \mathrm{Sym}^2(\mathbb{V}_{\mathrm{dup}}) \oplus \mathbb{V}_{\mathrm{dup}}^{\oplus d},$$

following the definition of direct sum and tensor product in Definition C.4, C.5. This gives the duplication consistent sequence for graphs. Specifically, $\mathbb{V}_{\mathrm{dup}}^G = \{(V_n), (\varphi_{N,n}), (\mathsf{G}_n)\}$ where $V_n = \mathbb{R}_{\mathrm{sym}}^{n \times n} \times \mathbb{R}^{n \times d}$ for each $n$ and the embedding for $n \mid N$ is given by,

$$\varphi_{N,n} \colon \mathbb{R}_{\mathrm{sym}}^{n \times n} \times \mathbb{R}^{n \times d} \hookrightarrow \mathbb{R}_{\mathrm{sym}}^{N \times N} \times \mathbb{R}^{N \times d}$$
$$(A, X) \mapsto (A \otimes (\mathbb{1}_{N/n} \mathbb{1}_{N/n}^\top), X \otimes \mathbb{1}_{N/n}),$$

which can be interpreted as replacing each node in the graph with $N/n$ duplicated copies. The symmetric group $\mathsf{S}_n$ acts on $V_n$ by $g \cdot (A, X) = (gAg^\top, gX)$.

The space $V_\infty$ can be identified with the space with all step graphons (and signals) in a similar way: Given $(A, X) \in \mathbb{R}^{n \times n}_{\text{sym}} \times \mathbb{R}^{n \times d}$, define $W_A : [0,1]^2 \to \mathbb{R}$, $f_X : [0,1] \to \mathbb{R}^d$ such that $W_A$ takes value $A_{i,j}$ on the interval $I_{i,n} \times I_{j,n} \subset [0,1]^2$ for $i, j = 1, \ldots, n \in [n]$, and $f_X$ takes value $X_i$ on the interval $I_{i,n} \subset [0,1]$ for $i = 1, \ldots, n$ . We call $(W_A, f_X)$ the *induced step graphon* from $(A, X)$. Under this identification, permutations $\mathsf{S}_n$ permute the $n$ intervals $I_{i,n}$ and act on $(W, f)$ by $\sigma \cdot (W, f) = (\sigma \cdot W, \sigma \cdot f) = (W^{\sigma^{-1}}, f \circ \sigma^{-1})$, where $W^{\sigma^{-1}}$ is defined by $W^{\sigma^{-1}}(x, y) := W(\sigma^{-1}(x), \sigma^{-1}(y))$. The limit group $\mathsf{G}_\infty$ is the union of such interval permutations.

**The $p$-norm on $\mathbb{V}^G_{\text{dup}}$.** Fix $\|\cdot\|_{\mathbb{R}^d}$ a norm on $\mathbb{R}^d$. We equip $V_n$ with a $p$-norm given by

$$
\|(A, X)\|_{\overline{p}} := \max(\|A\|_{\overline{p}}, \|X\|_{\overline{p}})
$$
$$
= \begin{cases} \max\left( \left( \frac{1}{n^2} \sum_{i,j \in [n]} |A_{ij}|^p \right)^{1/p}, \left( \frac{1}{n} \sum_{i=1}^n \|X_{i:}\|^p_{\mathbb{R}^d} \right)^{1/p} \right) & p \in [1, \infty), \\ \max\left( \max_{i,j \in [n]} |A_{ij}|, \max_{i=1}^n \|X_{i:}\|_{\mathbb{R}^d} \right) & p = \infty. \end{cases} \tag{12}
$$

It is easy to check by Proposition C.8 that this extends to a norm on $V_\infty$. Under the identification with step graphons, this norm on $V_\infty$ coincides with the standard $L^p$-norm given by

$$
\|(W, f)\|_p := \max(\|W\|_p, \|f\|_p)
$$
$$
= \begin{cases} \max\left( \left( \iint |W(x, y)|^p\, dx\, dy \right)^{1/p}, \left( \int |f(x)|^p\, dx \right)^{1/p} \right) & p \in [1, \infty), \\ \max\left( \sup_{x,y} |W(x, y)|, \sup_x |f(x)| \right) & p = \infty. \end{cases}
$$

That is, for any $(A, X) \in \mathbb{R}^{n \times n}_{\text{sym}} \times \mathbb{R}^{n \times d}$, $\|(A, X)\|_{\overline{p}} = \|(W_A, f_X)\|_p$. The symmetrized metric is

$$
\overline{\mathrm{d}}_p((W, f), (W', f')) = \inf_{\sigma \in \mathsf{G}_\infty} \{ \max(\|W - \sigma \cdot W'\|_p, \|f - \sigma \cdot f'\|_p) \}. \tag{13}
$$

**Operator $p$-norm on $\mathbb{V}^G_{\text{dup}}$.** Fix $\|\cdot\|_{\mathbb{R}^d}$ a norm on $\mathbb{R}^d$ and $p \in [1, \infty]$. We equip $V_n$ with a norm given by

$$
\|(A, X)\|_{\text{op},p} := \max\left( \frac{1}{n} \|A\|_{\text{op},p}, \|X\|_{\overline{p}} \right), \tag{14}
$$

where $\|A\|_{\text{op},p}$ is the operator norm of $A$ with respect to the $\ell_p$ norm, i.e.

$$
\|A\|_{\text{op},p} = \max_{\|x\|_p = 1} \|Ax\|_p,
$$

and $\|X\|_{\overline{p}}$ is the normalized $\ell_p$-norms with respect to $\|\cdot\|_{\mathbb{R}^d}$ defined in (10). It is easy to check by Proposition C.8 that this extends to a norm on $V_\infty$.

Let $T_W$ be the shift operator of graphon $W$ defined by $T_W(f)(u) := \int_0^1 W(u, v) f(v) dv$. Under the identification with step graphons, the norm on $V_\infty$ coincides with

$$
\|(W, f)\|_{\text{op},p} := \max(\|T_W\|_{\text{op},p}, \|f\|_p),
$$

where $\|T_W\|_{\text{op},p} := \sup_{\|f\|_p = 1} \|T_W(f)\|_p$, and the norm on $\|f\|_p$ is the $L^p$ norm as before. That is, for any $(A, X) \in \mathbb{R}^{n \times n}_{\text{sym}} \times \mathbb{R}^{n \times d}$, $\|(A, X)\|_{\text{op},p} = \|(W_A, f_X)\|_{\text{op},p}$. For $p \in [1, \infty)$, the completion with respect to this metric is

$$
\overline{V_\infty} = \{ W \in L^p([0,1]^2) : W(x, y) = W(y, x) \} \times \{ f \in L^p([0,1], \mathbb{R}^d) \},
$$

the space of $L^p$-graphon signals. We do not characterize $\overline{V_\infty}$ for $p = \infty$ since it requires additional technicalities. However, it contains all bounded and continuous graphon signals. The symmetrized metric is

$$
\overline{\mathrm{d}}_p((W, f), (W', f')) = \inf_{\sigma \in \mathsf{G}_\infty} \{ \max(\|T_W - T_{\sigma \cdot W'}\|_{\text{op},p}, \|f - \sigma \cdot f'\|_p) \}. \tag{15}
$$

**Cut norm on $\mathbb{V}^G_{\text{dup}}$.** We can also equip $V_n$ with the *cut norm* (on matrices and vectors),

$$
\|(A, X)\|_\square := \max(\|A\|_\square, \|X\|_\square) = \max\left( \frac{1}{n^2} \max_{S,T \subseteq [n]} \left| \sum_{i \in S, j \in T} A_{ij} \right|, \frac{1}{n} \max_{S \subseteq [n]} \left\| \sum_{i \in S} X_{i:} \right\|_{\mathbb{R}^d} \right). \tag{16}
$$

It is easy to check by Proposition C.8 that this extends to a norm on $V_\infty$. Under the identification with step graphons, this norm on $V_\infty$ coincides with the cut norm on graphons and graphon signals

$$\|(W, f)\|_\square := \max\left(\|W\|_\square, \|f\|_\square\right)$$

$$= \max\left(\sup_{S,T\subseteq[0,1]} \left|\int_{S\times T} W(x,y)dxdy\right|, \sup_{S\subseteq[0,1]} \left\|\int_S f(x)dx\right\|_{\mathbb{R}^d}\right),$$

where the supremum is taken over all measurable sets $S, T$. The cut norm on graphon is studied in-depth in [54]. Though hard to compute, it has strong combinatorial interpretations. Hence, it has played an important role in the work of GNN transferability, and has been extended to the graphon signals in [47], which we have adopted here.

The symmetrized metric is

$$\bar{\mathrm{d}}((W, f), (W', f')) := \inf_{\sigma\in\mathsf{G}_\infty} \{\max\left(\|W - \sigma\cdot W'\|_\square, \|f - \sigma\cdot f'\|_\square\right)\}.$$

It can be proved that this exactly coincides with the *cut distance* below, defined on graphon signals (extending the original definition of graphon *cut distance* from [54], similarly to the version in [47]):

$$\bar{\mathrm{d}}((W, f), (W', f')) = \inf_{\sigma\in S_{[0,1]}} \{\max\left(\|W - \sigma\cdot W'\|_\square, \|f - \sigma\cdot f'\|_\square\right)\}, \tag{17}$$

where $S_{[0,1]}$ is the group of measure-preserving bijections $\sigma : [0,1] \to [0,1]$ with measurable inverse. The proof follows analogously to [8, Lemma 3.5]. Specifically, we first verify that the definitions agree on step graphons. Then, since both definitions are continuous with respect to the cut norm $\|\cdot\|_\square$, they must also agree on the limit points.

While the cut norm is hard to work with directly, it is topologically equivalent to the operator 2-norms considered previously on a bounded domain. This means that any function continuous with respect to one of these norms is also continuous with respect to the other.

**Proposition G.1.** *If $\|W\|_\infty < r$ and $\|f\|_\infty < r$, then*

$$\|(W, f)\|_\square \leq \|(W, f)\|_{\mathrm{op},2} \lesssim \|(W, f)\|_\square^{1/2}.$$

*Consequently, for $p \in (1, \infty)$, $\|\cdot\|_\square$ and $\|\cdot\|_p$ are topologically equivalent on the space*

$$\left\{W : [0,1]^2 \to [-r, r] \text{ measurable}, \ W(x,y) = W(y,x)\right\} \times \left\{f : [0,1] \to [-r, r] \text{ measurable}\right\}.$$

*Proof.* Without loss of generality, let $r = 1$. Consider the norm

$$\|T_W\|_{\mathrm{op},\infty,1} := \sup_{\|f\|_\infty, \|g\|_\infty \leq 1} \left|\int_0^1 \int_0^1 W(u,v)f(u)g(v)dudv\right|.$$

By [37, Equation 4.4], $\|W\|_\square \leq \|T_W\|_{\mathrm{op},\infty,1} \leq 4\|W\|_\square$. By [37, Lemma E.6], if $\|W\|_\infty \leq 1$,

$$\|T_W\|_{\mathrm{op},\infty,1} \leq \|T_W\|_{\mathrm{op},2} \leq \sqrt{2}\|T_W\|_{\mathrm{op},\infty,1}^{1/2}.$$

Combining the inequalities,

$$\|W\|_\square \leq \|T_W\|_{\mathrm{op},2} \leq 2^{3/2}\|W\|_\square^{1/2}.$$

Moreover, by [47, Appendix A.2], $\|f\|_\square \leq \|f\|_1 \leq 2\|f\|_\square$. If $\|f\|_\infty \leq 1$, then $\|f\|_2^2 \leq \|f\|_1 \leq \|f\|_2$. Combining the inequalities,

$$\|f\|_\square \leq \|f\|_2 \leq 2^{1/2}\|f\|_\square^{1/2}.$$

Therefore, we conclude that

$$\|(W, f)\|_\square \leq \|(W, f)\|_{\mathrm{op},p} \leq 2^{3/2}\|(W, f)\|_\square^{1/2},$$

as claimed. $\qquad\square$

## G.2   Message Passing Neural Networks (MPNNs)

**Background.** MPNN parametrizes a sequence of functions $(\text{MPNN}_n \colon \mathbb{R}^{n \times n}_{\text{sym}} \times \mathbb{R}^{n \times d_1} \to \mathbb{R}^{n \times d_L})$ by composition of message passing layers. The $l$-th message passing layer

$$\text{MP}^{(l)}_n \colon \mathbb{R}^{n \times n}_{\text{sym}} \times \mathbb{R}^{n \times d_l} \to \mathbb{R}^{n \times n}_{\text{sym}} \times \mathbb{R}^{n \times d_{l+1}}, \quad (A, X^{(l)}) \mapsto (A, X^{(l+1)})$$

is given by

$$X^{(l+1)}_{i:} = \phi^{(l)}\left(X^{(l)}_{i:}, \text{Agg}_{j \in \mathcal{N}_i} \psi^{(l)}\left(X^{(l)}_{i:}, X^{(l)}_{j:}, A_{ij}\right)\right), \quad i = 1, \dots, n, \tag{18}$$

where $\text{Agg}$ is a permutation-invariant aggregation function such as sum, mean, or max; $\mathcal{N}_i := \{j : A_{ij} \neq 0\}$ denotes the neighborhood of node $i$ in the input graph; the message function $\psi^{(l)} \colon \mathbb{R}^{d_l} \times \mathbb{R}^{d_l} \times \mathbb{R} \to \mathbb{R}^{h_l}$ and the update function $\phi^{(l)} \colon \mathbb{R}^{d_l} \times \mathbb{R}^{h_l} \to \mathbb{R}^{d_{l+1}}$ are independent of the graph size $n$. Composing $L$ message-passing layers defines an MPNN, mapping $(A, X^{(1)}) \mapsto X^{(L)}$.

Observe that MPNN is permutation-equivariant: $\text{MPNN}_n(gAg^\top, gX) = g\text{MPNN}_n(A, X)$ for all $g \in \mathsf{S}_n$. If we want a permutation-invariant function, this is followed by a read-out operation taking the form of DeepSet. In this work, we focus on the equivariant case.

MPNN is a general framework for GNNs based on local message passing: [31] formulates multiple GNNs as MPNNs with specific choices of $\phi, \psi, \text{Agg}$; other state-of-the-art GNNs can be simulated by MPNN on a transformed graph [38]. Moreover, $\phi$ and $\psi$ can also be parameterized with MLPs to provide good flexibility.

**Transferability analysis of MPNNs.** The following corollary gives sufficient conditions on the layers of MPNNs to obtain Lipschitz transferability.

**Corollary G.2.** *Consider one message passing layer,* $(\text{MP}^{(l)}_n)$, *as defined in* (18), *with the following properties.*

1. *The message function $\psi^{(l)}$ takes the form $\psi^{(l)}(x_1, x_2, w) := w\xi(x_2)$, where $\xi \colon \mathbb{R}^{d_l} \to \mathbb{R}^{h_l}$ is $L_\xi$ Lipschitz with respect to $\|\cdot\|_{\mathbb{R}^{d_l}}, \|\cdot\|_{\mathbb{R}^{h_l}}$.*

2. *The aggregation used is the normalized sum aggregation $\text{Agg}_{j \in \mathcal{N}_i} := \frac{1}{n}\sum_{j \in \mathcal{N}_i}$.*

3. *The update function $\phi^{(l)}$ is $L_\phi$ Lipschitz, i.e. for all $(x, y), (x', y') \in \mathbb{R}^{d_l} \times \mathbb{R}^{h_l}$,*

$$\|\phi^{(l)}(x, y) - \phi^{(l)}(x', y')\|_{\mathbb{R}^{d_{l+1}}} \leq L_\phi \max\left(\|x - x'\|_{\mathbb{R}^{d_l}}, \|y - y'\|_{\mathbb{R}^{h_l}}\right)$$

*Endow the space of duplication-consistent sequences with the operator $p$-norm as defined in* (14), *where $p \in [1, \infty)$. Then, the sequence of maps $(\text{MP}^{(l)}_n)$ is locally Lipschitz transferable.*

**Remark G.3.** *That is,* $(\text{MPNN}_n)$, *which is a composition of message-passing layers, is locally Lipschitz transferable. Consequently it extends to a function $\text{MPNN}_\infty$ on the space of graphon signals, which is $L(r)$-Lipschitz on $B(0, r)$ for all $r > 0$ with respect to the symmetrized operator $p$-metric defined in* (15)). *By Proposition G.1, the sequence of maps $(\text{MPNN}_n)$ is continuously transferable with respect to the cut norm* (16) *on $B(0, r)$. The GNN studied in [70] is a special case of our MPNN considered here; meanwhile, ours is a special case of [47], which directly establishes Lipschitzness with respect to the cut distance by analysis on the graphon space. While our results are not new, our proof technique—following Proposition 5.1—is new and generally applicable to various models.*

*Proof.* We decompose $\text{MP}^{(l)}_n$ as a composition of the following maps, modelling each of the intermediate spaces using the duplication consistent sequences endowed with compatible norms. For the metric on the product spaces, we always use the $L^\infty$ product metric, i.e., taking the maximum over

the individual components. Consider the following three functions

$$f_n^{(1)} \colon \mathbb{R}^{n \times n}_{\text{sym}} \times \mathbb{R}^{n \times d_l} \to \mathbb{R}^{n \times n}_{\text{sym}} \times \mathbb{R}^{n \times d_l} \times \mathbb{R}^{n \times h_l}$$

$$\text{given by} \quad (A, X) \mapsto \left( A, X, \begin{bmatrix} \xi(X_{1:}) \\ \vdots \\ \xi(X_{n:}) \end{bmatrix} \right),$$

$$f_n^{(2)} \colon \mathbb{R}^{n \times n}_{\text{sym}} \times \mathbb{R}^{n \times d_l} \times \mathbb{R}^{n \times h_l} \to \mathbb{R}^{n \times n}_{\text{sym}} \times \mathbb{R}^{n \times d_l} \times \mathbb{R}^{n \times h_l}$$

$$\text{given by} \quad (A, X, Y_0) \mapsto (A, X, \frac{1}{n} A Y_0),$$

$$f_n^{(3)} \colon \mathbb{R}^{n \times n}_{\text{sym}} \times \mathbb{R}^{n \times d_l} \times \mathbb{R}^{n \times h_l} \to \mathbb{R}^{n \times n}_{\text{sym}} \times \mathbb{R}^{n \times d_{l+1}}$$

$$\text{given by} \quad (A, X, Y) \mapsto (A, \tilde{X}),$$

where $\tilde{X}_{i:} = \phi^{(l)}(X_{i:}, Y_{i:})$. It is straightforward to check that each of them is compatible with respect to the duplication embedding. We now check the Lipschitz transferability.

- The sequence $(f_n^{(1)})$ is Lipschitz transferable because

$$\left\| f_n^{(1)}(A, X) - f_n^{(1)}(A', X') \right\|_{\text{op},p} \le (1 \vee L_\xi) \left\| (A, X) - (A', X') \right\|_{\text{op},p}.$$

- The sequence $(f_n^{(2)})$ is locally Lipschitz transferable because we can bound its Jacobian. In particular, the Jacobian acts on $(H_A, H_X, H_Y)$ via

$$Df_n^{(2)}(A, X, Y_0)[H_A, H_X, H_Y] = \left( H_A, H_X, \frac{1}{n}(AH_Y + H_A Y) \right).$$

Hence, on $\{(A, X, Y_0) : \|(A, X, Y_0)\|_{\text{op},p} < r\}$,

$$\left\| Df_n^{(2)}(A, X, Y_0)[H_A, H_X, H_Y] \right\|_{\text{op},p}$$

$$\le \max \left( \frac{1}{n} \|H_A\|_{\text{op},p}, \|H_X\|_{\overline{p}}, \frac{1}{n} \|A\|_{\text{op},p} \|H_Y\|_{\overline{p}} + \frac{1}{n} \|H_A\|_{\text{op},p} \|Y\|_{\overline{p}} \right)$$

$$\le (1 \vee 2r) \|(H_A, H_X, H_Y)\|_{\text{op},p},$$

i.e., $f_n^{(2)}$ is $(1 \vee 2r)$ Lipschitz on this space.

- The sequence $(f_n^{(3)})$ is Lipschitz transferable because

$$\|f_n^{(3)}(A, X, Y) - f_n^{(3)}(A', X', Y')\|_{\overline{p}} = \max \left( \frac{1}{n} \|A - A'\|_{\text{op},p}, \|\tilde{X} - \tilde{X}'\|_{\overline{p}} \right)$$

where

$$\|\tilde{X} - \tilde{X}'\|_{\overline{p}}^p \le \frac{1}{n} \sum_{i=1}^n \|\phi^{(l)}(X_{i:}, Y_{i:}) - \phi^{(l)}(X'_{i:}, Y'_{i:})\|_{\mathbb{R}^{d_{l+1}}}^p$$

$$\le L_\phi^p \frac{1}{n} \sum_{i=1}^n \left( \|X_{i:} - X'_{i:}\|_{\mathbb{R}^{d_l}} + \|Y_{i:} - Y'_{i:}\|_{\mathbb{R}^{h_l}} \right)^p$$

$$\le L_\phi^p 2^{p-1} \frac{1}{n} \sum_{i=1}^n \left( \|X_{i:} - X'_{i:}\|_{\mathbb{R}^{d_l}}^p + \|Y_{i:} - Y'_{i:}\|_{\mathbb{R}^{h_l}}^p \right)$$

$$\text{(by Jensen's inequality } \left( \frac{a+b}{2} \right)^p \le \frac{a^p + b^p}{2} \text{ for all } a, b)$$

$$\le L_\phi^p 2^p \|(X, Y) - (X', Y')\|_{\overline{p}}^p.$$

So $f_n^{(3)}$ is $(1 \vee 2L_\phi)$-Lipschitz.

Finally, apply Proposition 5.1, $(\text{MP}_n^{(l)})$ is locally Lipschitz transferable; completing the proof. $\quad\square$

Following this result, we can directly apply Propositions D.4 and D.5, along with the convergence rates described in Appendix D.3, which immediately yields the following transferability result.

**Corollary G.4** (Transferability of MPNN)**.** *For MPNNs satisfying the assumptions in Corollary G.2, we have the following transferability results.*

1. *(**Uniform grid sampling**) Let $(A_n, X_n) \in \mathbb{R}^{n \times n}_{\mathrm{sym}} \times \mathbb{R}^{n \times d}$ be a sequence of graph signals sampled from the same graphon signal $(W, f)$ via the "uniform grid" sampling scheme, i.e., taking $(A_n)_{ij} = W\left(\frac{i-1}{n}, \frac{j-1}{n}\right), (X_n)_{i:} = f\left(\frac{i-1}{n}\right)$ for all $i, j \in [n]$. Suppose $W : [0,1]^2 \to [0,1]$ and $f : [0,1] \to \mathbb{R}^d$ are both Lipschitz. Then,*

$$\|[\mathrm{MPNN}_n(A_n, X_n)] - [\mathrm{MPNN}_m(A_m, X_m)]\|_2 \lesssim n^{-1} + m^{-1}.$$

2. *(**Graphon sampling**) Let $(A_n, X_n) \in \mathbb{R}^{n \times n}_{\mathrm{sym}} \times \mathbb{R}^{n \times d}$ be a sequence of graph signals sampled from the same graphon signal $(W, f)$ via the "graphon" sampling scheme, i.e. taking $(A_n)_{ij} \sim \mathrm{Ber}(W(x_i, x_j)), (X_n)_{i:} = f(x_i)$ for all $i, j \in [n]$, where $x_1, \ldots, x_n$ are sampled i.i.d. from $\mathrm{Unif}([0,1])$. Suppose $W : [0,1]^2 \to [0,1]$ is symmetric and measurable, and $f \in L^\infty([0,1], \mathbb{R}^d)$. Then,*

$$\mathbb{E}\left[\overline{\mathrm{d}}_2\left([\mathrm{MPNN}_n(A_n, X_n)], [\mathrm{MPNN}_m(A_m, X_m)]\right)\right] \lesssim (\log n)^{-1/4} + (\log m)^{-1/4}.$$

The first part of the previous corollary recovers the transferability results in [70], yielding an improved convergence rate of $O(n^{-1})$ and thus strengthening the previously established bounds of $O(n^{-1/2})$. The second part of the corollary resembles the setting considered in [41], although the random sampling scheme used there operates at a different sparsity level, with $A_{ij} \sim \mathrm{Ber}(\alpha_n W(x_i, x_j))$ and $\alpha_n \sim \frac{\log n}{n}$. As a result, our result is not directly comparable.

### G.3  Constructing new transferable GNNs: GGNN and continuous GGNN

**Background: Invariant Graph Networks (IGN).** Invariant Graph Networks (IGN) [56] are a class of GNN architectures that alternate between linear $\mathsf{S}_n$-equivariant layers and nonlinearities. They follow a design paradigm that differs fundamentally from MPNNs. Specifically, a $D$-layer 2-IGN parameterizes an $\mathsf{S}_n$-equivariant function $(\mathbb{R}^n)^{\otimes 2} \to (\mathbb{R}^n)^{\otimes 2}$ as a composition

$$W_n^{(D)} \circ \rho_n^{(D-1)} \circ \cdots \circ \rho_n^{(1)} \circ W_n^{(1)},$$

where for each $i$ we have the following.

- The linear maps $W_n^{(i)} : ((\mathbb{R}^n)^{\otimes 2})^{\oplus d_i} \cong \mathbb{R}^{n^2 \times d_i} \to ((\mathbb{R}^n)^{\otimes 2})^{\oplus d_{i+1}} \cong \mathbb{R}^{n^2 \times d_{i+1}}$ arer $\mathsf{S}_n$-equivariant. Here, $d_i$ denotes the number of feature channels. [56] provides a parameterization of $W_n^{(i)}$ as a linear combination of basis maps: In the special case where $d_i = d_{i+1} = 1$, the linear layer $W_n^{(i)}$ can be written as a linear combination of 17 basis functions (two of them are biases), where the coefficients $\alpha, \beta$ are the learnable parameters:

$$\begin{aligned} W_n^{(i)}(A) = {} & \alpha_1 A + \alpha_2 A^\top + \alpha_3 \mathrm{diag}(\mathrm{diag}^*(A)) + \alpha_4 A \mathbb{1}\mathbb{1}^\top + \alpha_5 \mathbb{1}\mathbb{1}^\top A + \alpha_6 \mathrm{diag}(A\mathbb{1}) \\ & + \alpha_7 A^\top \mathbb{1}\mathbb{1}^\top + \alpha_8 \mathbb{1}\mathbb{1}^\top A^\top + \alpha_9 \mathrm{diag}(A^\top \mathbb{1}) + \alpha_{10}(\mathbb{1}^\top A\mathbb{1})\mathbb{1}\mathbb{1}^\top \\ & + \alpha_{11}(\mathbb{1}^\top A\mathbb{1})\mathrm{diag}(\mathbb{1}) + \alpha_{12}(\mathbb{1}^\top \mathrm{diag}^*(A))\mathbb{1}\mathbb{1}^\top + \alpha_{13}(\mathbb{1}^\top \mathrm{diag}^*(A))\mathrm{diag}(\mathbb{1}) \\ & + \alpha_{14}\mathrm{diag}^*(A)\mathbb{1}^\top + \alpha_{15}\mathbb{1}\mathrm{diag}^*(A)^\top + \beta_1 \mathbb{1}\mathbb{1}^\top + \beta_2 \mathrm{diag}(\mathbb{1}). \end{aligned}$$

(19)

For general $d_i, d_{i+1}$, the number of basis terms becomes $17 d_i d_{i+1}$.

- The activations $\rho_n^{(i)} : ((\mathbb{R}^n)^{\otimes 2})^{\oplus d_{i+1}} \cong \mathbb{R}^{n^2 \times d_{i+1}} \to ((\mathbb{R}^n)^{\otimes 2})^{\oplus d_{i+1}} \cong \mathbb{R}^{n^2 \times d_{i+1}}$ apply a nonlinearity (e.g., ReLU) entry-wise.

To improve expressivity, [56] proposed extending the architecture to use higher-order tensors in the intermediate layers. When the maximum tensor order is $k$, the architecture is referred to as a $k$-IGN. While this is theoretically tractable, due to the high memory cost and implementation challenges

associated with higher-order tensors, in practice, only $k$-IGNs for $k \leq 2$ have been implemented to the best of our knowledge. In this work, we focus exclusively on 2-IGNs.

The basis in (19) is inherently dimension-agnostic, allowing IGN to serve as an any-dimensional neural network that parameterizes functions on inputs of arbitrary size $n$ using a fixed set of parameters. This feature fundamentally relies on representation stability, which is discussed in greater detail in [50].

**Incompatibility of IGN.** 2-IGN is incompatible with the subspace $\mathbb{V}_{\mathrm{dup}}^G$. First, its basis functions are not properly normalized, and therefore cannot be extended to functions on graphons. For instance, the fourth basis function $\ell_4(A) = A\mathbb{1}\mathbb{1}^\top$ yields output entries of order $O(n)$, and should thus be normalized by a factor of $n^{-1}$. To address this issue, [11] introduces a normalized version of 2-IGN, defined by

$$
\begin{aligned}
W_n^{(i)}(A) = {} & \alpha_1 A + \alpha_2 A^\top + \alpha_3 \mathrm{diag}(\mathrm{diag}^*(A)) + \alpha_4 \frac{1}{n} A\mathbb{1}\mathbb{1}^\top + \alpha_5 \frac{1}{n}\mathbb{1}\mathbb{1}^\top A + \alpha_6 \frac{1}{n}\mathrm{diag}(A\mathbb{1}) \\
& + \alpha_7 \frac{1}{n} A^\top \mathbb{1}\mathbb{1}^\top + \alpha_8 \frac{1}{n}\mathbb{1}\mathbb{1}^\top A^\top + \alpha_9 \frac{1}{n}\mathrm{diag}(A^\top \mathbb{1}) + \alpha_{10}\frac{1}{n^2}(\mathbb{1}^\top A\mathbb{1})\mathbb{1}\mathbb{1}^\top \\
& + \alpha_{11}\frac{1}{n^2}(\mathbb{1}^\top A\mathbb{1})\mathrm{diag}(\mathbb{1}) + \alpha_{12}(\mathbb{1}^\top \mathrm{diag}^*(A))\mathbb{1}\mathbb{1}^\top + \alpha_{13}(\mathbb{1}^\top \mathrm{diag}^*(A))\mathrm{diag}(\mathbb{1}) \\
& + \alpha_{14}\mathrm{diag}^*(A)\mathbb{1}^\top + \alpha_{15}\mathbb{1}\mathrm{diag}^*(A)^\top + \beta_1\mathbb{1}\mathbb{1}^\top + \beta_2\mathrm{diag}(\mathbb{1}).
\end{aligned}
\tag{20}
$$

However, the normalized 2-IGN is still not compatible. Consider the third basis function $\ell_3(A) := \mathrm{diag}(\mathrm{diag}^*(A))$. It fails to satisfy the compatibility condition:

$$
\ell_3(A \otimes \mathbb{1}_m) \neq \ell_3(A) \otimes \mathbb{1}_m, m \geq 2,
$$

as the left-hand side yields a diagonal matrix, while the right-hand side generally does not. In fact, all basis maps that output diagonal matrices share this incompatibility.

Nonetheless, our Proposition 5.1 immediately provides a constructive recipe for making 2-IGN transferable: we start from a basis for linear equivariant layers $W_n^{(i)}$—which is compatible under duplication—and then select only the basis elements which have a finite operator norm as $n$ grows. Furthermore, we use nonlinearities $\rho_n^{(i)}$ which are compatible and Lipschitz continuous. Following this recipe, we introduce two modified versions of 2-IGN:

> **Generalizable Graph Neural Network (GGNN)**: Compatible with respect to $\mathbb{V}_{\mathrm{dup}}^G$, locally Lipschitz transferable under the $\infty$-norm.

> **Continuous GGNN**: Compatible with respect to $\mathbb{V}_{\mathrm{dup}}^G$, locally Lipschitz transferable under the operator 2-norm, and continuously transferable under the cut-norm.

We highlight that this is a general methodology for constructing transferable equivariant networks: the framework established in [50] yields bases for compatible equivariant linear layers. We can then select only those basis elements whose operator norms do not grow with dimension, which we have shown yields a transferable neural network.

**GGNN architecture.** A $D$-layer GGNN parameterizes an $\mathsf{S}_n$-equivariant function $\mathbb{R}_{\mathrm{sym}}^{n \times n} \times \mathbb{R}^{n \times d_1'} \to \mathbb{R}_{\mathrm{sym}}^{n \times n} \times \mathbb{R}^{n \times d_D'}$ defined via the composition

$$
W_n^{(D)} \circ \rho_n^{(D-1)} \circ \cdots \circ \rho_n^{(1)} \circ W_n^{(1)},
$$

where for the following conditions hold for all $i$.

- The linear map $W_n^{(i)} \colon (\mathbb{R}_{\mathrm{sym}}^{n \times n})^{\oplus d_i} \oplus (\mathbb{R}^n)^{\oplus d_i'} \to (\mathbb{R}_{\mathrm{sym}}^{n \times n})^{\oplus d_i} \oplus ((\mathbb{R}^n)^{\oplus d_i'})^{\oplus S}$ is a is $\mathsf{S}_n$-equivariant and compatible with the duplication embedding.

- The functions $\rho_n^{(i)} \colon (\mathbb{R}_{\mathrm{sym}}^{n \times n})^{\oplus d_i} \oplus ((\mathbb{R}^n)^{\oplus d_i'})^{\oplus S} \to (\mathbb{R}_{\mathrm{sym}}^{n \times n})^{\oplus d_i} \oplus (\mathbb{R}^n)^{\oplus d_i'}$ that are compatible with respect with the duplication embedding.

Here, $d$ and $d'$ are feature channels of $A$ and $X$, respectively— we fix $d_1 = d_D = 1$.

For the ease of notation, we assume $d_i = d_{i+1} = 1$ (The general case follows analogously). The maps $W_n^{(i)}, \rho_n^{(i)}$ are given by

$$W_n^{(i)}(A, X) = (A', (X_s')_{s=0}^S)$$
$$= \left( \alpha_1 A + \alpha_2 \frac{\mathbb{1}^\top A \mathbb{1}}{n^2} \mathbb{1}\mathbb{1}^\top + \alpha_3 \frac{\mathrm{Tr}(A)}{n} \mathbb{1}\mathbb{1}^\top + \alpha_4 \frac{1}{n}(A\mathbb{1}\mathbb{1}^\top + \mathbb{1}\mathbb{1}^\top A) \right.$$
$$+ \alpha_5 (\mathrm{diag}(A)\mathbb{1}^\top + \mathbb{1}\mathrm{diag}(A)^\top) + \sum_{j=1}^{d_i'} \left[ \alpha_{6,j}(X_{:,j}\mathbb{1}^\top + \mathbb{1}X_{:,j}^\top) + \alpha_{7,j}\frac{1}{n}(\mathbb{1}^\top X_{:,j})\mathbb{1}\mathbb{1}^\top \right]$$
$$+ \beta_1 \mathbb{1}\mathbb{1}^\top,$$
$$\left. X\Theta_{1,s} + \frac{1}{n}\mathbb{1}\mathbb{1}^\top X\Theta_{2,s} + \frac{1}{n}A\mathbb{1}\theta_{1,s}^\top + \mathrm{diag}(A)\theta_{2,s}^\top + \frac{\mathrm{Tr}(A)}{n}\mathbb{1}\theta_{3,s}^\top + \frac{\mathbb{1}^\top A\mathbb{1}}{n^2}\mathbb{1}\theta_{4,s}^\top + \mathbb{1}\beta_{2,s}^\top \right),$$
$$\rho_n^{(i)}(A, (X_s)_{s=0}^S) = \left( A, \ \sigma\left( \sum_{s=0}^S n^{-s} A^s X_s \right) \right)$$

(21)

where $\alpha, \theta, \beta$ are learnable parameters, and $\sigma \colon \mathbb{R} \to \mathbb{R}$ is an arbitrary $L$-Lipschitz entrywise nonlinearity.

Consider the input and output spaces as (variants of) $\mathbb{V}_{\mathrm{dup}}^G$, the duplication consistent sequences for graph signals. The linear layer $W_n$ in (21) parameterizes all linear $\mathsf{S}_n$-equivariant maps between these two spaces that are also compatible with the duplication embedding.

The GGNN model is a modification of the 2-IGN (19) with three key differences. Firstly, we treat the adjacency matrix and node features separately so that each layer has a graph and a signal component. Moreover, we explicitly require the matrix component to be symmetric. Secondly, we impose the compatibility with respect to the duplication embedding on the linear layers. This leads to both proper normalization of each basis function and a reduction in the total number of basis functions. In particular, all basis functions that output a diagonal matrix are removed. Thirdly, for the nonlinearity $\rho_n$, instead of the entrywise nonlinearity used in IGN, we adopt a message-passing-like nonlinearity. The form of the nonlinearity mirrors the GNN model studied in [70]. Particularly, in (21), if we set all the $\alpha$'s to 0 except $\alpha_1 = 1$, and all the $\theta$'s to 0 except $\Theta_{1,s}$, then we exactly recover the transferable GNN in [70]. Therefore, our model is at least as expressive as the GNN in [70].

**Transferability analysis of GGNN.** Even though we only impose compatibility by design, we can still prove that GGNN is Lipschitz transferable with respect to some norm. Albeit this norm is arguably too weak.

**Corollary G.5.** *Consider one layer of GGNN, $(\mathrm{GGNN}_n(A, X) = \rho_n^{(i)} \circ W_n^{(i)})$, as defined in (21), where the entrywise nonlinearity $\sigma$ is $L_\sigma$-Lipschitz. Endow the space of duplication-consistent sequences with the $\infty$-norm as defined in (12) (with respect to $\|\cdot\|_\infty$ on $\mathbb{R}^{d_i'}$). Then, the sequence of maps $(\mathrm{GGNN}_n)$ is locally Lipschitz transferable. Consequently, $(\mathrm{GGNN}_n)$ extends to a function $\mathrm{GGNN}_\infty$ on the space of graphon signals, which is $L(r)$-Lipschitz on $B(0, r)$ with respect to the symmetrized metric defined in (13) with $p = \infty$.*

*Proof.* The sequence of linear maps $(W_n^{(i)})$ in (21) is Lipschitz transferable because

$$\|W_n^{(i)}\|_{\mathrm{op}} \leq \max\left\{ |\alpha_1| + |\alpha_2| + |\alpha_3| + 2|\alpha_4| + 2|\alpha_5| + \sum_{j=1}^{d_i'}(|\alpha_{6,j}| + |\alpha_{7,j}|), \right.$$
$$\left. \|\Theta_{1,s}\|_{\mathrm{op},1,1} + \|\Theta_{2,s}\|_{\mathrm{op},1,1} + \|\theta_{1,s}\|_\infty + \|\theta_{2,s}\|_\infty + \|\theta_{3,s}\|_\infty + \|\theta_{4,s}\|_\infty \right\},$$

where $\|\Theta\|_{\mathrm{op},1,1} = \max_j \sum_k |\theta_{k,j}|$ is the max $\ell_1$ norm of a column.

For the nonlinearity $(\rho_n^{(i)})$, we consider its Fréchet derivative (since all norms are equivalent in finite-dimensional vector spaces, this is independent of the norm chosen):

$$D\rho_n^{(i)}(A, X_0, \dots, X_S)[H, H_0, \dots, H_S] = \left( H, \sum_{s=0}^{S} n^{-s} \left( \sum_{k=0}^{s-1} A^k H A^{s-1-k} \cdot X_s + A^s H_s \right) \right).$$

Hence,

$$\|D\rho_n^{(i)}(A, X_0, \dots, X_S)[H, H_0, \dots, H_S]\|_\infty$$

$$\leq \max\left( \|H\|_\infty, \sum_{s=0}^{S} \left( \sum_{k=0}^{s-1} \|A\|_\infty^{s-1}\|H\|_\infty\|X_s\|_\infty + \|A\|_\infty^s\|H_s\|_\infty \right) \right)$$

$$\leq \max\left( \|H\|_\infty, \sum_{s=0}^{S} sr^s\|H\|_\infty + r^s\|H_s\|_\infty \right)$$

$$\leq \underbrace{\left( 1 \vee \sum_{s=0}^{S} (sr^s + r^s) \right)}_{=:L(r)} \cdot \|(H, H_0, \dots, H_S)\|_\infty.$$

Therefore, for all $n$, $\rho_n^{(i)}$ is $L_\sigma L(r)$-Lipschitz on the set

$$B_n(0, r) = \{(A, X_0, \dots, X_S) : \|A, X_0, \dots, X_S\|_\infty < r\}.$$

Applying Proposition 5.1, the sequence of maps $(\mathrm{GGNN}_n)$ is locally Lipschitz transferable, where the extension $\mathrm{GGNN}_\infty$ is $(L_\sigma L(\|W_n\|_{\mathrm{op}}r)\|W_n\|_{\mathrm{op}})$-Lipschitz on the set

$$B(0, r) = \{(W, f) : \|(W, f)\|_\infty < r\}.$$

$\square$

**Continuous GGNN architecture.** We aim to further restrict GGNN to construct a model that is transferable with respect to the cut norm. By Proposition G.1, we consider endowing the consistent sequence with the operator 2-norm, which is easier to analyze. The *Continuous GGNN* is a variant of GGNN with an additional constraint on the linear layers $W_n^{(i)}$, requiring them to have bounded operator norm: $\|W_n\|_{\mathrm{op}} < \infty$ (with respect to the operator 2-norm). This constraint effectively leads to a further reduction in the set of basis functions:

$$W_n^{(i)}(A, X) = (A', (X'_s)_{s=0}^S) = \left( \alpha_1 A + \alpha_2 \frac{\mathbb{1}^\top A \mathbb{1}}{n^2} \mathbb{1}\mathbb{1}^\top + \alpha_4 \frac{1}{n}(A\mathbb{1}\mathbb{1}^\top + \mathbb{1}\mathbb{1}^\top A) \right.$$

$$\left. + \sum_{j=1}^{k} \left[ \alpha_{6,j}(X_{:,j}\mathbb{1}^\top + \mathbb{1}X_{:,j}^\top) + \alpha_{7,j}\frac{1}{n}(\mathbb{1}^\top X_{:,j})\mathbb{1}\mathbb{1}^\top \right], \quad (22)$$

$$X\Theta_{1,s} + \frac{1}{n}\mathbb{1}\mathbb{1}^\top X\Theta_{2,s} + \frac{1}{n}A\mathbb{1}\theta_{1,s}^\top + \frac{\mathbb{1}^\top A\mathbb{1}}{n^2}\mathbb{1}\theta_{4,s}^\top \right),$$

Therefore, the hypothesis class of continuous GGNN forms a strict subset of that of GGNN, with the additional constraint enabling improved transferability. Meanwhile, for the same reasons outlined for GGNN, the continuous GGNN is also at least as expressive as the GNN proposed in [70]. We use $(\mathrm{cGGNN}_n)$ to denote the sequence of functions of continuous GGNN.

**Transferability analysis of Continuous GGNN.**

**Corollary G.6.** *Consider one layer of the continuous GGNN, $(\mathrm{cGGNN}_n(A, X) = \rho_n^{(i)} \circ W_n^{(i)})$, as defined in (22), where the entrywise nonlinearity $\sigma$ is $L_\sigma$-Lipschitz. Endow the space of duplication-consistent sequences with the operator 2-norm as defined in (14) (with respect to $\|\cdot\|_\infty$ on $\mathbb{R}^{d'_i}$). Then, the sequence of maps $(\mathrm{cGGNN}_n)$ is locally Lipschitz transferable. Therefore, $(\mathrm{cGGNN}_n)$ extends to a function $\mathrm{cGGNN}_\infty$ on the space of graphon signals, which is $L(r)$-Lipschitz on $B(0, r)$ with respect to the symmetrized operator 2-metric defined in (15) with $p = 2$.*

**Remark G.7.** *By Proposition G.1, the sequence of maps* $(\mathrm{cGGNN}_n)$ *is continuously transferable with respect to the cut distance* (17) *on the space*

$$\{(W, f) : \|(W, f)\|_{\mathrm{op},2} < r, \|W\|_\infty, \|f\|_\infty < r\}.$$

*Moreover, for the convergence and transferability results stated in Proposition D.4,D.5, one can additionally obtain quantitative rates of convergence with respect to the cut distance.*

*Proof.* First, by construction, the sequence of maps $(W_n^{(i)})$ is Lipschitz transferable because

$$\|W_n^{(i)}\|_{\mathrm{op}} \leq \max \left\{ |\alpha_1| + |\alpha_2| + 2|\alpha_4| + \sum_{j=1}^{d_i'} (|\alpha_{6,j}| + |\alpha_{7,j}|), \right.$$

$$\left. \|\Theta_{1,s}\|_{\mathrm{op},1} + \|\Theta_{2,s}\|_{\mathrm{op},1} + \|\theta_{1,s}\|_\infty + \|\theta_{4,s}\|_\infty \right\} < \infty.$$

For the nonlinearity $(\rho_n^{(i)})$, the action of its Jacobian yields

$$D\rho_n^{(i)}(A, X_0, \dots, X_S)[H, H_0, \dots, H_S] = \left( H, \sum_{s=0}^{S} n^{-s} \left( \sum_{k=0}^{s-1} A^k H A^{s-1-k} \cdot X_s + A^s H_s \right) \right).$$

Hence,

$$\|D\rho_n^{(i)}(A, X_0, \dots, X_S)[H, H_0, \dots, H_S]\|_{\mathrm{op},2}$$

$$\leq \max \left( n^{-1}\|H\|_{\mathrm{op},2}, \sum_{s=0}^{S} \left( \sum_{k=0}^{s-1} \frac{\|A\|_{\mathrm{op},2}^{s-1}}{n^{s-1}} \cdot \frac{\|H\|_{\mathrm{op},2}}{n} \cdot \|X_s\|_{\overline{2}} + \frac{\|A\|_{\mathrm{op},2}^s}{n^s} \cdot \|H_s\|_{\overline{2}} \right) \right)$$

$$\leq \max \left( n^{-1}\|H\|_{\mathrm{op},2}, \sum_{s=0}^{S} s r^s \cdot \frac{\|H\|_{\mathrm{op},2}}{n} + r^s \|H_s\|_{\overline{2}} \right)$$

$$\leq \underbrace{\left( 1 \vee \sum_{s=0}^{S} (s r^s + r^s) \right)}_{=:L(r)} \cdot \|(H, H_0, \dots, H_S)\|_{\mathrm{op},2}.$$

Therefore, for all $n$, $\rho_n^{(i)}$ is $L_\sigma L(r)$-Lipschitz on the set

$$\{(A, X_0, \dots, X_S) : \|A, X_0, \dots, X_S\|_{\mathrm{op},2} < r\}.$$

Applying Proposition 5.1, the sequence of maps $(f_n)$ is locally Lipschitz transferable, where the extension $\mathrm{cGGNN}_\infty$ is $(L_\sigma L(\|W_n\|_{\mathrm{op}} r)\|W_n\|_{\mathrm{op}})$-Lipschitz on the set

$$B(0, r) = \{(W, f) : \|(W, f)\|_{\mathrm{op},2} < r\}.$$

$\square$

We can directly apply Propositions D.4 and D.5, together with the convergence rates established in Appendix D.3. This leads to transferability results for continuous GGNNs that are exactly the same as those for MPNNs, as stated in Corollary G.4.

**Related work on IGN transferability.** We discuss two closely related works, [11] and [35], that address the transferability of IGNs. Interpreting their results within our theoretical framework offers a better understanding of IGN transferability. As shown in our work, the normalized 2-IGN is not compatible with the duplication-consistent subspace $\mathbb{V}_{\mathrm{dup}}^G$, and thus fails to satisfy the convergence and transferability in Proposition 3.2. At first glance, this observation may seem to contradict [11, Theorem 2]. However, this is not the case. While [11] introduces cIGN, a "graphon analogue of IGN," and proves its continuity in the graphon space, it is crucial to note that the discrete IGN does not extend to cIGN in general:

$$\mathrm{IGN}_n(A_n, X_n) \neq \mathrm{cIGN}([A_n], [X_n]).$$

Therefore, the convergence of cIGN established in Theorem 2 of [11] does not imply the convergence or transferability of the finite-dimensional IGN model. Moreover, [11, Definition 6] introduces a constraint that resembles our compatibility condition, formulated through a restricted variant termed "IGN-small." Our definition of compatibility clarifies this notion and enables explicit constructions and practical implementations of compatible, transferable versions of IGNs.

In a more recent work, [35] adopts an approach similar to ours by imposing additional constraints on the linear layers of IGN, specifically requiring them to have bounded operator norm. This leads to the Invariant Graphon Network (IWN) model. Unlike our construction, IWN retains standard point-wise nonlinearities. As shown in [35, Proposition 5.5], it is precisely these point-wise nonlinearity layers that cause IWN to be discontinuous with respect to the cut norm, rendering it generally non-transferable in our setting. Interestingly, [35, Appendix G.4] observes that under suitable assumptions, IWN restricted to the space of simple-graph inputs (i.e., adjacency matrices with 0/1 entries) is Lipschitz continuous with respect to the cut norm and hence admit Lipschitz extensions. This implies convergence and transferability of IWN specifically under the "graphon sampling" scheme, where edges are Bernoulli-randomized. However, the limit of this convergence result does not align with the behavior on weighted graphs (i.e., adjacency matrices with non-binary entries). This discrepancy highlights the cost of lacking cut-norm continuity over the full space. This phenomenon may also explain Figure 4(c) in our GGNN experiments, where outputs on graphs sampled from the Erdős–Rényi model appear to converge (with diminishing error bars), yet to a different limit than those on the corresponding fully connected weighted graphs derived from the same graphon.

In the case of 2-IGN, our continuous cGGNN model provides a remedy for the lack of cut-norm continuity by replacing point-wise nonlinearities with message-passing-style operators, thereby ensuring Lipschitz continuity with respect to the cut norm and circumventing the issue. However, our construction is currently limited to 2-IGN and does not generalize to higher-order $k$-IGNs. As noted in [35, Section 5.1], cut-norm discontinuity is inherent to the $k$-WL hierarchy and is likely unavoidable for all higher-order GNNs with better expressivity.

**Remarks on expressive power.** The expressive power of GNNs has been widely studied in terms of their ability to distinguish non-isomorphic graphs, where a GNN is said to be $k$-WL expressive if it is as powerful as $k$-WL testing [36]. While this work does not explore expressivity, we note that the standard $k$-WL expressivity is not appropriate for studying the expressive power of graphon-compatible GNNs (i.e., GNNs that are compatible with the duplication-consistent sequence and thus extend to graphon space). This is because different-sized graphs corresponding to the same step graphon are always distinguishable by WL, but are considered equivalent by any graphon-compatible GNN. Instead, it is necessary to consider a variant of $k$-WL specifically designed for graphons [7, 35].

## H   Example 3 (point clouds): details and missing proofs from Section 5.3

In this appendix, we study the transferability of two any-dimensional architectures for point clouds. We start by presenting the consistent sequences we consider. Then, in Section H.2 we study the DS-CI model proposed in [5], which is known to be very expressive but computationally expensive. Finally, in Section H.3, we introduce a novel architecture that turns out to be much cheaper to compute.

### H.1   Duplication consistent sequence for point clouds

The duplication consistent sequence for point clouds $\mathbb{V}_{\mathrm{dup}}^P = \{(V_n), (\varphi_{N,n}), (\mathsf{G}_n)\}$ is defined as follows. The index set is again $\mathbb{N} = (\mathbb{N}, \cdot \mid \cdot)$. For each $n$, the vector spaces are $V_n = \mathbb{R}^{n \times k}$, with the group $\mathsf{G}_n = \mathsf{S}_n \times \mathsf{O}(k)$ acting on $V_n$ by

$$(g, h) \cdot X = gXh^\top.$$

For any $n \mid N$, the embedding is given by,

$$\varphi_{N,n} \colon \mathbb{R}^{n \times k} \hookrightarrow \mathbb{R}^{N \times k}$$
$$X \mapsto X \otimes \mathbb{1}_{N/n},$$

and the group embedding is

$$\theta_{N,n} \colon \mathsf{S}_n \times \mathsf{O}(k) \hookrightarrow \mathsf{S}_N \times \mathsf{O}(k)$$
$$(g, h) \mapsto (g \otimes I_{N/n}, h).$$

Analogous to the case of sets, we can identify each matrix $X \in \mathbb{R}^{n \times k}$ with a step function $f_X :$ $[0,1] \to \mathbb{R}^k$, thereby interpreting $V_\infty$ as the space of step functions with discontinuities at rational points $\mathbb{Q}$. We also view the orbit of $X$ as an empirical probability measure $\frac{1}{n} \sum_{i=1}^n \delta_{X_{i:}}$. Equivalently, this identifies the orbit of a step function $f \in V_\infty$ with $\mu_f = \mathrm{Law}(f(T))$ where $T \sim \mathrm{Unif}[0,1]$.

The orthogonal group $\mathsf{O}(k)$ acts on probability measures via push-forward: for $g \in \mathsf{O}(k)$ and a measure $\mu$, the action is given by $g \cdot \mu = g^\# \mu$, where $g^\# \mu(B) = \mu(g^{-1}(B))$ for all measurable sets $B \subseteq \mathbb{R}^k$. The orbit space of $V_\infty$ can be identified with the orbit space of empirical probability measures on $\mathbb{R}^k$ under the action of $\mathsf{O}(k)$.

**Norm on $\mathbb{V}_{\mathrm{dup}}^P$.** Consider Euclidean norm $\|\cdot\|_2$ on $\mathbb{R}^k$ which corresponds to the inner product preserved by elements of $\mathsf{O}(k)$. We equip each $V_n$ with the normalized $\ell_p$ norm:

$$\|X\|_{\overline{p}} = \begin{cases} \left(\frac{1}{n} \sum_{i=1}^n \|X_{i:}\|_2^p\right)^{1/p} & p \in [1, \infty) \\ \max_{i=1}^n \|X_{i:}\|_2 & p = \infty \end{cases}$$

By Proposition C.8, it is straightforward to verify that this norm extends naturally to $V_\infty$, and that the limit space in this case can be identified with $\overline{V_\infty} = L^p([0,1]; \mathbb{R}^k)$ of functions $f \colon [0,1] \to \mathbb{R}^k$ with norm $\|f\| = \left(\int_0^1 \|f(t)\|_2^p \, dt\right)^{1/p} < \infty$.

Analogous to the case of sets, the corresponding space of orbit closures can be identified with the space of orbit closures of probability measures on $\mathbb{R}^k$ (with finite $p$-th moments) under the $\mathsf{O}(k)$-actions. The symmetrized metric is given by:

$$\overline{\mathrm{d}}_p(f, g) = \inf_{g \in \mathsf{O}(k)} W_p(g \cdot \mu_f, \mu_g) \quad \text{for } f, g \in \overline{V_\infty}, \tag{23}$$

where $W_p$ is the Wasserstein $p$-distance with respect to the $\ell_2$-norm on $\mathbb{R}^k$.

## H.2 DeepSet for Conjugation Invariance (DS-CI)

The DS-CI model [5] is given by

$$\mathrm{DS\text{-}CI}_n : \mathbb{R}^{n \times k} \to \mathbb{R}$$

$$V \mapsto \mathrm{MLP}_c \Bigg( \mathrm{DeepSet}_{(1)} \left( \{f_j^d(VV^\top)\}_{j=1,\dots,n} \right),$$

$$\mathrm{DeepSet}_{(2)} \left( \{f_\ell^o(VV^\top)\}_{\ell=1,\dots,n(n-1)/2} \right),$$

$$\mathrm{MLP}_{(3)} \left( f^*(VV^\top) \right) \Bigg),$$

where for symmetric matrix $X \in \mathbb{R}^{n \times n}_{\mathrm{sym}}$, the invariant features are given by $f_j^d(X) =$ the $j$-th largest of the numbers $X_{11}, \dots, X_{nn}$, by $f_\ell^o(X) =$ the $\ell$-th largest of the numbers $X_{ij}, 1 \le i < j \le n$, and by $f^*(X) = \sum_{i \ne j} X_{ii} X_{ij}$.

We define *normalized DS-CI* with appropriate normalization: replacing $\mathrm{DeepSet}_{(1)}, \mathrm{DeepSet}_{(2)}$ with their normalized version (i.e. replacing the sum aggregation with the mean aggregation), and replacing $f^*(X)$ with $\overline{f^*}(X) = \frac{1}{n(n-1)} \sum_{i \ne j} X_{ii} X_{ij}$. We denote the sequence of functions of normalized DS-CI by $(\overline{\mathrm{DS\text{-}CI}}_n)$.

**Transferability analysis of normalized DS-CI.** Normalized DS-CI is not compatible with respect to $\mathbb{V}_{\mathrm{dup}}^P$. To see this, observe that under duplication, we have

$$(V \otimes \mathbb{1}_{N/n})(V \otimes \mathbb{1}_{N/n})^\top = (VV^\top) \otimes (\mathbb{1}_{N/n} \mathbb{1}_{N/n}^\top),$$

Therefore, diagonal elements of $VV^\top$ become off-diagonal elements in $(VV^\top) \otimes (\mathbb{1}_{N/n} \mathbb{1}_{N/n}^\top)$, so

$$\overline{\mathrm{DeepSet}}_{(2)} \left( \{f_\ell^o((V \otimes \mathbb{1}_{N/n})(V \otimes \mathbb{1}_{N/n})^\top)\}_{\ell=1}^{N(N-1)/2} \right) \ne \overline{\mathrm{DeepSet}}_{(2)} \left( \{f_\ell^o(VV^\top)\}_{\ell=1}^{n(n-1)/2} \right),$$

$$\overline{f^*} \left( (V \otimes \mathbb{1}_{N/n})(V \otimes \mathbb{1}_{N/n})^\top \right) \ne \overline{f^*}(VV^\top).$$

However, we can make some additional adjustments to ensure compatibility: we define the *compatible DS-CI* by modifying the inputs of $\overline{\text{DeepSet}}_{(2)}$ to be $\{f_l^a(VV^\top)\}_{l=1,\dots,n^2}$, where $f_l^a(X)$ denotes the $l$-th largest value among the entries $X_{ij}$ for $1 \leq i, j \leq n$. Additionally, we replace $\overline{f}^*(X)$ with

$$\tilde{f}_*(X) := \frac{1}{n^2} \sum_{i,j} X_{ii}X_{ij}.$$

We denote the sequence of functions of compatible DS-CI by $(\text{C-DS-CI}_n)$. We prove that this model is locally Lipschitz transferable.

**Corollary H.1.** *Endow* $\mathbb{V}_{\text{dup}}^P$ *with the normalized $\ell_p$ norm with $p \in [1, \infty]$. Assume that all activation functions in the MLPs used for DS-CI are Lipschitz. Then the sequence of maps* $(\text{C-DS-CI}_n)$ *is Lipschitz transferable on the space* $\{f : \|f\|_\infty < r\}$ *for all $r > 0$.*

*Proof.* By Proposition 5.1, it is sufficient to verify the compatibility and Lipschitz continuity of each individual layer.

- The sequence of maps

$$(\mathbb{R}^{n\times k}, \|\cdot\|_{\overline{p}}) \to (\mathbb{R}^n, \|\cdot\|_{\overline{p}}), \quad V \mapsto \text{diag}^*(VV^\top)$$

is $(2r)$-Lipschitz transferable. Indeed, it is $\mathsf{S}_n$-equivariant, $\mathsf{O}(k)$-invariant, and

$$\text{diag}^* \left((V \otimes \mathbb{1}_m)(V \otimes \mathbb{1}_m)^\top\right) = \text{diag}^*\left((VV^\top) \otimes (\mathbb{1}_m\mathbb{1}_m^\top)\right)$$
$$= \text{diag}^*(VV^\top) \otimes \mathbb{1}_m,$$

$$\left\|\text{diag}^*(VV^\top) - \text{diag}^*(WW^\top)\right\|_{\overline{p}} = \left(\frac{1}{n}\sum_{i=1}^n \left|\|V_{i:}\|_2^2 - \|W_{i:}\|_2^2\right|^p\right)^{1/p}$$
$$= \left(\frac{1}{n}\sum_{i=1}^n |\langle V_{i:} - W_{i:}, V_{i:} + W_{i:}\rangle|^p\right)^{1/p}$$
$$\leq \left(\frac{1}{n}\sum_{i=1}^n \|V_{i:} - W_{i:}\|_2^p \cdot (2r)^p\right)^{1/p}$$
$$\text{(by Cauchy-Schwarz)}$$
$$= 2r\|V - W\|_{\overline{p}},$$

for $V, W$ satisfying $\|V\|_\infty = \max_i \|V_{i:}\|_2 < r, \|W\|_\infty = \max_i \|W_{i:}\|_2 < r$.

- The sequence of maps

$$(\mathbb{R}^{n\times k}, \|\cdot\|_{\overline{p}}) \to (\mathbb{R}^{n^2}, \|\cdot\|_{\overline{p}}), \quad V \mapsto \text{vec}(VV^\top)$$

is $(2r)$-Lipschitz transferable, where the codomain is equipped with a consistent sequence structure as follows: for $g \in \mathsf{S}_n$, define $\pi(g) = g^\top \otimes g \in \mathsf{S}_{n^2}$, and let $g$ act on $\mathbb{R}^{n^2}$ by $g \cdot x = \pi(g)x$. The symmetric groups $(\mathsf{S}_n)$ are embedded into each other as usual, and the vector spaces are embedded by $\varphi_{nm,n}: \mathbb{R}^{n^2} \to \mathbb{R}^{(nm)^2}$ where

$$\varphi_{nm,n}(x) = \text{vec}(\text{reshape}_n(x) \otimes \mathbb{1}_m\mathbb{1}_m^\top),$$

and $\text{reshape}_n: \mathbb{R}^{n^2} \to \mathbb{R}^{n\times n}$ is the inverse of vec on $n \times n$ matrices. Since these are all linear maps, so is $\varphi_{nm,n}$. We then have for all $V \in \mathbb{R}^{n\times k}, g \in \mathsf{S}_n, h \in \mathsf{O}(k)$ that

$$\text{vec}\left((V \otimes \mathbb{1}_m)(V \otimes \mathbb{1}_m)^\top\right) = \text{vec}\left((VV^\top) \otimes (\mathbb{1}_m\mathbb{1}_m^\top)\right) = \varphi_{nm,n}(\text{vec}(VV^\top)),$$
$$\text{vec}\left((gVh^\top)(gVh^\top)^\top\right) = \text{vec}(gVV^\top g^\top) = \pi(g)\text{vec}(VV^\top).$$

Furthermore,

$$\|\text{vec}(VV^\top) - \text{vec}(WW^\top)\|_{\overline{p}} = \left( \frac{1}{n^2} \sum_{i,j} |\langle V_{i:}, V_{j:} \rangle - \langle W_{i:}, W_{j:} \rangle|^p \right)^{1/p}$$

$$\leq \left( \frac{1}{n^2} \sum_{i,j} (|\langle V_{i:}, V_{j:} - W_{j:} \rangle| + |\langle V_{i:} - W_{i:}, W_{j:} \rangle|)^p \right)^{1/p}$$

$$\leq \left( \frac{1}{n^2} \sum_{i,j} 2^{p-1} r^p \left( \|V_{j:} - W_{j:}\|_2^p + \|V_{i:} - W_{i:}\|_2^p \right) \right)^{1/p}$$

$$\text{(Using } (a+b)^p \leq 2^{p-1}(a^p + b^p) \text{ and Cauchy-Schwarz)}$$

$$= 2r \|V - W\|_{\overline{p}},$$

for $V, W$ satisfying $\|V\|_\infty = \max_i \|V_{i:}\|_2 < r, \|W\|_\infty = \max_i \|W_{i:}\|_2 < r$.

- The scalar maps

$$(\mathbb{R}^{n \times k}, \|\cdot\|_{\overline{p}}) \to (\mathbb{R}, |\cdot|), \quad V \mapsto \tilde{f}^*(VV^\top)$$

is $(4r^3)$-Lipschitz transferable. Indeed, it is $\mathsf{S}_n \times \mathsf{O}(k)$ invariant and

$$\tilde{f}^*((V \otimes \mathbb{1}_m)(V \otimes \mathbb{1}_m)^\top) = \tilde{f}^*(VV^\top),$$

$$\left| \tilde{f}^*(VV^\top) - \tilde{f}^*(WW^\top) \right|$$

$$\leq \frac{1}{n^2} \sum_{i,j} \left| \langle V_{i:}, V_{j:} \rangle \|V_{i:}\|_2^2 - \langle W_{i:}, W_{j:} \rangle \|W_{i:}\|_2^2 \right|$$

$$\leq \frac{1}{n^2} \sum_{i,j} |\langle V_{i:}, V_{j:} \rangle - \langle W_{i:}, W_{j:} \rangle| \, \|V_{i:}\|_2^2 + |\langle W_{i:}, W_{j:} \rangle| \left| \|V_{i:}\|_2^2 - \|W_{i:}\|_2^2 \right|$$

$$\leq 4r^3 \|V - W\|_{\overline{1}} \qquad \text{(by the previous two computations)}$$

$$\leq 4r^3 \|V - W\|_{\overline{p}},$$

for $V, W$ satisfying $\|V\|_\infty = \max_i \|V_{i:}\|_2 < r, \|W\|_\infty = \max_i \|W_{i:}\|_2 < r$.

- If the activation functions used are Lipschitz, then MLPs are Lipschitz. By Corollary F.4, the normalized DeepSet is Lipschitz transferable, assuming the constituent MLPs are Lipschitz.

Thus, our compatible DS-CI architecture is a composition of Lipschitz layers. □

Finally, we conclude that the normalized DS-CI is "approximately transferable" since it is asymptotically equivalent to the compatible DS-CI up to an error of $O(n^{-1})$.

**Lemma H.2.** *If the activations in all the MLPs used are Lipschitz, then for any sequence of inputs* $V^{(n)} \in \mathbb{R}^{n \times k}$, $|\text{C-DS-CI}_n(V^{(n)}) - \overline{\text{DS-CI}}_n(V^{(n)})| = O(n^{-1})$

*Proof.* Assume for $x \in \mathbb{R}^n$, $\overline{\text{DeepSet}}_{(2)}(x) = \sigma(\frac{1}{n} \sum_i \rho(x_i))$, and $\sigma, \rho$ are $L_\sigma, L_\rho$ Lipschitz respectively. Then, we have

$$\left| \overline{f}^*(VV^\top) - \tilde{f}^*(VV^\top) \right|$$

$$\leq \left( \frac{1}{n(n-1)} - \frac{1}{n^2} \right) \sum_{i \neq j} |(VV^\top)_{ii}(VV^\top)_{ij}| + \frac{1}{n^2} \sum_i (VV^\top)_{ii}^2 = O(n^{-1})$$

and, moreover,

$$\left|\overline{\text{DeepSet}}_{(2)}\Big(\{f_\ell^o(VV^\top)\}_{\ell=1,\ldots,n(n-1)/2}\Big) - \overline{\text{DeepSet}}_{(2)}\Big(\{f_\ell^a(VV^\top)\}_{\ell=1,\ldots,n^2}\Big)\right|$$

$$\leq L_\sigma\left(\left(\frac{1}{n(n-1)} - \frac{1}{n^2}\right)\sum_{i\neq j}\big|\rho((VV^\top)_{ij})\big| + \frac{1}{n^2}\sum_i\big|\rho((VV^\top)_{ii})\big|\right) = O(n^{-1}).$$

Since every layer is Lipschitz, this leads to an overall error of $O(n^{-1})$. $\qquad\square$

Following the analysis above, we can directly apply Propositions D.4 and D.5, together with the convergence rates established in Appendix D.3. Moreover, observe that the $O(n^{-1})$ discrepancy between normalized DS-CI and compatible DS-CI is dominated by the convergence rate of interest. This immediately yields the following transferability result.

**Corollary H.3** (Transferability of normalized DS-CI). *Let $(X_n) \in \mathbb{R}^{n\times k}$ be a sequence of matrices sampled from the same underlying distribution $\mu \in \mathcal{P}_3(\mathbb{R}^k)$ with bounded support in the following way: first, sample $Y_n \in \mathbb{R}^{n\times k}$ with rows drawn i.i.d. from $\mu$. Then, let $G \in \mathrm{O}(k)$ be a (random or deterministic) rotation matrix, sampled independently of $Y_n$, and define $X_n = Y_nG$. Then,*

$$\mathbb{E}\left[\big|\overline{\text{DS-CI}}_n(X_n) - \overline{\text{DS-CI}}_m(X_m)\big|\right]$$

$$\lesssim \begin{cases} n^{-1/2} + m^{-1/2} & \text{if } k = 1, \\ n^{-1/2}\log(1+n) + m^{-1/2}\log(1+m) & \text{if } k = 2, \\ n^{-1/k} + m^{-1/k} & \text{if } k > 2. \end{cases}$$

### H.3 Constructing new transferable models: SVD-DS

We propose the SVD-DS model defined as follows:

$$\overline{\text{SVD-DS}}_n : \mathbb{R}^{n\times k} \to \mathbb{R}, \quad X \mapsto \overline{\text{DeepSet}}_n(XV),$$

where $X = UDV^\top$ is the singular value decomposition (SVD) for $X$ with ordered singular values. We proceed to show that it is locally transferable almost everywhere on its domain, and that its performance is competitive with DS-CI while being more computationally efficient.

**Transferability analysis of SVD-DS.** We extend the SVD to elements in the limit space $\overline{V_\infty} = L^2([0,1],\mathbb{R}^k)$ and analyze its continuity, yielding the following transferability result. Recall our definition of *locally Lipschitz transferable at a point* in Appendix D.1.

**Corollary H.4.** *Endow the duplication consistent sequences with the normalized $\ell_2$ norm induced by $\|\cdot\|_2$ on $\mathbb{R}^k$. Observe that*

$$\|X\|_{\overline{2}} := \left(\frac{1}{n}\sum_{i=1}^n \|X_{i:}\|_2^2\right)^{1/2} = \frac{1}{\sqrt{n}}\|X\|_F,$$

*where $\|\cdot\|_F$ denotes the Frobenius norm of a matrix. Then, the sequence of maps $(\overline{\text{SVD-DS}}_n)$ is compatible and locally Lipschitz transferable at every point except for a set of measure zero, corresponding to points with non-distinct singular values.*

**Remark H.5.** *This transferability result is weaker than the "$L(r)$-locally Lipschitz transferability" defined in Definition 3.1, since our model may be discontinuous at points with non-distinct singular values. Therefore, in this case our transferability results in Propositions D.4 and D.5 only apply to sequences $(x_n)$ converging to a limit $x \in \overline{V_\infty}$ with distinct singular values.*

*Proof.* Decompose $\overline{\text{SVD-DS}}_n$ as the composition

$$\mathbb{V}_{\text{dup}}^P = \{\mathbb{R}^{n\times k}\} \xrightarrow{X\mapsto XV} \mathbb{U} = \{\mathbb{R}^{n\times k}\} \xrightarrow{\overline{\text{DeepSet}}} \mathbb{V}_\mathbb{R},$$

where $\mathbb{V}_\mathbb{R}$ is the trivial consistent sequence over $\mathbb{R}$, and the consistent sequence $\mathbb{U}$ consists of vector spaces $U_n = \mathbb{R}^{n\times k}$ under the duplication embedding $\otimes\mathbb{1}$. The group $\mathsf{S}_n \times \mathrm{O}(k)$ acts on $U_n$ by $(g,h)\cdot X = gX$, i.e., the action of $\mathrm{O}(k)$ is trivial. By Corollary F.4, the normalized DeepSet map

is Lipschitz transferable, assuming the constituent MLPs are Lipschitz. It remains to show that the SVD-based map $X \mapsto XV$ extends to a function that is locally Lipschitz at every point with distinct singular values. We show this in Proposition H.7 below after extending the SVD to all of $\overline{V_\infty}$ and considering its ambiguities. $\qquad\square$

Following the analysis above, we can directly apply Propositions D.4 and D.5, together with the convergence rates established in Appendix D.3. These results immediately yield a transferability result for SVD-DS.

**Corollary H.6** (Transferability of SVD-DS)**.** *Let* $(X_n) \in \mathbb{R}^{n \times k}$ *be a sequence of matrices sampled from the same underlying distribution* $\mu \in \mathcal{P}_4(\mathbb{R}^k)$ *in the following way: First, sample* $Y_n \in \mathbb{R}^{n \times k}$ *with rows drawn i.i.d. from* $\mu$*. Then, let* $G \in \mathrm{O}(k)$ *be a (random or deterministic) rotation matrix, sampled independently of* $Y_n$*, and define* $X_n = Y_n G$*. Suppose the second moment* $\mathbb{E}_{x \sim \mu} x x^\top \in \mathbb{R}^{k \times k}$ *of* $\mu$ *has* $k$ *distinct eigenvalues, then*

$$
\mathbb{E}\left[\left|\overline{\text{SVD-DS}}_n(X_n) - \overline{\text{SVD-DS}}_m(X_m)\right|\right]
$$
$$
\lesssim \begin{cases} n^{-1/4} + m^{-1/4} & \text{if } k < 4, \\ n^{-1/4}\log^{1/2}(1+n) + m^{-1/4}\log^{1/2}(1+m) & \text{if } k = 4, \\ n^{-1/k} + m^{-1/k} & \text{if } k > 4. \end{cases}
$$

Note that the functional SVD is locally Lipschitz only at points where the singular values are distinct. This motivates our assumption on the distribution $\mu$ in the Corollary. To see this, observe that when $n \geq k$, the singular values of $X_n$ are the same as those of $Y_n$, and are the square roots of the eigenvalues of $Y_n^\top Y_n = \sum_{i=1}^n x_i x_i^\top$, where $x_i$ are the rows of $Y_n$, sampled i.i.d. from $\mu$. The functional singular values of $X_n$ (i.e. $\frac{1}{\sqrt{n}}$ of the usual matrix singular values) is then $\frac{1}{n}\sum_{i=1}^n x_i x_i^\top$. By the law of large numbers, each entry of this matrix converges almost surely:

$$
\left(\frac{1}{n}\sum_{i=1}^n x_i x_i^\top\right)_{mn} \xrightarrow{\text{a.s.}} \Sigma_{mn} \text{ for all } m, n \in [k],
$$

where $\Sigma := \mathbb{E}_{x \sim \mu}[xx^\top]$ is the second-moment of $\mu$. It follows from Weyl's theorem that each eigenvalue converges almost surely to those of $\Sigma$:

$$
\left|\lambda_j\left(\frac{1}{n}\sum_{i=1}^n x_i x_i^\top\right) - \lambda_j(\Sigma)\right| \leq \left\|\frac{1}{n}\sum_{i=1}^n x_i x_i^\top - \Sigma\right\|_2 \xrightarrow{a.s.} 0 \text{ for all } j \in [k].
$$

Therefore, if $\Sigma$ has distinct eigenvalues, then with probability one, the empirical matrix has distinct eigenvalues for all sufficiently large $n$, ensuring that the functional SVD is locally Lipschitz at this point.

**Functional SVD and its local Lipschitz continuity.** We can identify the space $\overline{V_\infty} = L^2([0,1], \mathbb{R}^k)$ with $\mathcal{L}(L^2([0,1]), \mathbb{R}^k)$, the space of bounded linear maps $L^2([0,1]) \to \mathbb{R}^k$ endowed with the Hilbert-Schmidt norm $\|\cdot\|_{HS}$. In more detail, each $X \in L^2([0,1], \mathbb{R}^k)$ can be written as a sequence of rows $X = (f_1, \ldots, f_k)^\top$ where $f_i \in L^2([0,1])$, and such $X$ defines the bounded linear map $Xf = (\langle f_1, f\rangle, \ldots, \langle f_k, f\rangle)^\top$. Conversely, any bounded linear map $X \colon L^2([0,1]) \to \mathbb{R}^k$ is of the form $Xf = (\langle f_1, f\rangle, \ldots, \langle f_k, f\rangle)^\top$ for some $f_1, \ldots, f_k \in L^2([0,1])$ which we view as the columns of $X$, and $\|X\|_{HS}^2 = \sum_{i=1}^k \|f_i\|_2^2$. Here $V_n = \mathbb{R}^{n \times k}$ is viewed as the subspace of $\overline{V_\infty}$ with piecewise-constant columns $f_i$ on consecutive intervals of length $1/n$.

Note that $X$ vanishes identically on $\mathcal{V}_k = \mathrm{span}\{f_1, \ldots, f_k\}^\perp$, while $X \colon \mathcal{V}_k \to \mathbb{R}^k$ is a linear map between finite-dimensional vector spaces and therefore admits a singular value decomposition. Thus, there exists positive numbers $\sigma \in \mathbb{R}_{\geq 0}^k$, orthonormal $v_1, \ldots, v_k \in \mathbb{R}^k$, and orthonormal functions $u_1, \ldots, u_k \in L^2([0,1])$ satisfying

$$
X = \sum_{i=1}^k \sigma_i \langle u_i, \cdot\rangle v_i. \tag{24}
$$

If $X \in \mathbb{R}^{n \times k}$ and $X = \sum_{i=1}^k \tilde{\sigma}_i \tilde{u}_i \tilde{v}_i^\top$, is the usual SVD of $X$, then $\sigma_i = \tilde{\sigma}_i/\sqrt{n}$, $v_i = \tilde{v}_i$, and $u_i(t) = \sqrt{n}[\tilde{u}_i]_{\lceil nt \rceil}$ is the functional SVD of $X$ as in (24). Conversely, if (24) is the functional SVD

of such an $X$ then $X = \sum_{i=1}^{k} (\sigma_i \sqrt{n})([u_i(j/n)]_{j=1}^{k}/\sqrt{n})v_i^\top$ is the usual SVD of $X$. Note that the right singular vectors $V$ are the same in both SVDs.

If for any $X \in V_n$ we let $\sigma(X)$ be its (functional) singular values from (24) and $\tilde{\sigma}(X)$ be its usual singular values, then

$$\|X\|_{\overline{2}}^2 = \frac{1}{n}\|X\|_F^2 = \frac{1}{n}\sum_{i=1}^{k}\tilde{\sigma}_i(X)^2 = \sum_{i=1}^{k}\sigma_i(X)^2, \tag{25}$$

and by Mirsky's inequality [61],

$$\|\sigma(X) - \sigma(Y)\|_2 = \frac{1}{\sqrt{n}}\|\tilde{\sigma}(X) - \tilde{\sigma}(Y)\|_2 \le \frac{1}{\sqrt{n}}\|X - Y\|_F = \|X - Y\|_{\overline{2}}. \tag{26}$$

Furthermore, whenever $V \in \mathbb{R}^{k \times k}$ and $X \in V_n$ we have

$$\|XV\|_{\overline{2}} = \frac{1}{\sqrt{n}}\|XV\|_F \le \frac{1}{\sqrt{n}}\|V\|_F \tilde{\sigma}_1(X) = \|V\|_F \sigma_1(X). \tag{27}$$

Note that the final bounds in all of the above inequalities are independent of $n$, continuous in $\|\cdot\|$, and hold for all $X \in V_\infty$. We therefore conclude that they also hold for all $X \in \overline{V_\infty}$, with $\|\cdot\|_{\overline{2}}$ replaced with $\|\cdot\|_{HS}$.

There is an ambiguity in the above decomposition, since if $(u_i), (v_i)$ satisfy (24) then so do $(s_i u_i), (s_i v_i)$ for any choice of signs $s_i \in \{\pm 1\}$. Furthermore, if the singular values $(\sigma_i)$ are distinct then this is the only ambiguity in (24), see [13]. To disambiguate the SVD, we therefore choose signs so that $v_i > -v_i$ in lexicographic order. When the entries of the $v_i$ are all nonzero, this amounts to requiring the first row of $V = [v_1, \ldots, v_k]$ to be positive. We proceed to prove that the map $X \mapsto V(X) = [v_1, \ldots, v_k]$ with this choice of signs is locally Lipschitz continuous on a dense subset of $\overline{V_\infty}$.

**Proposition H.7.** *Fix $X_0 \in \overline{V_\infty}$ with distinct singular values and all-nonzero entries in its right singular vectors $(v_i)$. Let $\mathrm{gap}_p(X_0) = \min_{2 \le i \le k}\{\sigma_{i-1}(X_0)^p - \sigma_i(X_0)^p\}$ be the minimum gap between $p$-th powers of (functional) singular values of $X_0$, and set*

$$B(X_0) = \frac{\sqrt{8}(2\sigma_1(X_0) + 1)}{\mathrm{gap}_2(X_0)}. \tag{28}$$

*For any $\widehat{X} \in \overline{V_\infty}$ satisfying*

$$\|X_0 - \widehat{X}\|_{HS} \le 1 \wedge \frac{\mathrm{gap}_1(X_0)}{2\sqrt{k}} \wedge \frac{1}{2B(X_0)} \min_{1 \le i,j \le k} |[v_i(X_0)]_j|,$$

*the following two hold true.*

1. *The matrix $\widehat{X}$ has distinct singular values, and all nonzero entries of $v_i(\widehat{X})$ have the same sign as those of $v_i(X_0)$.*

2. *We have*
$$\|V(X_0) - V(\widehat{X})\|_F \le kB(X_0)\|X_0 - \widehat{X}\|_{HS}, \tag{29}$$
   *and*
$$\|X_0 V(X_0) - \widehat{X}V(\widehat{X})\|_{HS} \le (k\sigma_1(X_0)B(X_0) + 1)\|X_0 - \widehat{X}\|_{HS}. \tag{30}$$

*Proof.* We start by establishing the first claim. If $\|X_0 - \widehat{X}\|_{HS} \le \frac{\mathrm{gap}_1(X_0)}{2\sqrt{k}}$, then for any $2 \le i \le k$ we have by

$$\sigma_{i-1}(\widehat{X}) - \sigma_i(\widehat{X}) = \sigma_{i-1}(\widehat{X}) - \sigma_{i-1}(X_0) + \sigma_{i-1}(X_0) - \sigma_i(X_0) + \sigma_i(X_0) - \sigma_i(\widehat{X})$$

$$\ge (\sigma_{i-1}(X_0) - \sigma_i(X_0)) - \sum_{i=1}^{k}|\sigma_i(\widehat{X}) - \sigma_i(X_0)|$$

$$\ge \mathrm{gap}_1(X_0) - \sqrt{k}\|\sigma(X_0) - \sigma(\widehat{X})\|_2 \qquad \text{(Cauchy-Schwarz)}$$

$$\ge \frac{\mathrm{gap}_1(X_0)}{2} > 0. \qquad \text{(Mirsky's inequality (26))}$$

Thus, $\widehat{X}$ has distinct singular values.

Let $\widetilde{\mathrm{gap}}_p(X_0) = \min_{2 \le i \le k}\{\tilde{\sigma}_{i-1}(X_0)^p - \tilde{\sigma}_i(X_0)^p\}$ be the minimum gap between $p$-th powers of (usual) singular values of $X_0$. For $X_0, \widehat{X} \in V_n$, the result [90, Thm. 4] shows that for each $i \in [k]$

$$\min_{s \in \{\pm 1\}} \|v_i(X_0) - s \cdot v_i(\widehat{X})\|_2$$

$$\le \frac{\sqrt{8}(2\tilde{\sigma}_1(X_0) + \tilde{\sigma}_1(X_0 - \widehat{X}))\|X_0 - \widehat{X}\|_F}{\widetilde{\mathrm{gap}}_2(X_0)}$$

$$= \frac{\sqrt{8n}(2\sigma_1(X_0) + \sigma_1(X_0 - \widehat{X}))\|X_0 - \widehat{X}\|_F}{n\mathrm{gap}_2(X_0)} \qquad \text{(since } \tilde{\sigma}_i(X) = \sigma_i(X)\sqrt{n}\text{)}$$

$$= \frac{\sqrt{8}(2\sigma_1(X_0) + \sigma_1(X_0 - \widehat{X}))\|X_0 - \widehat{X}\|_{\overline{2}}}{\mathrm{gap}_2(X_0)} \qquad \text{(since } \|X\|_{\overline{2}} = \|X\|_F/\sqrt{n}\text{)}$$

$$\le \frac{\sqrt{8}(2\sigma_1(X_0) + \|X_0 - \widehat{X}\|_{\overline{2}})\|X_0 - \widehat{X}\|_{\overline{2}}}{\mathrm{gap}_2(X_0)} \qquad \text{(by (25), } \sigma_1(X)^2 \le \|X\|_{\overline{2}}^2 = \sum_{i=1}^k \sigma_i(X)^2\text{)}$$

$$\le \frac{\sqrt{8}(2\sigma_1(X_0) + 1)\|X_0 - \widehat{X}\|_{\overline{2}}}{\mathrm{gap}_2(X_0)} \qquad \text{(since } \|\widehat{X} - X_0\|_{\overline{2}} \le 1\text{)}$$

$$= B(X_0)\|X_0 - \widehat{X}\|_{\overline{2}}.$$

The final bound is independent of $n$ and hence applies on all of $V_\infty$. It is also continuous in $\|\cdot\|_{\overline{2}}$ on the dense subset of $V_\infty$ consisting of operators with distinct singular values, so taking closures we conclude that this bound applies for any $X_0, \widehat{X} \in \overline{V_\infty}$ such that $\mathrm{gap}_2(X_0) > 0$. Combining the last line above with our bound on $\|\widehat{X} - X_0\|_{\overline{2}}$, we get

$$\min_{s \in \{\pm 1\}} \max_{1 \le j \le k} \left|[v_i(X_0)]_j - s \cdot [v_i(\widehat{X})]_j\right| \le \min_{s \in \{\pm 1\}} \left\|v_i(X_0) - s \cdot v_i(\widehat{X})\right\|_2 \le \frac{1}{2} \min_{1 \le i, j \le k} |[v_i(X_0)]_j|. \tag{31}$$

Thus, the sign $s$ achieving the above minimum is the one making *all* entries of $v_i(\widehat{X})$ have the same sign as those of $v_i(X_0)$. Since the first entries of $v_i(X_0)$ and of $v_i(\widehat{X})$ are positive, we conclude that $s = 1$ achieves the above minimum.

To prove the second claim, we combine the bounds above, which yields

$$\|V(X_0) - V(\widehat{X})\|_F \le \sum_{i=1}^k \|v_i(X_0) - v_i(\widehat{X})\|_2$$

$$= \sum_{i=1}^k \min_{s \in \{\pm 1\}} \|v_i(X_0) - s \cdot v_i(\widehat{X})\|_2$$

$$\le kB(X_0)\|X_0 - \widehat{X}\|_{HS},$$

as claimed. Finally, applying the triangle inequality gives

$$\|X_0 V(X_0) - \widehat{X} V(\widehat{X})\|_{HS} \le \|X_0(V(X_0) - V(\widehat{X}))^\top\|_{HS} + \|(X_0 - \widehat{X})V(\widehat{X})\|_{HS}$$

$$\le kB(X_0)\sigma_1(X_0)\|X_0 - \widehat{X}\|_{HS} + \|X_0 - \widehat{X}\|_{HS}, \qquad \text{(by (27))}$$

yielding the last claim. $\qquad\square$

**SVD-DS is computationally more efficient than DS-CI.** When $k \ll n$ (for example, for us $k = 2$ or 3), to evaluate SVD-DS on a given input of size $n \times k$, we need to compute its SVD—which takes $O(nk^2)$ flops [32, Section 8.6]—and then evaluate $\overline{\mathrm{DeepSet}}$ on the output, which takes $O(nC(k))$ where $C(k)$ is the cost of evaluating the involved fixed-size MLPs taking inputs of size $k$. Moreover, during training, we can compute the SVD of the dataset once in advance. In contrast, for DS-CI we need to form $VV^\top$ at a cost of $O(n^2k)$, and this needs to be differentiated through during training. After forming $VV^\top$, we evaluate normalized DeepSets on its entries at a cost of $O(n^2)$. Thus, SVD-DS is much faster to train and deploy than DS-CI.

**Transferability plots.** The numerical experiments illustrating the transferability of SVD-DS and normalized and compatible DS-CI is shown in Figure 6.

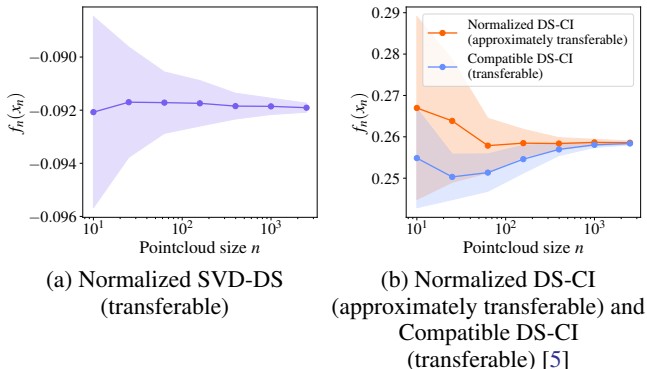

(a) Normalized SVD-DS
(transferable)

(b) Normalized DS-CI
(approximately transferable) and
Compatible DS-CI
(transferable) [5]

Figure 6: Transferability of invariant models on point clouds with respect to $(\mathbb{V}_{\mathrm{dup}}^P, \|\cdot\|_{\overline{p}})$. The plot shows outputs of untrained, randomly initialized models for a sequence of point clouds $X_n \in \mathbb{R}^{n \times k}$, where each point is sampled i.i.d. from $\mathcal{N}(0, I_k)$. The error bars extend from the mean to $\pm$ one standard deviation over 100 random samples. The figure shows that transferable models generate convergent outputs. Figure (b) shows the asymptotic equivalence between the normalized DS-CI and compatible DS-CI as proved in Lemma H.2.

# I    Size generalization experiments: details from Section 6

In this section, we provide details of our size generalization experiments. All experiments were implemented using the `PyTorch` framework and trained on a single NVIDIA A5000 GPU. Specific training and model configurations are provided in the descriptions of the individual experiments. For all experiments, the training dataset consists of inputs with a fixed, small dimension $n_{\mathrm{train}}$. For evaluation, we use a series of test datasets where the input dimension $n_{\mathrm{test}}$ is progressively larger than $n_{\mathrm{train}}$.

## I.1    Size generalization on sets

We consider two any-dimensional learning tasks on sets, where the target functions have different properties, so that different models are expected to perform better. In both experiments, we compare the size generalization of three models: DeepSet, normalized DeepSet, and PointNet as analyzed in Appendix F.2. The maps $\sigma$ and $\rho$ in the model are parametrized by three fully connected layers with a hidden dimension of 50 and ReLU activation functions. Training was performed by minimizing the MSE loss using AdamW with an initial learning rate of $0.001$ and a weight decay of $0.1$. The learning rate was halved if the validation loss did not improve for 50 consecutive epochs. Each model was trained for 1000 epochs, with each run taking less than three minutes to complete.

### I.1.1    Experiment 1: Population statistics

We adopt the experimental setup from [91, Section 4.1.1], which comprises four distinct tasks on population statistics. In all four tasks, the datasets consist of sets where each set contains i.i.d. samples from a distribution $\mu$, where $\mu$ itself is sampled from a parameterized distribution family. The objective is to learn either the entropy or the mutual information of the distribution $\mu$.

While the original experiment in [91] focused on training and testing with set sizes $n_{\mathrm{train}} = n_{\mathrm{test}} = [300, 500]$, we instead evaluate size generalization. During the training stage, the dataset consists of $N = 2048$ sets, each of size $n_{\mathrm{train}} = 500$. This dataset is randomly split into training, validation, and test data with proportions $50\%, 25\%$, and $25\%$, respectively. During the evaluation stage, the trained model is tested on a sequence of datasets with set sizes $n_{\mathrm{test}} \in \{500, 1000, 1500, \ldots, 4500\}$, each consisting of $N = 512$ sets. The descriptions of the four tasks, as originally presented in [91], are provided below:

**Rotation:** Generate $N$ datasets of size $M$ from $\mathcal{N}(0, R(\alpha)\Sigma R(\alpha)^T)$ for random $\Sigma$ and $\alpha \in [0, \pi]$. The goal is to learn the marginal entropy of the first dimension.

**Correlation:** Generate sets from $\mathcal{N}(0, [\Sigma, \alpha\Sigma; \alpha\Sigma, \Sigma])$ for random $\Sigma$ and $\alpha \in (-1, 1)$. The goal is to learn the mutual information between the first 16 and last 16 dimensions.

**Rank 1:** Generate sets from $\mathcal{N}(0, I + \lambda vv^T)$ for random $v \in \mathbb{R}^{32}$ and $\lambda \in (0, 1)$. The goal is to learn the mutual information.

**Random:** Generate sets from $\mathcal{N}(0, \Sigma)$ for random $32 \times 32$ covariance matrices $\Sigma$. The goal is to learn the mutual information.

For all tasks, the target functions are scalar functions on the underlying probability measure $\mu$, and are continuous with respect to the Wasserstein $p$-distance. Based on Appendix F.2.1, normalized DeepSet is well-aligned with the task from a continuity perspective and PointNet aligns from a compatibility perspective, while unnormalized DeepSet lacks alignment altogether. The results are summarized in Figure 7, showing that stronger task-model alignment improves both in-distribution and size-generalization performance.

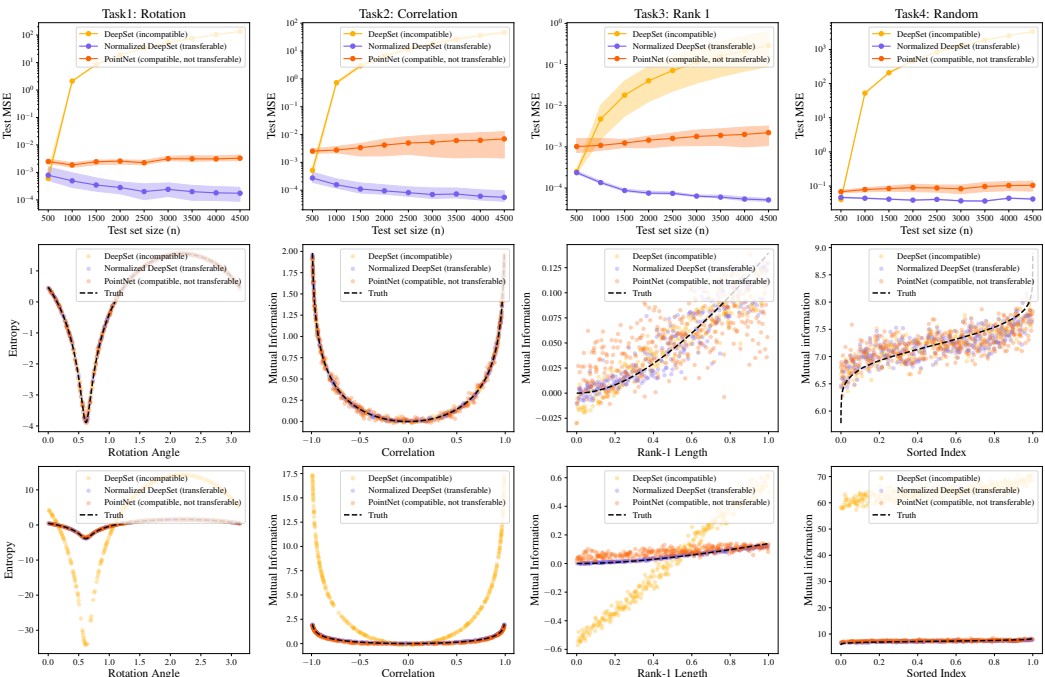

Figure 7: Size generalization for population statistics experiment. All models are trained on set size $n_{\text{train}} = 500$ and tested on set sizes $n_{\text{test}} \in \{500, 1000, \ldots, 4500\}$. **Top**: MSE (log scale) vs. test set size. The solid line denotes the mean, and the error bars extend from the minimum to the maximum test MSE over 10 randomly initialized trainings. Normalized DeepSet performs better than PointNet, which in turn outperforms DeepSet. **Middle**: Test-set predictions of the three models vs. ground truth for $n_{\text{test}} = n_{\text{train}} = 500$. All models perform similarly. **Bottom**: Test-set predictions vs. ground truth for $n_{\text{test}} = 4500 \gg n_{\text{train}} = 500$. DeepSet has blown-up outputs due to scaling. Normalized DeepSet outperforms PointNet in Task 3.

### I.1.2 Experiment 2: Maximal distance from the origin

We consider the following data and task: each dataset consists of sets where each set contains $n$ two-dimensional points sampled as follows. First, a center is sampled from $\mathcal{N}(0, I_2)$ and a radius is sampled from $\text{Unif}([0, 1])$, which together define a circle. The set then consists of $n$ points sampled uniformly along the circumference. The goal is to learn the maximum Euclidean norm among the points in each set. Equivalently, our goal is to learn the so-called Hausdorff distance $d_H(\{0\}, X) = \sup_{x \in X} \|x\|$; see (11). Hence, the target function depends only on the support of the point cloud and is continuous with respect to the Hausdorff distance.

Once more, we evaluate size generalization. During the training stage, the dataset consists of $N = 5000$ sets, each of size $n_{\text{train}} = 20$. The dataset is randomly split into training, validation, and test data with proportions $80\%, 10\%$, and $10\%$, respectively. During the evaluation stage, the trained model is tested on a sequence of datasets with set sizes $n_{\text{test}} \in \{20, 40, \dots, 200\}$, each consisting of $N = 1000$ sets. For this task, PointNet aligns from a continuity perspective, normalized DeepSet from a compatibility perspective, and DeepSet does not align with the learning task. The results are summarized in Figure 8, showing that the model performance improves with task-model alignment.

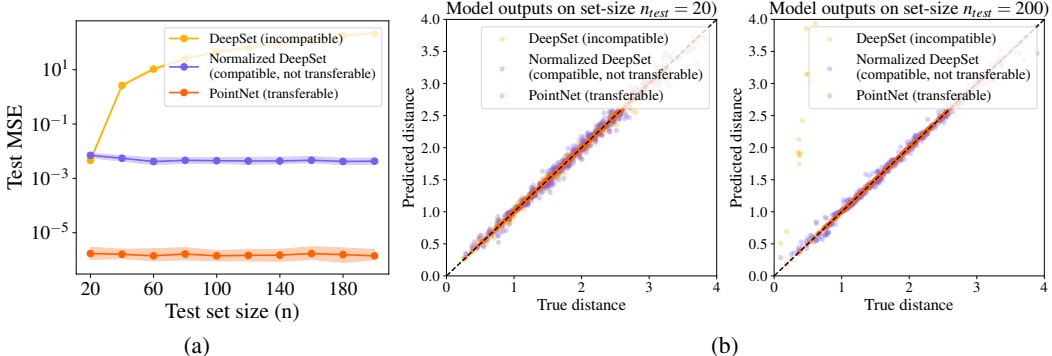

Figure 8: Size generalization on max-distance-from-origin task. All models are trained on set size $n_{\text{train}} = 20$ and tested on set sizes $n_{\text{test}} \in \{20, 40, \dots, 200\}$. **Figure (a):** Test MSE (log scale) vs. test set size. The solid line denotes the mean, and the error bars extend from the minimum to the maximum test MSE over 10 randomly initialized trainings. PointNet performs better than normalized DeepSet, which in turn outperforms DeepSet. **Figure (b):** Test-set predictions vs. ground truth for set size $n_{\text{test}} = n_{\text{train}} = 20$. The middle figure shows predictions for $n_{\text{train}} = 20$, while the right-most figure shows predictions for $n_{\text{test}} = 200$. PointNet is very accurate in both cases. DeepSet exhibits incorrect blown-up outputs in the latter case.

## I.2 Size generalization on graphs

For graphs, the task is to learn a homomorphism density of degree three. To formally describe the task, we first describe its inputs. We consider a dataset consisting of $N$ attributed graphs $(A, x)$ generated according to the following two procedures (the first is described and reported in the main paper):

1. Each graph is a fully connected weighted graph whose adjacency matrix has entries $A_{ij} = A_{ji} \overset{\text{i.i.d.}}{\sim} \text{Unif}([0,1])$ for $i \leq j$, and node features $x_i \overset{\text{i.i.d.}}{\sim} \text{Unif}([0,1])$.

2. First, sample the number of clusters $K$ uniformly from $\{10, \dots, 20\}$, and construct a $K \times K$ symmetric probability matrix $P$ with entries sampled uniformly from $[0,1]$. The resulting stochastic block model (SBM) is used to generate an undirected, simple graph with edges sampled as $A_{ij} = A_{ji} \overset{\text{i.i.d.}}{\sim} \text{Ber}(P_{z_i, z_j})$ for $i \leq j$, where $z_i \overset{\text{i.i.d.}}{\sim} \text{Unif}(\{1, \dots, K\})$ are cluster assignments. Node features are given by $x_i = \gamma_{z_i}$, where $\gamma \in \mathbb{R}^K$ with i.i.d. entries $\text{Unif}([0,1])$.

The task is to learn the rooted, signal-weighted homomorphism density of degree three sending

$$\mathbb{R}^{n \times n}_{\text{sym}} \times \mathbb{R}^n \to \mathbb{R}^n, \quad (A, x) \mapsto y, \quad \text{via} \quad y_i = \frac{1}{n^2} \sum_{j, k \in [n]} A_{ij} A_{jk} A_{ki} x_i x_j x_k.$$

Note that when $x = \mathbb{1}$, $y_i$ corresponds to a normalized count of triangles centered at node $i$. Thus, this can be interpreted as a signal-weighted triangle density. This formulation is related to the signal-weighted homomorphism density studied in [35], which generalizes the notion of homomorphism density extensively studied in graphon theory [54].

We conduct experiments to evaluate the size generalization performance of the following models: the GNN from [70], the normalized 2-IGN [56], and our proposed GGNN and continuous GGNN. ReLU is used as the entry-wise activation function in all models. We choose the number of layers

and hidden dimensions such that each model has approximately 2k parameters to ensure an arguably fair comparison. During training, we use a dataset of $N = 5000$ graphs, each with $n_{\text{train}} = 50$ nodes. This dataset is randomly split into training, validation, and test with proportions $60\%, 20\%,$ and $20\%$, respectively. For evaluation, we test the trained models on datasets of graph sizes $n_{\text{test}} \in \{50, 200, 500, 1000, 2000\}$, each containing $N = 1000$ graphs. Training is performed by minimizing the MSE loss using AdamW with an initial learning rate of $0.001$ and weight decay of $0.1$. Each model is trained for 500 epochs. Training a single run takes less than 3 minutes for the GNN model, and approximately 6–9 minutes for the IGN, GGNN, and continuous GGNN models, which are more computationally intensive. We note that evaluation on large graphs is particularly time- and memory-intensive for these models, taking up to several hours. Memory limitations restrict the maximum graph size we can evaluate to $n = 2000$.

The results of the size generalization experiments are summarized in Figure 9. Let us elaborate on these results. Since the target function naturally extends to the graphon-level signal-weighted triangle density $(W, f) \mapsto g$, given by

$$g(x) = \int_{[0,1]^2} W(x,y)W(y,z)W(z,x)f(x)f(y)f(z)\,dy\,dz,$$

which is continuous with respect to the cut norm, models that are continuous under this topology—such as the GNN and continuous GGNN—are aligned with the task at the level of continuity. Our proposed continuous GGNN, which is provably transferable and possibly more expressive than the GNN, achieves the best performance. Although GGNN is not transferable under the cut norm, it is transferable under a weaker topology (see Appendix G.3), enabling it to perform reasonably well. In contrast, the 2-IGN model, even after proper normalization, exhibits divergent outputs for larger graph sizes, indicating a lack of compatibility with the task.

Finally, we remark that the expressive power of various GNN architectures with respect to homomorphism densities has been extensively studied. Prior work has shown that common GNNs—including those considered in this study—are generally unable to express high-degree homomorphism densities [14, 35]. However, our results demonstrate that GNNs can still achieve strong performance on this task when evaluated over certain large parametric families of random graph models. This does not contradict prior theoretical findings, as our results pertain to an *average-case* evaluation, while the negative results in the literature are established in the *worst-case* setting.

### I.3 Size generalization on 3D point clouds

For our final set of experiments, we follow the setup of Section 7.2 in [5], which uses ModelNet10. We select 80 point clouds from Class 2 (chair) and 80 from Class 7 (sofa), and split them into 40 training and 40 testing samples per class. Each dataset has $40 \times 40$ cross-class pairs. The objective is to learn the third lower bound of the Gromov-Wasserstein distance in [60] (Definition 6.3). We prove in Appendix I.4 that it is continuous with respect to the Wasserstein $p$-distance.

Unlike [5], which downsampled all point clouds to 100 points, we focus on size generalization: training is done on $n_{\text{train}} = 20$, and testing is done on $n_{\text{test}} \in \{20, 100, 200, 300, 500\}$. We compare the size generalization of 3 models: unnormalized SVD-DS, (normalized) SVD-DS and normalized DS-CI. For each pair of inputs $V, V'$, we predict the GW lower bound via:

$$\widehat{\text{GW}}(V, V') = a\|W(f(V) - f(V'))\|^2 + b,$$

where $f : \mathbb{R}^{n \times k} \to \mathbb{R}^t$ is the $\mathsf{S}_n, \mathsf{O}(k)$-invariant model, $t = 10$, and $W \in \mathbb{R}^{t \times t}, a, b \in \mathbb{R}$ are learnable. All DeepSet components $(\sigma, \phi)$ are fully connected ReLU networks. We choose the number of layers and hidden dimensions such that each model has approximately 2k parameters to ensure a fair comparison. Training is performed by minimizing the MSE loss using AdamW with an initial learning rate of $0.01$ and weight decay of $0.1$. Each model is trained for 3000 epochs. Training a single run takes less than 3 minutes.

The experiment results are summarized in Figure 10. Normalized DS-CI and normalized SVD-DS, both aligned with the target at the continuity level, achieve good performance. While normalized SVD-DS underperforms compared to normalized DS-CI, it offers superior time and memory efficiency.

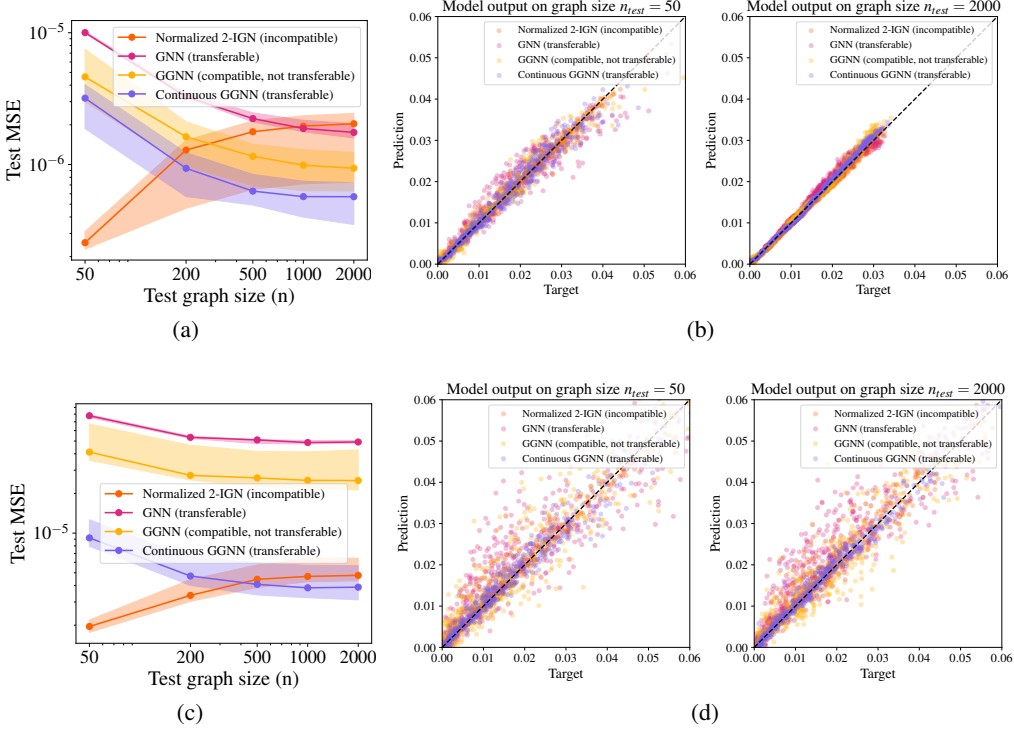

Figure 9: GNN size generalization results on weighted triangle density. **Figures (a)-(b)** show the results on fully connected weighted graphs (the first data generation procedure), and **Figures (c)-(d)** show the results on simple graphs sampled from SBM (the second data generation procedure). Figures (a) and (c) display Test MSE vs. test graph size. The solid line denotes the mean, and the error bar extend from the $20\%$ to $80\%$ percentile test MSE over 10 randomly initialized trainings. Continuous GGNN performs the best for both random graph models. Figures (b) and (d) display Test-set predictions vs. ground truth. The middle columns displays results for graph size $n_{\text{test}} = n_{\text{train}} = 50$, while the right-most column displays results for $n_{\text{test}} = 2000$ and $n_{\text{train}} = 50$.

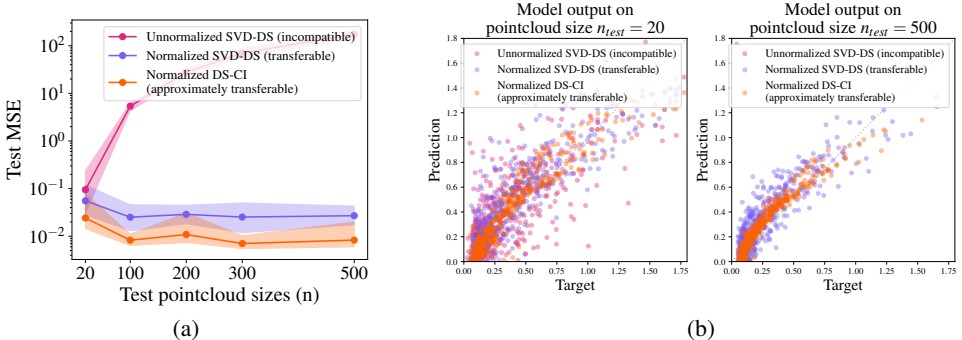

Figure 10: Size generalization results for point clouds. **(a):** Test-set MSE (log scale) vs. point cloud size $n$. The solid line denotes the mean and the error bars extend from the min to max test MSE over 10 trials, each taking the best of 5 random initializations. **(b):** Test-set predictions vs. ground truth for graph size $n_{\text{test}} = n_{\text{train}} = 20$, and $n_{\text{test}} = 500 \gg n_{\text{train}} = 20$.

## I.4 Continuity of Gromov-Wasserstein distance and its third lower bound

The following is based on [60], though these continuity results are not stated there. Let $\mu \in \mathcal{P}_p(\mathbb{R}^k)$ be a probability measure, and associate with it the "metric measure space" $(\mathcal{X}, d_{\mathcal{X}}, \mu)$ where $\mathcal{X} = \text{supp}(\mu)$ and $d_{\mathcal{X}}(x, y) = \|x - y\|_p$. Given two such measures $\mu, \nu \in \mathcal{P}_p(\mathbb{R}^k)$, the

Gromov-Wasserstein distance between their associated metric spaces is defined by

$$\mathfrak{D}_p(\mu,\nu) = \inf_{\pi \in \mathcal{M}(\mu,\nu)} \|\Gamma_{X,Y}\|_{L^p(\pi \otimes \pi)} \tag{32}$$

$$= \inf_{\pi \in \mathcal{M}(\mu,\nu)} \left( \int \left| \|x - x'\|_p - \|y - y'\|_p \right|^p d\pi(x,y)\, d\pi(x',y') \right)^{1/p}, \tag{33}$$

where $\Gamma_{X,Y}(x,y,x',y') = \left| \|x - x'\|_p - \|y - y'\|_p \right|$ and $\mathcal{M}(\mu,\nu)$ is the set of couplings between $\mu$ and $\nu$. The G-W distance admits the following lower bound [60, Def. 6.3].

$$\mathfrak{D}_p \geq \mathrm{TLB}_p(\mu,\nu) = \inf_{\pi \in \mathcal{M}(\mu,\nu)} \|\Omega_{\mu,\nu}\|_{L^p(\pi)},$$
$$\text{where } \Omega_{\mu,\nu}(x,y) = \inf_{\pi' \in \mathcal{M}(\mu,\nu)} \|\Gamma(x,y,\cdot,\cdot)\|_{L^p(\pi')}. \tag{34}$$

We aim to show that both $\mathfrak{D}_p$ and $\mathrm{TLB}_p$ are continuous with respect to the Wasserstein-$p$ metric on $\mathcal{P}_p$. More precisely, the following Lipschitz bounds hold.

**Proposition I.1.** *Let $\mu, \nu, \mu', \nu' \in \mathcal{P}_p(\mathbb{R}^k)$. Then*

$$|\mathfrak{D}_p(\mu,\nu) - \mathfrak{D}_p(\mu',\nu')| \leq W_p(\mu,\mu') + W_p(\nu,\nu'). \tag{35}$$

*Proof.* By the triangle inequality for $\mathfrak{D}_p$, which holds by [60, Thm. 5.1(a)], we have

$$\mathfrak{D}_p(\mu,\nu) \leq \mathfrak{D}_p(\mu,\mu') + \mathfrak{D}_p(\mu',\nu') + \mathfrak{D}_p(\nu,\nu').$$

By [60, Thm. 5.1(d)], we further have $\mathfrak{D}_p(\mu,\mu') \leq W_p(\mu,\mu')$ and similarly for $\mathfrak{D}_p(\nu,\nu')$. Combining these inequalities and interchanging the roles of $(\mu,\nu)$ and $(\mu',\nu')$, we get the claim. $\square$

The above proof only used the triangle inequality and the bound $\mathfrak{D}_p \leq W_p$ for the G-W distance. Since $\mathrm{TLB}_p \leq \mathfrak{D}_p \leq W_p$, the latter property also holds for TLB. Thus, it suffices to prove $\mathrm{TLB}_p$ satisfies the triangle inequality, hence is similarly Lipschitz in $W_p$.

**Lemma I.2** (Triangle inequality for $\Omega_{\mu,\nu}$). *Let $\mu, \nu, \xi \in \mathcal{P}_p(\mathbb{R}^k)$. We have $\Omega_{\mu,\nu}(x,y) \leq \Omega_{\mu,\xi}(x,z) + \Omega_{\xi,\nu}(z,y)$ for all $x, y, z$ in the relevant supports. Furthermore, we have $\mathrm{TLB}_p(\mu,\nu) \leq \mathrm{TLB}_p(\mu,\xi) + \mathrm{TLB}_p(\xi,\nu)$.*

*Proof.* Note that the usual triangle inequality for $\|\cdot\|_p$ gives $\Gamma(x,y,x',y') \leq \Gamma(x,z,x',z') + \Gamma(z,y,z',y')$ for any $x, y, z, x', y', z' \in \mathbb{R}^k$. For any couplings $\pi_1 \in \mathcal{M}(\mu,\xi)$ and $\pi_2 \in \mathcal{M}(\xi,\nu)$, the Gluing Lemma [82, Lemma 7.6] guarantees the existence of a coupling $\pi \in \mathcal{M}(\mu,\xi,\nu)$ whose corresponding marginals are $\pi_1, \pi_2$. Let $\pi_3 \in \mathcal{M}(\mu,\nu)$ be the marginal of $\pi$ on its first and third coordinates. Then

$$\Omega_{\mu,\nu}(x,y)^p \leq \|\Gamma(x,y,\cdot,\cdot)\|^p_{L^p(\pi_3)} = \|\Gamma(x,y,\cdot,\cdot)\|^p_{L^p(\pi)}$$
$$\leq \|\Gamma(x,z,\cdot,\cdot) + \Gamma(z,y,\cdot,\cdot)\|_{L^p(\pi)} \leq \|\Gamma(x,z,\cdot,\cdot)\|_{L^p(\pi_1)} + \|\Gamma(z,y,\cdot,\cdot)\|_{L^p(\pi_2)}.$$

Since this holds for all couplings $\pi_1, \pi_2$ as above, we obtain the first claim.

The second claim is proved analogously. For couplings $\pi_1, \pi_2, \pi_3, \pi$ as above, we have

$$\mathrm{TLB}_p(\mu,\nu) \leq \|\Omega_{\mu,\nu}\|_{L^p(\pi_3)} = \|\Omega_{\mu,\nu}\|_{L^p(\pi)} \leq \|\Omega_{\mu,\xi} + \Omega_{\xi,\nu}\|_{L^p(\pi)} \leq \|\Omega_{\mu,\xi}\|_{L^p(\pi_1)} + \|\Omega_{\xi,\nu}\|_{L^p(\pi_2)},$$

and taking infimums over $\pi_1, \pi_2$ completes the proof. $\square$

**Proposition I.3.** *Let $\mu, \nu, \mu', \nu' \in \mathcal{P}_p(\mathbb{R}^k)$. Then*

$$|\mathrm{TLB}_p(\mu,\nu) - \mathrm{TLB}_p(\mu',\nu')| \leq W_p(\mu,\mu') + W_p(\nu,\nu'). \tag{36}$$

*Proof.* The proof is now identical to that of Proposition I.1. $\square$

The above argument generalizes to measures on different abstract metric spaces, we did not use the fact that all measures involved were over $\mathbb{R}^k$.

## J   Limitations of this work

This work provides a theoretical framework for transferability based on consistent sequences. It applies to several machine learning models that use a fixed number of parameters to define functions on any-dimensional inputs. However, this theory does not capture all possible ways for inputs to grow in dimension. In particular, it does not capture settings where there is no limiting space containing all finite-sized inputs, and where such inputs can be compared. For example, how do we compare natural language inputs of different lengths to each other? Furthermore, while we believe our framework may extend to settings such as images (with varying resolutions and sizes), partial differential equations (across different scales), and sequences (of varying lengths), we did not explore these directions. We leave these investigations for future work. Finally, as discussed briefly in the related work, we do not consider the expressive power of neural networks on the limiting space. If the target function is not expressible, mere alignment in terms of compatibility and continuity—as discussed in Section 4—is insufficient to ensure good performance. Studying universal approximation theory on the limiting space is an important and promising direction for future research.

