# OpenReview forum: "On Transferring Transferability: Towards a Theory for Size Generalization"
_NeurIPS.cc/2025/Conference — NeurIPS 2025 spotlight_

### Official Review · Reviewer_H5KD · 2025-06-29

**Clarity:** 3
**Significance:** 3
**Originality:** 2
**Rating:** 4
**Confidence:** 3

**Summary:**

The paper addresses a key challenge: how models trained with low-dimensional samples transfer across dimension. The authors introduce a unified theoretical framework for transferability: the capacity of a model trained on low-dimensional data to maintain performance when applied to higher-dimensional inputs. It equates transferability across input sizes with continuity in a defined limit space. In this space, inputs of different sizes can be identified and compared via appropriate metrics based on the data and task. The authors propose design principles for building models with provable size-transfer guarantees. They also provide empirical experiments to support their theoretical claims.

**Questions:**

Please refer to questions in weaknesses.

**Ethical Concerns:**

["NO or VERY MINOR ethics concerns only"]

**Limitations:**

Yes

**Quality:**

3

**Strengths And Weaknesses:**

*Strengths*

The paper seems to provide a clear theoretical foundation for a problem of transferability in machine learning by formalizing the transferability as continuity in a limit space.
The paper is well-structured, beginning with intuitive motivation, followed by clear definitions, theoretical results, and practical verifications. The specific examples on sets, graphs, and point clouds as well as additional experiments are provided in appendices.

The paper supports the design principles for building models that generalize across input sizes (e.g., foundation models for graphs, GNN for combinatorial optimization, neural PDE solvers).

*Weaknesses*

I am not a domain expert in size generalization and I am not quite familiar with the existing theoretical literature in this area. As such, I am not in a position to assess the originality of the proposed theoretical contributions with full confidence. My comments below focus specifically on the empirical aspect of the paper.

To verify the theoretical framework, the paper includes several empirical validation. While in Section 6 and Appendix H, experiments for size generalization involving DeepSets, PointNet, and certain GNN variants (e.g., population statistics, maximal distance from the origin, learn a homomorphism density of degree three) are well-designed, the settings tend to be synthetic and controlled. I believe the work would benefit from more experiments under real-world conditions to evaluate the proposed design principles. For example, instantiate the proposed framework on popular architecture types across diverse real-world tasks and larger-scale datasets.

(Minor) What does ‘id’ refer to in line 96 $G_n=\\{id\\}$?

---

> ### Author Rebuttal · Authors · 2025-07-30
>
> Thank you for your constructive feedback. We are pleased that you found the paper well-structured and clearly presented, and that you appreciated the design of our empirical experiments. Below, we address each of the questions raised in detail.
>
> > originality of the proposed theoretical contributions
>
> We provide an original unifying framework for transferability and size generalization that significantly extends prior work on GNN transferability and is applicable to variety of models outside GNNs. This is supported by Reviewer ZWs5's comments: "Framing transferability as continuity in a suitably defined limit space, which could come from any any-dimensional modality, offers a principled perspective on size generalization that, to the best of my knowledge, is not present in the existing literature."
>
> Our additional contributions include introducing techniques to derive new transferability proofs, designing new transferable models (continuous GGNN and SVD-DS), and suggesting principled ways to modify existing models to achieve transferability. Furthermore, we point out the connection between this general form of transferability and generalization bounds, leading to a new formulation of generalization that has been missing in prior studies on GNN generalization.
> This work also highlights and formalizes the importance of task-model alignment for achieving size generalization. Additionally, we identify general cases that offer provable size generalization for all $n$---a strong result in size generalization that, to our knowledge, is absent in previous work.
>
> > (Minor) What does ‘id’ refer to in line 96 $G_n=\\{id\\}$?
>
> 'id' here refers to the identity element in a group. In this case, $G_n=\\{id\\}$ means that $G_n$ is the trivial group, containing only the identity element. This is mentioned in the Notation section (Appendix A, line 892 in the supplementary material), We will clarify this in the main paper.
>
> > I believe the work would benefit from more experiments under real-world conditions to evaluate the proposed design principles. For example, instantiate the proposed framework on popular architecture types across diverse real-world tasks and larger-scale datasets.
>
> We thank the reviewer for the suggestion, but have respectfully elected not to pursue real-world experiments. Because our paper is primarily a theoretical contribution, the role of our numerical examples was twofold: (1) to demonstrate that practical performance tracks our theoretical predictions and (2) to validate our conjecture that size generalization depends on the alignment between the task and the model’s inductive bias. Achieving both objectives requires settings in which the instance sizes grow to infinity while the learning tasks satisfy our assumptions. Although we believe these assumptions can hold in practice, we are not aware of any real-world datasets whose instances grow arbitrarily large. We would be happy to apply our framework to a real-data example if you could suggest a relevant dataset and learning task.

---

### Official Review · Reviewer_o4rd · 2025-07-01

**Clarity:** 3
**Significance:** 3
**Originality:** 3
**Rating:** 5
**Confidence:** 4

**Summary:**

This work presents a unified framework for analyzing the size generalization of machine learning models that process variable-sized inputs, such as sets, graphs, and point clouds. The key idea is that transferability across sizes can be characterized by a notion of model continuity within a limit space. This limit space is constructed from a sequence of spaces of different sizes, enabling a meaningful comparison of distances across varying dimensions (so-called consistent sequences). In this framework, stronger model continuity directly implies smaller size generalization errors. The paper applies this perspective to analyze the transferability of popular architectures, including DeepSets, PointNet, and various graph neural networks, thereby offering a unified explanation of their size-generalization behavior. Finally, it proposes design principles for developing new models with provable transferability guarantees.

**Questions:**

N/A

**Ethical Concerns:**

["NO or VERY MINOR ethics concerns only"]

**Limitations:**

Yes

**Quality:**

4

**Strengths And Weaknesses:**

**Strengths**
- This work provides a unified framework for the size generalization problem in different domains (sets, graphs, point clouds).
- It also provides intuition on how specific tasks may favor models with particular continuity properties. This theoretical insight is supported by experimental validation, which can offer practical guidance for selecting or designing models based on prior task knowledge.
- Although the paper is somewhat dense in terms of formal notions, it is well-motivated, clearly presented, and successfully conveys the underlying intuition.


**Weaknesses**
- The empirical evaluation is largely limited to synthetic dataset and synthetic task.

---

> ### Author Rebuttal · Authors · 2025-07-30
>
> Thank you very much for your positive feedback. We are glad you appreciated our unified framework, theoretical insights, and clear presentation.
>
> Regarding your comment that "the empirical evaluation is largely limited to synthetic dataset and synthetic task," we acknowledge that is the case. However, we decided not to pursue real-world experiments. Because our paper is primarily a theoretical contribution, the role of our numerical examples was twofold: (1) to demonstrate that practical performance tracks our theoretical predictions and (2) to validate our conjecture that size generalization depends on the alignment between the task and the model’s inductive bias. Achieving both objectives requires settings where instance sizes grow to infinity while still satisfying our assumptions. Although we believe these assumptions can hold in practice, we are unaware of real-world datasets whose instances grow arbitrarily large. We would happily apply our framework to a real-data example if you could suggest a relevant dataset and learning task.

---

> > ### Comment · Reviewer_o4rd · 2025-08-05
> > **Official Comment by Reviewer o4rd**
> >
> > Thank you for your explanation. I agree that a real-world dataset with arbitrarily large instances would be challenging to find. I will maintain my score for acceptance.

---

### Official Review · Reviewer_bCzQ · 2025-07-02

**Clarity:** 2
**Significance:** 2
**Originality:** 2
**Rating:** 3
**Confidence:** 3

**Summary:**

This paper introduces a rigorous and mathematically elegant framework to analyze when and how machine learning models can generalize across varying input sizes—a property referred to as transferability across dimensions.

**Questions:**

see weaknesses

**Ethical Concerns:**

["NO or VERY MINOR ethics concerns only"]

**Final Justification:**

I would like to thank author for their final response. considering the current response. My final decision is to keep my score unchanged.

> We respectfully disagree with the reviewer’s view that this assumption is unrealistic in practice. As we have mentioned in the previous reply, standard loss functions are bounded when the predictions and target outputs are bounded. This can usually be assumed in practice, especially since we assume a compact domain. We are not entirely sure about the precise meaning of "heavy-tailed loss distributions" as referenced by the reviewer. We can discuss or clarify further if the reviewer could provide references related to theoretical analysis in this context. Meanwhile, we would also like to emphasize again that the primary focus of our work is on size generalization. In our view, specific considerations related to the choice of loss function is not relevant to the contributions of our study.

There are many works on generalisation error of heavy-tailed loss function where can be applied here. please check these works [1,2,3,4].


> We would like to emphasize that the key contribution of this paper extends beyond presenting a new generalization bound (which occupies just half a page in Section 4 of the main paper). More importantly, it lies in the theoretical framework developed in Sections 2 and 3. This unifying framework enables the comparison of objects of varying sizes within a limit space, and the construction of the limit space and the associated distance metric followed from the consistent sequence structure. Without this construction, it would not be possible to meaningfully analyze Lipschitz continuity or gain insights into generalization behaviors involving objects of different sizes. Our generalization bound formalizes the scenarios where size generalization is achievable, rather than aiming to improve upon any existing bounds.

I think the contribution is incremental in this sense. The authors should clarify the main difference between their work and many works which are focused on generalization error and why it is interesting to study this problem.


**References:**

[1]: Alquier, P. and Guedj, B. Simpler PAC-Bayesian bounds for hostile data. Machine Learning, 107(5):887–902, 2018.

[2]: Cortes, C., Greenberg, S., and Mohri, M. Relative deviation
learning bounds and generalization with unbounded loss
functions. Annals of Mathematics and Artificial Intelli-
gence, 85:45–70, 2019.

[3]: Lugosi, G. and Neu, G. Generalization bounds via convex
analysis. In Conference on Learning Theory, pp. 3524–
3546. PMLR, 2022.

[4]: Haddouche, M. and Guedj, B. Pac-bayes generalisation
bounds for heavy-tailed losses through supermartingales.
arXiv preprint arXiv:2210.00928, 2022.

**Limitations:**

see weaknesses

**Quality:**

3

**Strengths And Weaknesses:**

## Strengths

- The paper makes a strong theoretical contribution, offering a unifying perspective on size generalization across different data domains.
- It is well-written and carefully structured, with illustrative examples and multiple scenarios discussed across settings like GNNs, DeepSets, and PointNet.

## Weaknesses and Questions

- **Notation and Clarity:** The paper is notation-heavy and may be difficult to follow for non-specialists. Although the general framework is clear, the narrative flow can be hard to track. Additionally, while GNNs are discussed in detail, it is unclear whether the focus is on node classification, graph classification, or other tasks, which could confuse readers unfamiliar with these distinctions.
- **Conceptual Clarification:** The distinction between transferability (as defined in this paper) and more established concepts such as transfer learning and meta-learning is not clearly articulated. Furthermore, it is unclear how this notion of transferability relates to the broader problem of distribution shift between training and test sets.
- **Bounded Loss Assumption:** Section 4 introduces the assumption of a bounded and Lipschitz-continuous loss function, which plays a central role in the generalization bound. However, the justification for this assumption is missing and its practical relevance is not discussed.
- **Complex Assumptions:** Assumption 4.1 is quite dense, combining multiple technical conditions. While mathematically sound, such compound assumptions reduce readability and are uncommon in theoretical learning papers without further breakdown or justification.
- **Convergence Rate Condition:** It is stated that  $\xi^{-1}(N)\rightarrow 0$ for $N\rightarrow \infty$, but no intuition or explanation is provided. Clarifying the role of this term and its dependence on the problem setup would enhance the theoretical transparency.
- **Proposition E.3:** Why for $n\rightarrow \infty$ the upper bound in (7) goes to 0?

---

> ### Author Rebuttal · Authors · 2025-07-30
>
> Thank you very much for the constructive feedback. While we would like to point out that some of the aspects you mentioned are indeed addressed in the Appendix, we understand that the length of the manuscript may have made these points less visible, and we appreciate you bringing these concerns to our attention. We will make an effort to clarify and include pointers. Below, we address each of the concerns and comments you posed.
>
> > Notation and Clarity: The paper is notation-heavy and may be difficult to follow for non-specialists. Although the general framework is clear, the narrative flow can be hard to track. Additionally, while GNNs are discussed in detail, it is unclear whether the focus is on node classification, graph classification, or other tasks, which could confuse readers unfamiliar with these distinctions.
>
> Our goal was to convey the general framework ideas to non-specialists while being organized and mathematically rigorous for specialists. We appreciate that you found the framework clear. We have made significant efforts to simplify the notation and definitions in the body of the paper and provide intuitive explanations to aid understanding, while preserving the detailed and rigorous version with additional notation in the Appendix. Unfortunately, it is challenging for us to simplify the presentation further since clear definitions are essential, and precise notation is necessary to achieve that clarity. To further improve accessibility for non-specialist readers, **we will include illustrations in the Appendix** that demonstrate how the framework applies to sets and graphs. These illustrations will visualize consistent sequences and sampling from limiting objects. Due to this year's rules, we cannot include them in this rebuttal.
>
> Our theoretical framework is general enough to apply to both node classification/regression tasks and graph classification/regression tasks; therefore, we did not explicitly distinguish between the two in the main paper. This distinction is addressed in lines 1025-1028 of the Appendix, where we explain that for graph-level tasks, we can simply use a trivial consistent sequence (one that is not size-dependent) to model the output space. After this small adjustment, all the analysis is identical for both node-level and graph-level tasks, so we do not emphasize this distinction further. We can add a comment about this to the body of the paper if the reviewer would like us to.
>
> > Conceptual Clarification: The distinction between transferability (as defined in this paper) and more established concepts such as transfer learning and meta-learning is not clearly articulated. Furthermore, it is unclear how this notion of transferability relates to the broader problem of distribution shift between training and test sets.
>
> We acknowledge the potential confusion that may arise from similar concept names. The term "transferability" in the paper specifically extends the concept of "GNN transferability," which is a well-established concept in the context of GNNs. GNN transferability refers to the phenomenon in which a GNN produces similar outputs for two (potentially differently sized) graphs sampled from the same underlying graphon, enabling performance transfer from smaller graphs to larger graphs. For more details, see [1], which discusses this concept as a pioneering work. Additional background information is provided in our related work section (lines 912-923 in the Appendix).
>
> We note that while GNN transferability is somewhat related to transfer learning, it is not the same. Transfer learning typically involves pretraining a model on one task and fine-tuning it for a different, new task. In contrast, GNN transferability involves training a model on smaller graphs and testing it on larger graphs---usually without any fine-tuning---and obtaining predictable outputs, provided the graphs are sampled from the same underlying graphon. To our knowledge, GNN transferability is unrelated to meta-learning since it only considers one task. **We will add a footnote to address this point.**
>
> **Our notion of transferability addresses a specific type of distribution shift (size generalization)**: we assume there exists an underlying latent distribution $\mu$ on an infinite-dimensional limit space (e.g. graphon space). This latent distribution induces a distribution $\mu_n$ on the space of size-$n$ objects (e.g. graphs with $n$ nodes) via sampling or discretization. When training on size-$n$ graphs distributed according to $\mu_n$, and testing on distribution $\mu$ on the limit space, a distribution shift occurs. However, this shift is "similar" enough to allow for meaningful generalization. This phenomenon is formally described in detail starting from line 1333 in Appendix E. We highlight that the explicit connection between transferability and the distribution shift underlying the generalization bounds is a novel contribution of our work. The closest work include [3,4], but they do not consider distribution shifts since the training and test are done on the same distribution. We will highlight this point in the main text.
>
> > Bounded Loss Assumption: Section 4 introduces the assumption of a bounded and Lipschitz-continuous loss function, which plays a central role in the generalization bound. However, the justification for this assumption is missing and its practical relevance is not discussed.
>
> Thank you for pointing this out. We highlight that this type of assumption is relatively standard in statistical learning theory (See e.g. [5,6,7]). **We will include the following justification in the next revision.** "We note that these assumptions are relatively standard [5,6,7]. Moreover, when the predictions and target outputs are bounded, standard loss functions, such as cross-entropy, L1-loss, L2-loss, and Huber loss, are all bounded and Lipschitz continuous. Moreover, many clipped loss functions also satisfy this assumption."
>
> > Complex Assumptions: Assumption 4.1 is quite dense, combining multiple technical conditions. While mathematically sound, such compound assumptions reduce readability and are uncommon in theoretical learning papers without further breakdown or justification.
>
> We agree with the reviewer that this is a rather technical assumption. Indeed, in Appendix E, we break down this assumption and provide detailed explanations and motivation to make it more readable and accessible. The explanations in Appendix E are organized into two parts: In the first part, we describe how directly applying the classical generalization framework from [2] to our defined limit space immediately yields a generalization bound that holds across any dimensions. In the second part, we discuss the limitations of this classical setup, explaining why it is unrealistic in certain scenarios. This motivates us to slightly modify the assumptions to account for the distribution shift between the training data and the test data. These discussions in Appendix E are intended to clarify and justify the assumptions in a more detailed manner for interested readers. Unfortunately, due to space constraints, we had to write Assumption 4.1 in a dense format to ensure that Section 4 remains concise, leaving more space for our theoretical framework established in the earlier sections, which are our primary contributions. **We will add a pointer directing the interested reader to Appendix E after Assumption 4.1.**
>
> > Convergence Rate Condition: It is stated that $\xi^{-1}(N)\rightarrow 0$ for $N\rightarrow \infty$, but no intuition or explanation is provided. Clarifying the role of this term and its dependence on the problem setup would enhance the theoretical transparency.
>
> We thank the reviewer for their comment. We realize that although we explained why $\xi^{-1}(N) \to 0$ in lines 1311-1313 of the Appendix, this explanation might be overlooked by readers because the generalization bound results are divided into two separate propositions, Proposition E.2 and Proposition E.3, in the Appendix. The remark appears in the earlier Proposition, where $\xi^{-1}$ is first introduced. We will make this clearer in the next revision.
>
> > Proposition E.3: Why for $n\rightarrow \infty$ the upper bound in (7) goes to 0?
>
> The upper bound in (7) does not go to $0$ as $n \to \infty$ alone; it goes to $0$ if **both** $n$ (the dimension of the training data) and $N$ (the number of data points in the training data) go to $\infty$. We mentioned this in Remark E.4. The result is because every term in the upper bound in (7) converges to $0$: we justified $\xi^{-1}(N)\to 0$ as $N\to\infty$ in lines 1311-1313; Moreover, $R(n)\to 0$ as $n\to\infty$ by the assumption in line 1348, and $\frac{2\log(1/\delta)}{N}\to 0$ as $N\to\infty$ trivially.
>
> References:
>
> [1] Ruiz, Luana, Luiz Chamon, and Alejandro Ribeiro. "Graphon neural networks and the transferability of graph neural networks." Advances in Neural Information Processing Systems 33 (2020): 1702-1712.
>
> [2] Xu, H. and Mannor, S., 2012. Robustness and generalization. Machine learning, 86(3), pp.391-423.
>
> [3] Levie, R., 2023. A graphon-signal analysis of graph neural networks. Advances in Neural Information Processing Systems, 36, pp.64482-64525.
>
> [4] Rauchwerger, L., Jegelka, S. and Levie, R., 2024. Generalization, expressivity, and universality of graph neural networks on attributed graphs. arXiv preprint arXiv:2411.05464.
>
> [5] Bach F. Learning theory from first principles. MIT press; 2024 Dec 24.
>
> [6] Shalev-Shwartz, S. and Ben-David, S., 2014. Understanding machine learning: From theory to algorithms. Cambridge university press.
>
> [7] Bartlett, P.L. and Mendelson, S., 2002. Rademacher and gaussian complexities: Risk bounds and structural results. Journal of machine learning research, 3(Nov), pp.463-482.

---

> > ### Comment · Area_Chair_rsCg · 2025-08-05
> >
> > Reviewer  bCzQ,
> > Could you please comment on the authors' rebuttal, as some reviewers seem to have a much more positive perception of the paper after the rebuttal? Has their rebuttal settled any of your reservations?
> > Thank you for your work so far.

---

> > > ### Comment · Reviewer_bCzQ · 2025-08-06
> > > **Response**
> > >
> > > Apologies for the delayed response. Thanks for your rebuttal.
> > >
> > > ---
> > >
> > > > **Proposition E.3 — Limit behaviour of the bound**
> > >
> > > My understanding is that one can first invoke the triangle inequality given in **Remark E.2**, and then apply the standard generalisation-error bound separately to each of the two terms on the right-hand side.
> > > Is this interpretation correct?
> > >
> > > I see the key contribution of the paper in **bounding the second term** via \(W_1(\mu,\mu_n)\) and then analysing the Wasserstein distance under different sampling schemes. The treatment of the **first term** seems more routine by comparison.
> > >
> > > ---
> > >
> > > > **Related work**
> > >
> > > Because the framework is applied to **node-classification** and **graph-classification**, it would be helpful to discuss prior generalisation-error results specific to those settings; several relevant papers are currently missing from the related-work section.
> > >
> > > ---
> > >
> > > > **Bounded-loss assumption**
> > >
> > > Assuming a bounded loss is convenient but unrealistic for practical choices such as the **cross-entropy** loss. A discussion (or extension) covering **heavy-tailed** loss distributions would make the contribution stronger; under the present bounded-loss assumption, the novelty appears limited.

---

> > > > ### Author Response · Authors · 2025-08-08
> > > >
> > > > Thanks for the reply. We hope that our replies below address the concerns raised, and we are happy to further clarify. We would appreciate it if the reviewer could reconsider their evaluation of the paper.
> > > >
> > > > > My understanding is that one can first invoke the triangle inequality given in Remark E.2, and then apply the standard generalisation-error bound separately to each of the two terms on the right-hand side. Is this interpretation correct?
> > > >
> > > > Yes, this is correct. However, as there is no triangle inequality in Remark E.2, we assumed this is a typo, and you are instead referring to the equation just below line 1358, following Remark E.4.
> > > >
> > > > > I see the key contribution of the paper in bounding the second term via (W_1(\mu,\mu_n)) and then analysing the Wasserstein distance under different sampling schemes. The treatment of the first term seems more routine by comparison.
> > > >
> > > > We would like to emphasize that the key contribution of this paper extends beyond presenting a new generalization bound (which occupies just half a page in Section 4 of the main paper). More importantly, it lies in the theoretical framework developed in Sections 2 and 3. This unifying framework enables the comparison of objects of varying sizes within a limit space, and the construction of the limit space and the associated distance metric followed from the consistent sequence structure. Without this construction, it would not be possible to meaningfully analyze Lipschitz continuity
> > > > or gain insights into generalization behaviors involving objects of different sizes. Our generalization bound formalizes the scenarios where size generalization is achievable,
> > > > rather than aiming to improve upon any existing bounds.
> > > >
> > > > > Because the framework is applied to node-classification and graph-classification, it would be helpful to discuss prior generalisation-error results specific to those settings; several relevant papers are currently missing from the related-work section.
> > > >
> > > > As discussed in the related work section, and to the best of our knowledge, most prior generalization bound results for GNNs are, first, restricted to finite or bounded sizes,  and second, do not account for distribution shifts between the training and test sets induced by changes of size. We believe these results are **not closely related** to ours, as our work focuses on objects of unbounded size and primarily addresses the setting of training on small-sized objects and testing on the limit space asymptotically. We have included discussion of works on generalization involving **unbounded sizes** that we are aware of, and we would be happy to address any missing relevant works if the reviewer can provide references.
> > > >
> > > > > Assuming a bounded loss is convenient but unrealistic for practical choices such as the cross-entropy loss. A discussion (or extension) covering heavy-tailed loss distributions would make the contribution stronger; under the present bounded-loss assumption, the novelty appears limited.
> > > >
> > > > We respectfully disagree with the reviewer’s view that this assumption is unrealistic in practice. As we have mentioned in the previous reply, standard loss functions are bounded when the predictions and target outputs are bounded. This can usually be assumed in practice, especially since we assume a compact domain. We are not entirely sure about the precise meaning of "heavy-tailed loss distributions" as referenced by the reviewer. We can discuss or clarify further if the reviewer could provide references related to theoretical analysis in this context. Meanwhile, we would also like to emphasize again that the primary focus of our work is on **size** generalization. In our view, specific considerations related to the choice of loss function is not relevant to the contributions of our study.

---

### Official Review · Reviewer_ZWs5 · 2025-07-02

**Clarity:** 4
**Significance:** 3
**Originality:** 3
**Rating:** 5
**Confidence:** 4

**Summary:**

This paper develops a general theoretical framework for transferability based on the notion of consistent sequences. This means modeling any-dimensional learning problems by identifying smaller instances of data with larger ones in a structured way, which results in a nested sequence of problem instances. These sequences are embedded into a common limit space, where transferability is then characterized by continuity. For graphs, using the duplication embedding recovers the classical graphon limit, which much of the prior work on GNN transferability is based on. The central goal of the paper is to generalize and unify these concepts, i.e., to simplify existing analyses and extend them beyond graphs. The authors apply their framework to sets, graphs (MPNNs/IGNs), and point clouds, showing how different architectures do or do not satisfy the conditions for transferability. A model is transferable if it defines a continuous function on the limit space and is compatible with the embeddings used. The paper also presents a size generalization theorem. The theory is supported by experiments on transferability and generalization across all modalities.

**Questions:**

- Re Proposition 4.2 (generalization bound): It might be helpful to elaborate a bit more on the assumptions and their practicality. As the rate crucially depends on the covering numbers $C_X(\varepsilon), C_Y(\varepsilon)$, what can we expect these to be for the different modalities covered in the paper? I.e., do you expect this bound to be non-vacuous in some concrete cases?
- In line 2165, you state that your continuous GGNN (as a variant of 2-IGN) is "likely more expressive than the GNN". To my knowledge, the 2-IGN you consider, at least for graph-level predictions, should be exactly 2-WL (not 2-FWL) expressive, and thus no more expressive than MPNNs (see, e.g., [1] or § 2.3.3 of [2]). I'd appreciate if you could check and clarify this point.
- Re the size generalization experiment in § I.2: Compared to the experiments for other modalities, the GNN results in Figure 7 appear slightly more noisy/inconclusive to me. Also, in line with the previous suggestion, I would not have expected any of the models you considered to be able to express this triangle density on the graph distributions you use. Some intuition at the end of § I.2 is provided, attributing this to average case instead of the worst-case setting that is often considered in the GNN expressivity literature, but it might still be interesting to find out what is actually learned. That said, I acknowledge this is only tangential to the core message of the paper.

[1] Floris Geerts. *The expressive power of kth-order invariant graph networks.* arXiv:2007.12035

[2] Stefanie Jegelka. *Theory of Graph Neural Networks: Representation and Learning.* arXiv:2204.07697

**Ethical Concerns:**

["NO or VERY MINOR ethics concerns only"]

**Final Justification:**

As the authors engaged meaningfully with my rebuttal (which had only raised minor points), the rebuttal has not really changed my positive view of the work. I think it will be a valuable addition to the conference and I am happy to recommend acceptance.

**Limitations:**

Yes

**Quality:**

4

**Strengths And Weaknesses:**

- In my view, the contribution is simple, elegant, and insightful. Framing transferability as continuity in a suitably defined limit space, which could come from any any-dimensional modality, offers a principled perspective on size generalization that, to the best of my knowledge, is not present in the existing literature. Rather than providing new heavily technical results or improving existing (tight) bounds, to me the contribution stands out more in clarity and unification of prior ideas.
- The paper is beautifully written (including the proofs) and quite easy to follow despite the theoretical depth. It really was a pleasure to read.
- Related work is extensively discussed, and off the top of my head, I couldn't think of any relevant prior work that might be missing. The placement of the work within the literature is quite clear.
- I do not have significant criticisms of the paper (for minor things, see Questions). Some may wish for more technical novelty, but I think that the paper succeeds in its aims.

---

> ### Author Rebuttal · Authors · 2025-07-30
>
> Thank you for the thorough review of our manuscript and your encouraging comments. It is rewarding to hear that you found the paper a pleasure to read and appreciated its clarity and theoretical insights. We also value the thoughtful questions raised, which we will address below.
>
> > Re Proposition 4.2 (generalization bound): It might be helpful to elaborate a bit more on the assumptions and their practicality. As the rate crucially depends on the covering numbers $C_X(\varepsilon) ,C_Y(\varepsilon)$, what can we expect these to be for the different modalities covered in the paper? I.e., do you expect this bound to be non-vacuous in some concrete cases?
>
> We briefly discussed two concrete cases where this bound is not vacuous, in Example E.5 and Example E.6 in the Appendix (lines 1367-1384). Example E.5 concerns the case of sets endowed with the normalized $\ell_{p}$ metric, which is related to the space of probability measures with the Wasserstein metric. Here, if $X$ is the space of probability measures supported on a fixed compact set, it is known that $X$ is compact, with an upper bound for $C_X$ provided in a cited result. Similarly, Example E.6 addresses the case of graphs endowed with the cut distance. In this context, for $X$ defined as a nice space of graphon signals, a cited result likewise determines a bound for the covering number $C_X$.
>
> We acknowledge that this discussion is a bit hidden and buried within the technical details, which makes it easily overlooked by readers. To enhance readability, we will add pointers and remarks.
>
> > In line 2165, you state that your continuous GGNN (as a variant of 2-IGN) is "likely more expressive than the GNN". To my knowledge, the 2-IGN you consider, at least for graph-level predictions, should be exactly 2-WL (not 2-FWL) expressive, and thus no more expressive than MPNNs (see, e.g., [1] or § 2.3.3 of [2]). I'd appreciate if you could check and clarify this point.
>
> To clarify, we are comparing the continuous GGNN with the GNN in [3]. Continuous GGNN is at least as expressive as GNN: in equation (20) (just below line 1860), if we set all the $\alpha$'s to $0$ except $\alpha_1 = 1$, and all the $\theta$'s to $0$ except $\Theta_{1,s}$, then we exactly recover the GNN. This result naturally follows the definition of nonlinearity (see equation (19)). We claimed that a continuous GGNN is likely more expressive than a GNN. This claim is supported by our size generalization experiment on triangle density, where continuous GGNN clearly outperforms GNN with a comparable number of parameters, both in terms of in-distribution and size generalization. We agree that the claim is not rigorous since we did not investigate the theoretical expressive power. We will change the word "likely" to "possibly".
>
> Regarding the k-WL expressivity perspective, we think there is an exciting direction to explore here.  There is a conceptual difference between the standard k-WL expressivity and the expressive power of graphon-compatible GNNs (i.e., GNNs that are compatible with the duplication-consistent sequence and thus extend to graphon space). This is because the different-sized graphs corresponding to the same step graphon are always distinguishable by WL, but considered equivalent by any graphon-compatible GNNs. Instead, it is necessary to consider a variant of k-WL specifically designed for graphons (see [4]). Similar conclusions hold: MPNNs with mean aggregation that are graphon-compatible are at most 1-WL expressive, while the graphon-compatible variant of 2-IGN achieves 2-WL expressivity [5]. These results may suggest that these architectures have equivalent expressive power in terms of k-WL graphon distinguishability. On the other hand, a more fine-grained understanding of the potential difference of their expressive power requires additional work, which we leave for future research.
>
> > Re the size generalization experiment in § I.2: Compared to the experiments for other modalities, the GNN results in Figure 7 appear slightly more noisy/inconclusive to me. Also, in line with the previous suggestion, I would not have expected any of the models you considered to be able to express this triangle density on the graph distributions you use. Some intuition at the end of § I.2 is provided, attributing this to average case instead of the worst-case setting that is often considered in the GNN expressivity literature, but it might still be interesting to find out what is actually learned. That said, I acknowledge this is only tangential to the core message of the paper.
>
> We agree that the GNN size generalization results in Figure 7 are less conclusive than others. As you pointed out, this is partly due to an expressivity issue. While we prove that transferable GNNs extend to graphons and are continuous with respect to the cut distance, this does not imply that they can approximate **all** continuous functions with respect to the cut distance. Indeed, this is known not to be the case for GNNs. In particular, none of the models we considered could express triangle density exactly.
>
> Unfortunately, we do not have a clear explanation for this issue beyond what we included in the original manuscript. To the best of our knowledge, there is no average-case theoretical analysis that captures this expressivity, i.e., whether GNNs can **approximate** homomorphism densities with good precision in expectation under certain distributional assumptions. We believe that this would be an interesting direction for future research.
>
> References:
>
> [3] Ruiz, Luana, Luiz Chamon, and Alejandro Ribeiro. "Graphon neural networks and the transferability of graph neural networks." Advances in Neural Information Processing Systems 33 (2020): 1702-1712.
>
> [4] Böker, J., Levie, R., Huang, N., Villar, S. and Morris, C., 2023. Fine-grained expressivity of graph neural networks. Advances in Neural Information Processing Systems, 36, pp.46658-46700.
>
> [5] Herbst, Daniel, and Stefanie Jegelka. "Higher-order graphon neural networks: Approximation and cut distance." arXiv preprint arXiv:2503.14338 (2025).

---

> > ### Comment · Reviewer_ZWs5 · 2025-08-04
> > **Response by Reviewer**
> >
> > I thank the authors for their detailed rebuttal; all of my points have been adequately addressed. I apologize for having overlooked Examples E.5/E.6 in the appendix; but I agree with the authors that it wouldn't hurt to reference them explicitly in the main text. Likewise, I agree that the expressivity aspects could be an interesting future direction. I maintain my positive assessment of the paper and will keep my score unchanged.

---

### Decision · Program_Chairs · 2025-09-17

**Decision:**

Accept (spotlight)

**Comment:**

In this (theoretical) paper, the authors introduce a general framework for understanding how machine learning models trained on small inputs can generalize to larger inputs (transferability). They show that transferability is mathematically equivalent to the continuity of the model's function within a specially constructed "limit space," which allows for the comparison of objects of different sizes. Based on this theory, the paper analyzes existing models for sets, graphs, and point clouds, proposes modifications to ensure they are transferable, and provides design principles for creating new models with provable size-generalization guarantees.

Most of the reviewers praised the paper  and the proposed approach for its elegance and ease of flow (reviewers ZWs5, o4rd and H5KD). Overall, the proofs are well structured, and the approach is original and compelling --- raising very few questions during the rebuttal period.

Only one reviewer (bCzQ) raised concerns on the novelty of the approach and its assumptions. In particular, one of the major sources of contention seems to be the fact that the authors use the fact that the loss is bounded. This concern has been addressed by the authors (who claim that the input space is bounded). The reviewer references work with heavy tail losses, but the pertinence of these references in the context of this discussion is dubious. This reviewer is also the only one who argues that: "I think the contribution is incremental in this sense. The authors should clarify the main difference between their work and many works which are focused on generalization error and why it is interesting to study this problem". Since this is a highly unspecific request and since (as per the other reviewers) the motivation of the paper is clear, we have no choice but to down-weight the significance of reviewer bCzQ's comment.